

# Characterization of Organosulfates in Secondary Organic Aerosol Derived from the Photooxidation of Long-Chain Alkanes

M. Riva[1], T. Da Silva Barbosa[2,3], Y.-H. Lin[1,a], E. A. Stone[4], A. Gold[1], and J. D. Surratt[1,*]

[1]Department of Environmental Sciences and Engineering, Gillings School of Global Public Health, The University of North Carolina at Chapel Hill, Chapel Hill, NC, USA

[2] CAPES Foundation, Brazil Ministry of Education, Brasilia, DF 70.040-020, Brazil

[3]Departamento de Química, Instituto de Ciências Exatas, Universidade Federal Rural do Rio de Janeiro, Seropédica, Brazil

[4]Department of Chemistry, University of Iowa, Iowa City, IA 52242, United States

[a] now at: Michigan Society of Fellows, Department of Chemistry, University of Michigan, Ann Arbor, MI, USA

* To whom correspondence should be addressed.

Jason D. Surratt, Department of Environmental Sciences and Engineering, Gillings School of Global Public Health, University of North Carolina at Chapel Hill, Chapel Hill, NC 27599 USA. Tel: 1-(919)-966-0470; Fax: (919)-966-7911; Email: surratt@unc.edu

The authors declare no conflict of interest.



## Abstract

We report the formation of aliphatic organosulfates (OSs) in secondary organic aerosol (SOA) from the photooxidation of $C_{10} - C_{12}$ alkanes. The results complement those from our laboratories reporting the formation of OSs and sulfonates from gas-phase oxidation of polycyclic aromatic hydrocarbons (PAHs). Both studies strongly support formation of OSs from gas-phase oxidation of anthropogenic precursors, hypothesized on the basis of recent field studies in which aromatic and aliphatic OSs were detected in fine aerosol collected from several major urban locations. In this study, dodecane, cyclodecane and decalin, considered to be important SOA precursors in urban areas, were photochemically oxidized in an outdoor smog chamber in the presence of either non-acidified or acidified ammonium sulfate seed aerosol. Effects of chemical structure, acidity and relative humidity on OS formation were examined. Aerosols collected from all experiments were characterized by ultra performance liquid chromatography coupled to electrospray ionization high-resolution quadrupole time-of-flight mass spectrometry (UPLC/ESI-HR-QTOFMS). Most of the OSs identified could be explained by formation of gaseous epoxide precursors with subsequent acid-catalyzed reactive uptake onto sulfate aerosol. The OSs identified here were also observed and quantified in fine urban aerosol samples collected in Lahore, Pakistan, and Pasadena, USA. Many of the OSs identified from the photooxidation of decalin and cyclodecane are isobars of known monoterpene organosulfates, and thus care must be taken in the analysis of alkane-derived organosulfates in urban aerosol.





## 1. Introduction


Atmospheric fine particulate matter (PM$_{2.5}$, aerosol with aerodynamic diameter $\leq 2.5$
µm) plays a major role in scattering and absorption of solar radiation, which impacts global
climate (Kroll and Seinfeld, 2008; Stevens and Boucher, 2012). PM$_{2.5}$ also participates in
heterogeneous chemical reactions, affecting the abundance and distribution of atmospheric
trace gases (Hallquist et al., 2009). Human exposure to PM$_{2.5}$ is associated with respiratory
and cardiovascular diseases (Elder and Oberdorster, 2006).
Typically, the largest mass fraction of PM$_{2.5}$ is organic, dominated by secondary
organic aerosol (SOA) formed by the oxidation of volatile organic compounds (VOCs).
Although SOA contributes a large fraction (20–90%, depending on location) of total PM$_{2.5}$
mass, current models predict less SOA than is generally observed during field measurements
(Kroll and Seinfeld, 2008; Hallquist et al., 2009). The underestimate is largely a consequence
of omission of intermediate volatility organic compounds (IVOC), such as alkanes or
polycyclic aromatic hydrocarbons (PAHs) (Pye and Pouliot, 2012; Tkacik et al., 2012) as
SOA precursors. Long-chain alkanes are important anthropogenic pollutants emitted by
combustion and vehicular sources representing up to 90% of the anthropogenic emissions in
certain urban areas (Fraser et al., 1997, Gentner et al., 2012). In the atmosphere, they are
rapidly depleted by reaction with OH and NO$_3$ radicals (Atkinson, 2000) yielding a large
variety of oxygenated compounds (Lim and Ziemann, 2005; 2009; Yee et al., 2012), which
could lead to SOA formation (Lim and Ziemman, 2009; Loza et al., 2014). SOA yields have
been measured for C$_7$-C$_{25}$ alkanes having linear, branched and cyclic structures (Lim and
Ziemman, 2009; Presto et al., 2010; Loza et al., 2014; Hunter et al., 2014). Structure plays a
key role in SOA yield, which increases with carbon number or the presence of cyclic features
and tends to decrease with branching as gas-phase fragmentation predominates (Carrasquillo
et al., 2014; Loza et al., 2014; Hunter et al., 2014).



The presence of organosulfates (OSs) has been demonstrated in several atmospheric
compartments, including atmospheric aerosol (Iinuma et al., 2007; Gómez-González et al.,
2008; Hawkins et al., 2010; Hatch et al., 2011; Kristensen et al., 2011; Stone et al., 2012;
Shalamzari et al., 2013; Hansen et al., 2014; Liao et al., 2015), rain (Altieri et al., 2009),
clouds and fog (Pratt et al., 2013; Boone et al., 2015) and several studies indicate that OSs
could contribute a substantial fraction (up to 30%) of the organic mass measured in ambient
$PM_{2.5}$ (Surratt et al., 2008; Tolocka and Turpin, 2012).
Although the variety of OSs identified from field measurements is quite large (Surratt
et al., 2008; Tao et al., 2014; Kuang et al., 2015; Wang et al., 2015), only a few OS precursors
have been unequivocally identified through laboratory studies. OSs have been generated in
SOA in smog chambers from OH, $NO_3$ or $O_3$ oxidation of BVOCs, including isoprene (Surratt
et al., 2007, Ng et al., 2008), 2-methyl-3-buten-2-ol (MBO) (Zhang et al., 2012; Mael et al.,
2015), unsaturated aldehydes (Schindelka et al., 2013; Shalamzari et al., 2014; Shalamzari et
al., 2015), monoterpenes (Iinuma et al., 2007; Iinuma et al., 2009; Surratt et al., 2008), and
sesquiterpenes (Liggio et al., 2006; Surratt et al., 2008; Iinuma et al., 2009; Noziere et al.,
2010; Chan et al., 2011) in the presence of acidified sulfate aerosol. However, the large
number of unidentified OSs having $C_2$ to $C_{25}$ skeletons observed in recent field studies are
clearly not derived from BVOC precursors, and suggest alkanes and aromatics as a major
source of hitherto unrecognized of OS precursors (Tao et al., 2014; Kuang et al., 2015; Wang
et al., 2015). Ma et al. (2014) have recently shown that the contribution of aromatic OSs could
represent up to two-thirds of the OSs identified in Shanghai. Aliphatic OSs were identified in
the ambient samples from urban locations (Tao et al., 2014; Kuang et al., 2015; Wang et al.,
2015), suggesting that gas-phase oxidation of long-chain or cyclic alkanes could be an
important source of OSs (Tao et al., 2014). At present, lack of authentic standards currently





prevents quantitation of the OSs contribution to $PM_{2.5}$ mass, underscoring the need to better
identify the OS precursors.

Studies on the impact of $NO_x$ and $O_3$ on SOA formation from oxidation of long-chain

alkanes (Loza et al., 2014; Zhang et al., 2014) have shown that the presence of $NO_x$ tends to
reduce SOA formation by reaction of peroxy radicals ($RO_2$) with NO, to yield alkoxy radicals
(RO). For alkanes containing fewer than 10 carbons, the fragmentation/decomposition of RO
radicals will produce higher volatility species (e.g., small carbonyls), which suppresses or
reduce SOA formation (Lim and Ziemann, 2005, 2009). Recent studies have shown that
increased aerosol acidity is a key variable in enhancing SOA formation through acid-
catalyzed reactive uptake and multiphase chemistry of oxidation products derived from
biogenic VOCs (BVOCs) such as isoprene (Surratt et al., 2010) and α-pinene (Iinuma et al.,
2009); however, no such studies have been reported for the oxidation of alkanes. Formation of
highly oxidized products, including OSs, demonstrates the importance of heterogeneous
processes, such as reactive uptake of epoxides onto acidic sulfate aerosol, in SOA formation
(Iinuma et al., 2009; Surratt et al., 2010; Chan et al., 2011; Lin et al., 2014; Shalamzari et al.,
2015). OSs may also be formed by either nucleophilic substitution of organic nitrates by
sulfate (Darer et al., 2011; Hu et al., 2011) or by heterogeneous oxidation of unsaturated
compounds involving sulfate anion radicals (Noziere et al., 2010; Schindelka et al., 2013;
Schone et al., 2014).

Formation of OSs from the gas-phase oxidation of the $C_{10}$ alkanes, cyclodecane

($C_{10}H_{20}$) and decalin (bicyclo[4.4.0]decane; $C_{10}H_{18}$), and $C_{12}$ alkane, dodecane ($C_{12}H_{26}$), in
the presence of sulfate aerosol under varying acidities is reported in this work. Effects of RH,
aerosol acidity and alkane structures on OS formation were investigated. SOA collected from
outdoor smog chamber experiments was chemically characterized by ultra performance liquid
chromatography interfaced to high-resolution quadrupole time-of-flight mass spectrometry





equipped with electrospray ionization (UPLC/ESI-HR-QTOFMS). In addition, effect of
solvent mixture (methanol vs acetonitrile/toluene) on OS quantification was investigated.
Finally, $PM_{2.5}$ samples collected from Lahore, Pakistan and Pasadena, USA were analyzed to
detect and quantify OSs identified from the smog chamber experiments.

**2. Experimental**
**2.1    Chamber Experiments.** Eighteen experiments were performed at the University of
North Carolina (UNC) outdoor smog chamber facility located at Pittsboro, NC. Details of this
facility have been previously described (Lee et al., 2004; Kamens et al., 2011). Briefly, it is a
274−$m^3$ dual chamber divided into two sides by a Teflon film curtain. One side referred as
"North Chamber" has an actual volume of 136 $m^3$, and the other side referred as "South
Chamber" has an actual volume of 138 $m^3$. Prior to each experiment, both sides of the
chamber were flushed using rural background air using an exhaust blower for at least 12
hours. Clean air was then injected into both sides of the chamber using a clean air generator to
reduce concentrations of background aerosol and VOCs. Experiments were performed under
two humidity conditions: at low RH (10−20%) and high RH (4−60%). For experiments
conducted at low RH (i.e., dry), the clean air generator was used after the preliminary venting
using rural air for approximately 48−72 hours. A scanning mobility particle sizer (SMPS, TSI
3080) was used to measure aerosol size distributions, including number and volume
concentrations inside both chambers. Before each experiment, the typical aerosol mass
concentration (assuming an aerosol density of 1 g cm$^{-3}$) background was less than ~ 3 $\mu$g m$^{-3}$
in humid conditions and less than ~ 0.2 $\mu$g m$^{-3}$ for dry experiments. Either non-acidified or
acidified ammonium sulfate seed aerosols were introduced into the chambers by atomizing
aqueous solutions of 0.06 M $(NH_4)_2SO_4$ or 0.06 M $(NH_4)_2SO_4$ + 0.06 M $H_2SO_4$, respectively.
After 15 min of atomization, ~ 40 $\mu$g m$^{-3}$ of seed aerosol was injected into the chambers.





After stabilization of aerosol volume concentrations, Teflon filters were collected (47 mm
diameter, 1.0 $\mu$m pore size, Tisch Environmental, EPA $PM_{2.5}$ membrane) over 45 min at a
sampling rate of ~ 25 L min$^{-1}$ in order to measure baseline aerosol composition prior to
injection of the SOA precursors. None of the aliphatic OSs produced from the oxidation of
studied alkanes were detected in the chamber background. Dodecane (Sigma-Aldrich, 99%),
cyclodecane (TCI, 94%) or decalin (Sigma-Aldrich, 99%, mixture of *cis + trans*) were
introduced into both sides of the chamber by passing a $N_2$ flow through a heated manifold
containing a known amount of liquid compound. Concentrations of alkanes were measured
online in each side every 10 minutes by a gas chromatograph with a flame ionization detector
(GC−FID, Model CP-3800, Varian), calibrated before each experiment with a standard
mixture of hydrocarbons. Isopropyl nitrite (IPN) (Pfaltz & Bauer, 97%) was used as an OH
radical precursor (Raff and Finlayson-Pitts, 2010) and was injected into both sides when VOC
signals were stable as measured by the GC−FID. $O_3$ and $NO_x$ concentrations were monitored
using UV photometric and chemiluminescent analyzers, respectively ($O_3$: Model 49P,
Thermo-Environmental; $NO_x$: Model 8101B, Bendix). Both instruments were calibrated as
described in previous work (Kamens et al., 2011). Dilution rate for each chamber was
monitored by sulfur hexafluoride ($SF_6$) measured using gas chromatography with electron
capture detection (GC−ECD). RH, temperature, irradiance and concentration of $O_3$ and $NO_x$
were recorded every minute. SOA formation from alkane photooxidation was monitored for
all experiments. 2−3 hours following IPN injection, which corresponds to the end of SOA
growth as measured by the SMPS, filter sampling was initiated. For each experiment, two
filters from each side of the chamber were collected for 45 min − 2 hours (sampling rate ~ 25
L min$^{-1}$) to characterize particle-phase reaction products. Based on SOA volume
concentrations measured by the SMPS, sampling time was adjusted to obtain an SOA mass of
about ~ 100 µg/filter. Experimental conditions are summarized in Table 1.





**2.2 Ambient Aerosol Collection.** Five filters collected in Lahore (Pakistan) between January 2007 and January 2008 (Stone et al., 2010) and eight filters collected in Pasadena (USA) during the 2010 California Research at the Nexus of Air Quality and Climate Change (CalNex) field study from 15 May − 15 June 2010 (Hayes et al., 2013), were analyzed for the OSs identified in smog chamber experiments. $PM_{2.5}$ was collected on prebaked quartz fiber filters (QFF, Pall Life Sciences, Tissuquartz, 47 mm for Lahore, 20.3 cm × 25.4 cm for Pasadena) using a medium-volume sampling apparatus at Lahore (URG-3000, Chapel Hill, NC, USA) and a high-volume sampler (Tisch Environmental, Cleves, OH, USA) at Pasadena. As stipulated previously at both urban sites, anthropogenic activities (e.g., vehicular exhaust, industrial sources, cooking, etc.) likely dominated the organic aerosol mass fraction of $PM_{2.5}$ (Stone et al., 2010; Hayes et al., 2013).

**2.3 Filter Extraction.** The impact of the solvent mixture on OS quantification was also explored in this work. Filters collected from smog chamber experiments were extracted using two different solvent mixtures. One filter was extracted using 22 mL of high-purity methanol (LC-MS CHROMASOLV-grade, Sigma-Aldrich, ≥ 99.9 %) under 45 min (25 min + 20 min) of sonication at room temperature while the second filter was extracted using 22 mL of a 70/30 (v/v) solvent mixture containing acetonitrile/toluene (CHROMASOLV-grade, for HPLC, Sigma-Aldrich, ≥ 99.9 %). Extracts were then blown dry under a gentle nitrogen stream at ambient temperature (Surratt et al., 2008; Zhang et al., 2011; Lin et al., 2012). Dry extracts were then reconstituted with 150 $\mu$L of either a 50:50 (v/v) solvent mixture of methanol and water (MilliQ water) or a 50:50 (v/v) solvent mixture of acetonitrile and water. Filters collected from field studies were extracted using methanol as solvent mixture and following the protocol described above; however, prior to drying, extracts were filtered through 0.2-$\mu$m PTFE syringe filters (Pall Life Science, Acrodisc) to remove insoluble particles or quartz filter fibers.





**2.4 Chemical Analysis.** Characterization of OSs in chamber experiments was performed
using ultra performance liquid chromatography interfaced to a high-resolution quadrupole
time-of-light mass spectrometer equipped with an electrospray ionization source (UPLC/ESI-
HR-Q-TOFMS, 6500 Series, Agilent) operated in the negative ion mode. Exact operating
conditions have been previously described (Lin et al., 2012). 5 $\mu$L sample aliquots were
injected onto a UPLC column (Waters ACQUITY UPLC HSS T3 column). Octyl sulfate
($C_8H_{17}O_4S^-$; Sigma-Aldrich) and 3-pinanol-2-hydrogen sulfate ($C_9H_{13}O_6S^-$) were used as
surrogate standards to quantify the identified aliphatic OSs.

**3.   Results and Discussion**
In the subsequent sections, detailed chemical characterization of OSs identified from the gas-
phase oxidation of dodecane, decalin and cyclodecane in the presence of ammonium sulfate
aerosol is presented. The presence of OSs was revealed by the appearance of characteristic
fragment ions at $m/z$ 79.95 ($SO_3^{\bullet/-}$), 80.96 ($HSO_3^-$) and/or 96.96 ($HSO_4^-$) in tandem mass
spectra (MS[2]) (Iinuma et al., 2007; Gómez-González et al., 2008; Surratt et al., 2008;
Shalamzari et al., 2013; 2014). Tentative structures, retention times and exact mass
measurements of OSs detected in this work are reported in Table S1. The low abundance of
some OSs precluded acquisition of high-resolution MS[2] data, and thus, structures have not
been proposed for these less-abundant parent ions.
**3.1 Characterization of OSs from Dodecane Photooxidation.** Seven OSs, including
isobaric compounds, were identified in SOA produced from the gas-phase oxidation of
dodecane in the presence of sulfate seed aerosol. None have previously been reported in
chamber experiments, although they have recently been observed in ambient fine aerosol
samples (Tao et al., 2014; Kuang et al., 2015). Concentrations of the products are reported in
Table S2. Three isobaric parent ions with $m/z$ 279 ($C_{12}H_{23}O_5S^-$, 279.1254) were identified in
SOA generated from dodecane oxidation in the presence of acidified ammonium sulfate





aerosol. Based on Yee et al. (2012; 2013) the products are tentatively assigned as 1,3-
dodecanone sulfate isomers. The $MS^2$ spectra of the products were identical, having product
ions diagnostic for a sulfate ester β to an abstractable proton (Surratt et al., 2008; Gómez-
González et al., 2008) at $m/z$ 199 ($C_{12}H_{23}O_2^-$, loss of neutral $SO_3$) and 97 ($HSO_4^-$), precluding
assignment of positional isomerism. A study on the OH oxidation of the linear alkane
octacosane indicates a strong preference for oxidation at terminal carbons, with the $C_2$
carbonyl predominating (Ruehl et al., 2013), suggesting the structure 2-dodecanone-4-sulfate
for OS-279 and possibly 3-dodecanone-5-sulfate and 4-dodecanone-6-sulfate isomers as
isobaric ions. Figures 1 and S1 present the $MS^2$ spectrum of OS-279 and proposed
fragmentation pathway, respectively. By chemical ionization mass spectrometry (CIMS)
operating in the negative mode, Yee et al. (2012) identified the formation of hydroperoxides
from the oxidation of dodecane under low−$NO_x$ conditions, confirming the predicted $RO_2$ −
$HO_2$ reaction pathway in the low−$NO_x$ regime. First-generation hydroperoxides can react with
OH by further oxidation to form low−volatility, more highly oxidized products or by
fragmentation/decomposition of alkoxy (RO) radicals to form products with higher volatility
(Yee et al., 2012, Carrasquillo et al., 2014). In our study, OH radicals were formed from IPN
photolysis without additional injection of NO. Under these conditions, $RO_2$ chemistry is
dominated by $RO_2$ + $HO_2$ and/or $RO_2$ + $RO_2$ reactions as discussed by Raff and Finlayson-
Pitts (2010). Although $RO_2$ radicals formed in the first oxidation step could also react with
NO formed by IPN photolysis, significant formation of ozone under chamber conditions (0.3-
0.6 ppm, depending on concentration of IPN injected) would rapidly quench NO. Thus $RO_2$ +
NO reactions are not anticipated to be significant (Raff and Finlayson-Pitts, 2010). Carbonyl
hydroperoxide (CARBROOH), which has been identified in the gas phase by Yee et al.
(2012), is likely involved in acid-catalyzed reactive uptake onto sulfate aerosol.
Heterogeneous chemistry of gas-phase organic peroxides has been previously suggested to



explain the formation of certain OSs and tetrols (Claeys et al., 2004; Riva et al., 2015b). Acid-
catalyzed perhydrolysis of hydroperoxides followed by reaction with sulfate anion radicals
could also be possible route to the formation of OS-279 (Figure 1). However, further
investigation is required to better understand how acidified sulfate seed aerosol takes up
organic peroxides from the gas phase and how particle-phase reactions might degrade organic
peroxides into OSs.
**3.2 Characterization of OSs from Decalin Photooxidation.** Gas-phase oxidation of cyclic
alkanes at room temperature and atmospheric pressure has received less attention than linear
or branched alkanes. However, recent studies have demonstrated that oxidations of cyclic
alkanes by OH radicals produce less–volatile oxygenated compounds and have larger SOA
yields (Yee et al., 2013; Hunter et al., 2014). Significant formation of OSs (up to 1 μg m$^{-3}$)
and SOA were observed in all experiments of decalin photooxidation (Tables 1 and S3),
revealing the high potential for bicyclic alkanes to form OSs. All OSs (25 OSs including
isomeric/isobaric structures) identified from the oxidation of decalin in the presence of
ammonium sulfate aerosol have been observed in ambient aerosol, underscoring the potential
importance of alkanes to OS formation in urban areas (Tao et al., 2014; Kuang et al., 2015;
Wang et al., 2015). MS$^2$ spectra were obtained for all OSs identified from decalin oxidation,
except for parent ions at *m/z* 195.0697 (OS-195) and 299.0805 (OS-299). All of the parent
ions show an intense product ion at *m/z* 96.96, indicative of an aliphatic sulfate ester.
Retention times and tentative structural assignments are given in Table S1.
Figures 2 and S2 present MS$^2$ spectra and fragmentation schemes of selected parent
ions at *m/z* 265.0749 (OS-265), 269.0696 (OS-269), 295.0494 (OS-295) and 326.0554 (OS-
326). MS$^2$ spectra and fragmentation schemes of other OSs are reported in Figure S3-S7.
These selected OSs exhibit specific fragmentation patterns and were, as described in the next
section, quantified and characterized in the fine urban aerosol samples. Four isomers of OS-



272 265 with composition $C_{10}H_{17}O_6S^-$ were identified in decalin-derived SOA collected from all

273 experiments. With regard to components of ambient SOA, it is important to mention that the

274 formation of isobaric OSs with the same elemental composition of $C_{10}H_{17}O_6S^-$ isobars have

275 also been previously identified in SOA produced from the gas-phase oxidation of

276 monoterpenes (Liggio et al., 2006; Surratt et al., 2008) and are not unique to decalin

277 oxidation. The product ion at nominal $m/z$ 97 ($HSO_4^-$) and loss of neutral $SO_3$ in the $MS^2$

278 spectrum (Figure 2a) is consistent with an aliphatic OS having a labile proton in a β position

279 (Attygalle et al., 2001). Absence of product ions corresponding to a loss of a terminal

280 carbonyl (−CO) or a carboxyl group (−$CO_2$), respectively (Romero and Oehme, 2005;

281 Shalamzari et al., 2014), and a composition corresponding to 2 double bond equivalencies

282 (DBEs) has thus been attributed to an internal carbonyl group and a six-membered ring. A

283 scheme leading to the structure proposed in Figure 2a is based on the abstraction of H at the

284 ring fusion and $C_1$–$C_2$ bond cleavage (Figure S8, pathway **a**) and subsequent reaction with $O_2$

285 followed by 1,5-H shifts leading to an epoxide and sulfate ester by reactive

286 uptake/heterogeneous chemistry.

287  The composition of the parent ion at $m/z$ 269.0696 ($C_9H_{17}O_7S^-$) corresponds to one

288 DBE. $MS^2$ spectrum yields products consistent with a sulfate ester β to an abstractable proton

289 and similar to OS-265, neither a terminal carbonyl nor a carboxyl functional group was

290 detected in the OS-269. As a result, the presence of hydroperoxide and/or hydroxyl

291 substituents is expected in order to help explain this molecular formula obtained from the

292 accurate mass measurement. In Figure 3, tentative pathways leading to the formation of OS-

293 267, OS-269 and OS-285 are proposed. Under low-$NO_x$ conditions abstraction of a proton α

294 to the ring fusion of decalin followed by reaction with $O_2$ leads to the 1-hydroperoxy radical

295 (Yee et al., 2013; Schilling Fahnestock et al., 2015), which in turn can react with another $RO_2$

296 radical to yield the corresponding alkoxyl radical ($C_{10}H_{17}O^{\bullet}$) (Atkinson, 2000). Cleavage of





the $C_1$–$C_2$ decalin bond, followed by reaction with a second $O_2$ molecule and $HO_2$ leads to a
terminal carbonyl hydroperoxide (tCARBROOH; $C_{10}H_{18}O_3$). As reported in recent studies,
the resulting $RO_2$ radical can undergo isomerization/auto-oxidation to yield a hydroperoxide
(Ehn et al., 2014; Jokinen et al., 2014; Mentel et al., 2015), which can lead to the formation of
OSs through reactive uptake (Mutzel et al., 2015). By analogy to other aldehydes, OH
preferentially abstracts the tCARBROOH aldehydic H (79:21 branching ratio; Kwok and
Atkinson 1995). The $RO_2$ radical produced from H–atom could react with $RO_2$ or $HO_2$ and
form an acyl-oxy radical (R(O)O$^\bullet$), which decarboxylates quickly (Chacon-Madrid et al.,
2013) to a hydroperoxyperoxy radical ($C_9H_{17}O_4^\bullet$). The hydroperoxyperoxy radical can react
via pathway **a** (Figure 3) leading to OS-267, previously unreported, or OS-269 or pathway **b**
(Figure 3) leading to OS-285. Pathway **a** proceeds via a 1,7-H shift followed by elimination
of OH from the resulting dihydroperoxy alkyl radical to give a dihydroperoxyepoxide by a
1,5-H shift and OH elimination analogous to the formation of isoprene epoxydiol (IEPOX)
(Paulot et al., 2009; Mael et al., 2015). The epoxide can then undergo acid-catalyzed ring
opening to give OS-269, which may be further oxidized to OS-267. The MS$^2$ spectrum of OS-
285 (Figure S5) shows product ions corresponding to $HSO_3^-$, $HSO_4^-$ and loss of neutral $SO_3$,
in accord with a sulfate ester β to a labile proton, but yields no further structural information.
The structure proposed for OS-285 is based on the reaction of the hydroperoxyperoxyl radical
intermediate in pathway **a** with $RO_2$ followed by a 1,6-H shift and addition of $O_2$ to give a
hydroxyhydroperoxyperoxyl radical ($C_9H_{17}O_5^\bullet$) leading to an epoxide by a 1,5-H shift and
OH elimination as described above (Iinuma et al., 2009; Surratt et al., 2010; Jacobs et al.,
2013; Mael et al., 2015).

In Figure 4, pathways from an initial 1-peroxy transient are proposed to products

designated OS-295, OS-311 and OS-326. Three isobaric ions corresponding to OS-295
($C_{10}H_{15}O_8S^-$) were identified in decalin-derived SOA under all experimental conditions.



Figure 2c shows the MS$^2$ spectrum of the parent ion at *m/z* 295. A product ion at *m/z* 251
corresponding to loss of $CO_2$ (Romero and Oehme, 2005; Shalamzari et al., 2014) is present
in addition to product ions consistent with a sulfate ester β to a labile H (Riva et al., 2015b).
Pathway **a** leads to the structure consistent with the MS$^2$ spectrum and 3 DBEs required by
the composition of the parent ion. The salient features of pathway **a** include oxidation of the
RO$_2$ to 2-decalinone, formation of a C$_{10}$ alkoxy radical followed by ring cleavage of the
C$_9$−C$_{10}$ decalin bond leading to a 4-(carboxy cyclohexyl)-1-hydroperoxybut-2-yl radical via
RO$_2$ chemistry, and a 1,7-H shift and acid-catalyzed ring opening of the epoxide resulting
from the addition/isomerization of the O$_2$ adduct (Paulot et al., 2009).

Two isobaric parent ions with identical MS$^2$ spectra were observed at *m/z* 311 (Figure

S7). The only observed product ion at *m/z* 97 is consistent with a sulfate ester, but not
informative with regard to a more refined assignment of molecular structure. Pathway **b** to a
hydroperoxide for the parent ion with 3 DBEs is proposed by analogy to the putative
hydroperoxide structures of OS-267, OS-269 and OS-285. Pathway **b** is characterized by a H-
abstraction from a carbon at the ring fusion of 2-decalinone leading to formation of an 2-
decalinone-6-oxyl radical followed by a sequence of ring cleavage, O$_2$ additions and H-shifts
to form a 4-(2,6-cyclohexyl)-2-hydroperoxybutan-1-oxide that can form the sulfate ester on
reactive uptake. Abstraction of H$_1$ rather than H$_6$ would lead to an isobaric structure.

Four isobaric ions corresponding to C$_{10}$H$_{16}$NO$_9$S$^-$ with identical MS$^2$ spectra (Figure

1d) were detected at nominal mass *m/z* 326. The loss of 63 mass units as neutral HNO$_3$
(Figure S2d) is in accord with a nitrate ester (Surratt et al., 2008), supported by the absence of
product ions from loss of NO or NO$_2$ (Kitanovski et al., 2012). Pathway **c**, to the parent ion at
*m/z* 326 proceeds from the reaction of the decalin-2-peroxy radical with NO to form decalin-
2-nitrate (C$_{10}$H$_{17}$NO$_3$) (Atkinson, 2000). From this point, a sequence of reactions identical to
pathway **b** yields the parent OS-326. It is important to mention that the formation of isobaric



OSs with the same elemental composition of $C_{10}H_{16}NO_9S^-$ isobars have also been identified
in SOA produced from the gas-phase oxidation of monoterpenes (Surratt et al., 2008).

The $MS^2$ spectrum for the single parent ion at $m/z$ 281 corresponding to the

composition $C_{10}H_{17}O_7^-$ (OS-281) gave product ions expected for a sulfate ester β to a labile
proton with 2 DBE, but no additional structural information (Figure S4). The pathway
proposed in Figure S8 pathway **b** is based on gas-phase oxidation of to a 4-(cyclohexan-2-
one)but-1-yl radical followed by reaction with $O_2$ and a 1,6-H shift followed by addition of a
second $O_2$, a 1,5-H shift and elimination of OH to give an epoxide. The direction of ring
opening of the internal epoxide by reactive uptake to give the final product is arbitrary. Three
isobaric parent ions at $m/z$ 297 corresponding to the composition $C_{10}H_{17}O_8S^-$ with 2 DBEs
were identified. Loss of water, $HSO_4^-$ and $SO_3$ as a neutral fragment in the $MS^2$ spectrum of
the major isobar (OS-297) is consistent with a hydroxyl-substituted sulfate ester β to a labile
proton (Figure S6). The scheme proposed in Figure S8 pathway **c** is based on the oxidation to
a 4-(cyclohexan-2-one)but-1-yl radical as in pathway **b**. However, in contrast to pathway **b**,
$RO_2$ formed by the addition of $O_2$ undergoes a 1,6-H shift followed by addition of a second
$O_2$ molecule, a 1,5-H shift and elimination of OH to yield an epoxide, which be reactively
taken up to give a sulfate ester. The direction of ring opening of the internal alkyl epoxide is
arbitrary.
**3.3 Characterization of OSs from Cyclodecane Photooxidation.** The concentrations of
OSs identified from gas-phase oxidation of cyclodecane are reported in Table S4. High levels
of OSs were observed in experiments performed under dry conditions with acidified
ammonium sulfate seed aerosol. The impact of acidity on OS formation will be discussed in
more detail in the following section. The $MS^2$ spectra of all cyclodecane products show only a
single product ion at nominal $m/z$ 97 corresponding to bisulfate (Figures S9 – S13), indicating
that the oxidation products are sulfate esters β to a labile proton. None of the fragment ions



observed in the MS$^2$ spectrum suggests neither a terminal carbonyl nor a carboxyl functional
group are present in the cyclodecane-OSs, consistent with retention of the cyclodecane ring
the oxidation products. Tentative structures proposed in Table S1 are based on DBE
calculations and retention of the cyclodecane ring supported by MS$^2$ data. Pathways proposed
in Figures S14 and S15 are initiated by H-abstraction and based on reaction sequences for
which precedent has been established: addition of O$_2$ to cycloalkyl radicals to give RO$_2$ which
either react with RO$_2$ to yield alkoxy radicals (Atkinson, 2000; Yee et al., 2012) or undergo
intramolecular H-shifts leading to generation of hydroperoxydes (Ehn et al., 2014; Jokinen et
al., 2014). The formation of compounds such as cyclodecanone (C$_{10}$H$_{18}$O) or cyclodecane
hydroperoxide (C$_{10}$H$_{20}$O$_2$) are proposed as intermediate products leading to epoxy-
compounds after additional oxidation/isomerization processes, as presented in Figures S14
and S15. Since authentic standards are unavailable and the MS$^2$ data do not allow specific
structural features to be assigned, the end products in pathways in Figures S14 and S15 are
arbitrary. Isobars may be explained by *cis/trans* epoxide ring opening or the span of an H-
shift (1,5-,1,6- and 1,7-H shifts are possible). In the case of OS-249, where *cis/trans* isomers
are not possible; the two isobaric structures may result from different H-shifts. OS-265 and
OS-281 are reported here for the first time in chamber studies.
**3.4 Effect of Alkane Structure on Relative SOA Yield.** Alkane structures appear to be
important determinants of the relative yields of OSs from dodecane, decalin and cyclodecane
photooxidation. Tables S2-S4 show that OS concentrations are significantly higher from the
photooxidation of decalin and cyclodecane than from dodecane. As reported in Table 1, SOA
formation from gas-phase oxidation of decalin and cyclodecane was much higher than during
photooxidation of dodecane, which could explain the larger amount of OSs identified.
Although the SOA formed from photooxidation of both cyclic alkanes was comparable, the



sum of OSs quantified from oxidation of decalin was 3-4 times higher. An investigation of the
reason for these differences is ongoing.
**3.5 Impact of Relative Humidity and Acidity on OS Formation.** Experiments were
performed under conditions reported in Table 1. As shown in Figure 5 and Tables S2-S4, the
presence of acidic aerosols significantly increases OS formation in most cases, as previously
observed for OSs in SOA generated from biogenic sources (Iinuma et al., 2007; Surratt et al.,
2007; Chan et al., 2011). Since differences in meteorology could impact experimental results
from the outdoor chamber, caution must be exercised in comparing experiments performed on
different days. However, same-day, side-by-side experiments allow for clear resolution of the
effects of aerosol acidity and seed composition on OS formation. When comparing
experiments performed under dry versus wet conditions with acidified ammonium sulfate
aerosol, higher RH conditions significantly reduce OS formation, likely attributable to an
increase in pH because of dilution by additional particle water. To better investigate the effect
of acidity on OS formation, products were divided in two groups (Figure 5), those whose
concentrations were increased by a factor $\geq 2$ (Group-1) and $\leq 2$ (Group-2). Figure 5 and
Tables S2-S4 show that OSs identified from dodecane photooxidation belong to Group-2,
with the exception of OS-279. OSs from decalin photooxidation, including OS-195, OS-269
and OS-297 belong to Group-2 as well. OSs can be formed via different pathways, including
acid-catalyzed ring-opening reactions of epoxy-containing SOA constituents, reactive uptake
of unsaturated compounds into the particle phase, or by reaction with the sulfate anion radical
(Rudzinski et al., 2009; Nozière et al., 2010; Schindelka et al., 2013; Schöne et al., 2014).
OSs may also result from nucleophilic substitution of nitrate by sulfate (Darer et al., 2011; Hu
et al., 2011). The impact of acidity on OS formation arising from the different pathways has
been investigated principally for reactive uptake of epoxy-compounds (Jacobs et al., 2013;
Lin et al., 2012; Gaston et al., 2014; Riedel et al., 2015) for which OS formation is strongly



enhanced under acidic conditions (Lin et al. (2012). However, a similar enhancement was not
observed in our study on PAH-OSs, which were not expected to result from epoxide
chemistry (Riva et al., 2015a). Based on these observations, the formation of Group-1 OSs are
hypothesized to be products of reactive uptake of gas-phase epoxides.
**3.6 Impact of Solvent Mixture on OS Quantification**. Additional filters were collected from
each side of the outdoor chamber and for each experiment to investigate the impact of solvent
mixture on OS quantification. Tao et al. (2014) have recently reported that less polar solvents
such as an acetonitrile (ACN)/toluene mixture are a better choice for extraction of long alkyl-
chain OSs from filters using a nanospray-desorption electrospray ionization mass
spectrometry where the extraction occurs *in situ* and the analyses are qualitative. Figure 6
demonstrates that, overall, concentrations of OSs (ng m$^{-3}$) from the photooxidation of
dodecane, decalin and cyclodecane seem to be more efficiently extracted by the ACN/toluene
mixture. Tables S2-S4, showing the ratios of the concentrations individual OSs extracted by
the ACN/toluene mixture divided by the concentration of OSs extracted by methanol,
indicates that all $C_{10}-$ and $C_{12}-$ OS products, including highly oxidized OS, appear more
efficiently extracted by the ACN/toluene mixture. For OSs smaller than $C_{10}$, extraction
efficiencies are about the same. As noted above, isobars of OSs identified from the oxidation
of alkanes have been observed in SOA generated from the oxidation of monoterpenes that are
currently used as tracers for monoterpene SOA chemistry (Hansen et al., 2014; Ma et al.,
2014). Hence, in addition to the caution that quantitation of alkane and monoterpene OSs is
uncertain in the absence of authentic standards, some monoterpene OSs may be
underestimated if not fully extracted because most studies use methanol as an extraction
solvent (Surratt et al., 2008; Iinuma et al., 2009). More work is, however, needed to better
characterize and elucidate the impact of solvent on the quantitation of biogenic and
anthropogenic OSs, especially compounds $> C_{10}$.





**3.7 OSs Derived from Long-Chain Alkanes in Ambient Fine Urban Aerosol.** Archived
fine urban aerosol samples collected at Lahore, Pakistan, and Pasadena, USA were used to
evaluate and quantify OSs identified in SOA produced from the photooxidation of long-chain
alkanes. Filters were initially extracted using methanol and comparison to OSs quantified
using another solvent mixture was not possible. As previously mentioned, seven parent ions
have been observed in laboratory studies. Therefore, extracted ion chromatograms (EICs)
obtained from smog chamber experiments were compared to those obtained from both urban
locations to confirm that observed OSs correspond to OSs identified in our lab study. Figures
7 and S16 present the EICs of OSs observed in both ambient and our smog chamber-
generated SOA. Table 2 identifies 12 OSs, along with concentrations, present in $PM_{2.5}$
collected from Lahore, Pakistan and Pasadena, USA and also observed in our smog-chamber-
generated SOA.
The high concentrations, especially at Lahore (Pakistan) of the OSs measured in the
ambient aerosol samples support their use as tracers for SOA produced from the oxidation
long-chain alkanes in urban areas. This is consistent with recent proposals (Tao et al., 2014).
OS-195 ($C_7H_{15}O_4S^-$), OS-249 ($C_{10}H_{17}O_5S^-$), OS-255 ($C_9H_{19}O_6S^-$), OS-267 ($C_{10}H_{19}O_6S^-$),
OS-281 ($C_{10}H_{17}O_7S^-$), OS-299 ($C_{10}H_{19}O_8S^-$), OS-307 ($C_{12}H_{19}O_7S^-$) and OS-311
($C_{10}H_{15}O_9S^-$) have been recently identified in ambient aerosol collected from the major urban
locations Shanghai and Hong Kong (Tao et al., 2014; Kuang et al., 2015; Wang et al., 2015).
In the absence of retention times and chromatographic conditions, OS isobars such as OS-249
or OS-279, which are currently assigned to biogenic-derived OSs (Ma et al., 2014), could also
arise from anthropogenic sources such as photooxidation of cyclodecane, especially in urban
areas.








## 4. Conclusions

The present study demonstrates the formation of OSs from the photooxidation of alkanes and
complements the smog chamber study on formation of OSs and sulfonates from
photooxidation of PAHs (Riva et al., 2015a). Together, the results strongly support the
importance of the contribution of anthropogenic precursors to OS in ambient urban $PM_{2.5}$
proposed on the basis of aromatic and aliphatic OSs in fine aerosol collected from several
major urban locations (Kundu et al., 2013, Tao et al., 2014). Chemical characterization of OSs
that were identified in SOA arising from the photooxidation of alkanes were performed and
tentative structures have been proposed for OSs identified from the photooxidation of decalin,
cyclodecane and docecane based on composition from exact mass measurement, DBE
calculations and the transformations expected from hydroxyl radical oxidation dominated by
$RO_2/HO_2$ chemistry. Enhancement of OS yields in the presence of acidified ammonium
sulfate seed is consistent with reactive uptake of gas-phase epoxides as the pathway for OS
formation. As previously proposed for IEPOX formation (Paulot et al. 2009), isomerization of
$RO_2$ species to β hydroperoxy alky radicals followed by elimination of OH, is a plausible
pathway to gas-phase epoxides. However, more work is required to validate pathway(s)
leading to the formation of gaseous epoxy-products. Of critical importance would be
investigations starting from authentic primary or secondary oxidation products suggested in
this study as putative intermediates to validate the proposed mechanisms. A novel pathway
involving reactive uptake of hydroperoxides followed by hydrolysis/sulfation reactions is
proposed to explain the formation of OS-279 ($C_{12}H_{23}O_5S^-$); however, more work is also
required to examine how acidified sulfate seed aerosols take up organic peroxides from the
gas phase and how particle-phase reactions might degrade organic peroxides into low-
volatility products such as the OSs.



**Acknowledgments**
The authors thank the Camille and Henry Dreyfus Postdoctoral Fellowship Program in
Environmental Chemistry for their financial support. The authors wish also to thank CAPES
Foundation, Ministry of Education of Brazil (award no. 99999.000542/2015-06) for their
financial support. This study was supported in part by the National Oceanic and Atmospheric
Administration (NOAA) Climate Program Office's AC4 program, award no.
NA13OAR4310064. The authors wish to thank Kasper Kristensen and Marianne Glasius
(Department of Chemistry, Aarhus University, Denmark) who synthesized the 3-pinanol-2-
hydrogen sulfate. The authors also thank Tauseef Quraishi, Abid Mahmood, and James
Shauer for providing filters collected in Lahore, in addition to the Government of Pakistan,
the Pakistani Higher Education Commission, and the United States Agency for International
Development (US-AID) for funding field sample collection in Pakistan.




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



**Table 1.** Summary of outdoor smog chamber conditions used for the photooxidation of long-
chain alkanes using isopropyl nitrite (IPN) as an OH radical precursor.

| Hydrocarbons (HCs) | Initial [HC] (ppb) | Chamber Side | Seed aerosol | Initial [IPN] (ppb) | T (K) | RH (%) | Final OA mass (µg m$^{-3}$) |
|---|---|---|---|---|---|---|---|
| Dodecane | 412 | N | Non-Acidified | 215 | 304-311 | 49-59 | 58 |
|  | 420 | S | Acidified | 212 | 305-311 | 51-63 | 65 |
| Dodecane | 422 | N | Non-Acidified | 215 | 302-308 | 15-20 | 49 |
|  | 427 | S | Acidified | 212 | 303-308 | 14-17 | 53 |
| Dodecane | 397 | N | Acidified | 215 | 304-309 | 45-52 | 52 |
|  | 409 | S | Acidified | 212 | 305-310 | 15-19 | 59 |
| Decalin | 175 | N | Non-Acidified | 138 | 302-309 | 48-45 | 204 |
|  | 180 | S | Acidified | 136 | 302-308 | 51-49 | 224 |
| Decalin | 199 | N | Non-Acidified | 138 | 305-306 | 13-13 | 200 |
|  | 204 | S | Acidified | 136 | 306-306 | 13-14 | 211 |
| Decalin | N.I. | N | Acidified | 138 | 302-306 | 43-54 | 245 |
|  | N.I. | S | Acidified | 136 | 301-306 | 9-12 | 270 |
| Cyclodecane | 257 | N | Non-Acidified | 172 | 298-301 | 53-61 | 218 |
|  | 263 | S | Acidified | 170 | 299-301 | 52-60 | 238 |
| Cyclodecane | 256 | N | Non-Acidified | 172 | 300-303 | 13-15 | 177 |
|  | 261 | S | Acidified | 170 | 300-302 | 13-14 | 210 |
| Cyclodecane | 245 | N | Acidified | 172 | 298-300 | 10-11 | 259 |
|  | 250 | S | Acidified | 170 | 299-300 | 51-49 | 270 |

*N and S design "North chamber" and "South Chamber", respectively; N.I.: No Information.*





**Table 2.** Concentrations (ng m$^{-3}$) of OSs identified in laboratory-generated dodecane, decalin and cyclodecane SOA and in fine aerosol collected from two urban locations.

| [M – H]$^-$ | Precursors | Lahore, Pakistan | | | | | Pasadena, USA | | | | | | | |
|---|---|---|---|---|---|---|---|---|---|---|---|---|---|---|
| | | 04-30-2007 | 05-06-2007 | 05-12-2007 | 11-02-2007 | 11-08-2007 | 05-17-2010 | 05-18-2010 | 05-19-2010 | 05-23-2010 | 05-24-2010 | 05-25-2010 | 05-28-2010 | 06-11-2010 |
| $C_7H_{13}O_5S^-$ (209.0472)[a,b] | Dodecane | 7.53 | 6.53 | 4.24 | 6.35 | 9.66 | *N.d.* | *N.d.* | 0.27 | 0.07 | 0.10 | *N.d.* | 0.09 | 0.21 |
| $C_9H_{17}O_5S^-$ (237.0786)[a,b] | Dodecane | 9.35 | 6.81 | 4.27 | 7.27 | 12.40 | 0.13 | 0.15 | 0.30 | 0.10 | 0.16 | 0.16 | 0.13 | 0.25 |
| $C_{10}H_{19}O_5S^-$ (251.0946)[a,c] | Cyclodecane | 10.40 | 7.51 | 4.08 | 13.17 | 20.96 | *N.d.* | *N.d.* | *N.d.* | *N.d.* | *N.d.* | *N.d.* | *N.d.* | *N.d.* |
| $C_{10}H_{17}O_6S^-$ (265.079)[a,c] | Cyclodecane | 2.83 | 2.45 | 2.15 | 2.86 | 7.63 | 0.18 | 0.21 | 0.35 | 0.14 | 0.15 | 0.16 | 0.15 | 0.36 |
| $C_9H_{15}O_7S^-$ (267.0554)[a,c] | Decalin | 0.98 | 1.87 | 1.93 | 2.19 | 6.53 | 0.21 | 0.21 | 0.58 | 0.11 | 0.21 | 0.20 | 0.16 | 0.40 |
| $C_9H_{17}O_7S^-$ (269.0700)[a,b] | Decalin | 2.04 | 3.02 | 2.22 | 2.62 | 7.56 | 0.42 | 0.38 | 0.58 | 0.26 | 0.40 | 0.38 | 0.35 | 0.56 |
| $C_{10}H_{15}O_7S^-$ (279.0556)[a,c] | Cyclodecane | 6.38 | 20.25 | 21.97 | 15.06 | 35.93 | 0.14 | 0.21 | 0.54 | 0.10 | 0.19 | 0.21 | 0.20 | 0.29 |
| $C_{12}H_{23}O_5S^-$ (279.1272)[c,d] | Dodecane | 14.57 | 12.18 | 3.41 | 9.50 | 19.56 | *N.d.* | *N.d.* | *N.d.* | *N.d.* | *N.d.* | *N.d.* | *N.d.* | *N.d.* |
| $C_9H_{17}O_8S^-$ (285.0651)[a,c] | Decalin | *N.d.* | 0.61 | *N.d.* | *N.d.* | 1.44 | 0.20 | 0.09 | 0.21 | 0.05 | 0.08 | 0.09 | 0.03 | 0.17 |
| $C_{10}H_{15}O_8S$- (295.0500)[a,c] | Decalin | *N.d.* | 0.53 | 0.48 | 0.54 | 3.78 | 0.17 | 0.22 | 0.65 | 0.08 | 0.17 | 0.24 | 0.19 | 0.52 |
| $C_{10}H_{17}O_8S^-$ (297.0650)[a,c] | Decalin | *N.d.* | 0.78 | 0.92 | 0.69 | *N.d.* | 0.13 | 0.08 | 0.43 | 0.07 | 0.10 | 0.09 | 0.10 | 0.24 |
| $C_{10}H_{16}NO_9S^-$ (326.0550)[a,c] | Decalin | 0.25 | 0.32 | 0.21 | *N.d.* | *N.d.* | *N.d.* | 0.13 | 0.22 | 0.06 | 0.09 | 0.11 | 0.12 | 0.11 |

[a] *Quantified using authentic OS (3-pinanol-2-hydrogen sulfate, $C_9H_{13}O_6S^-$ ),* [b] *OSs belonging to group 2,* [c] *OSs belonging to group 1,* [d] *quantified using octyl sulfate OS ($C_8H_{17}O_4S^-$ ). Different isomers for one ion have been summed; N.d.: not detected.*



**Figure 1.** Proposed formation pathway of OS-279 and its corresponding fragmentation routes. The suggested mechanism is based on identified products from previous study (Yee et al., 2012).

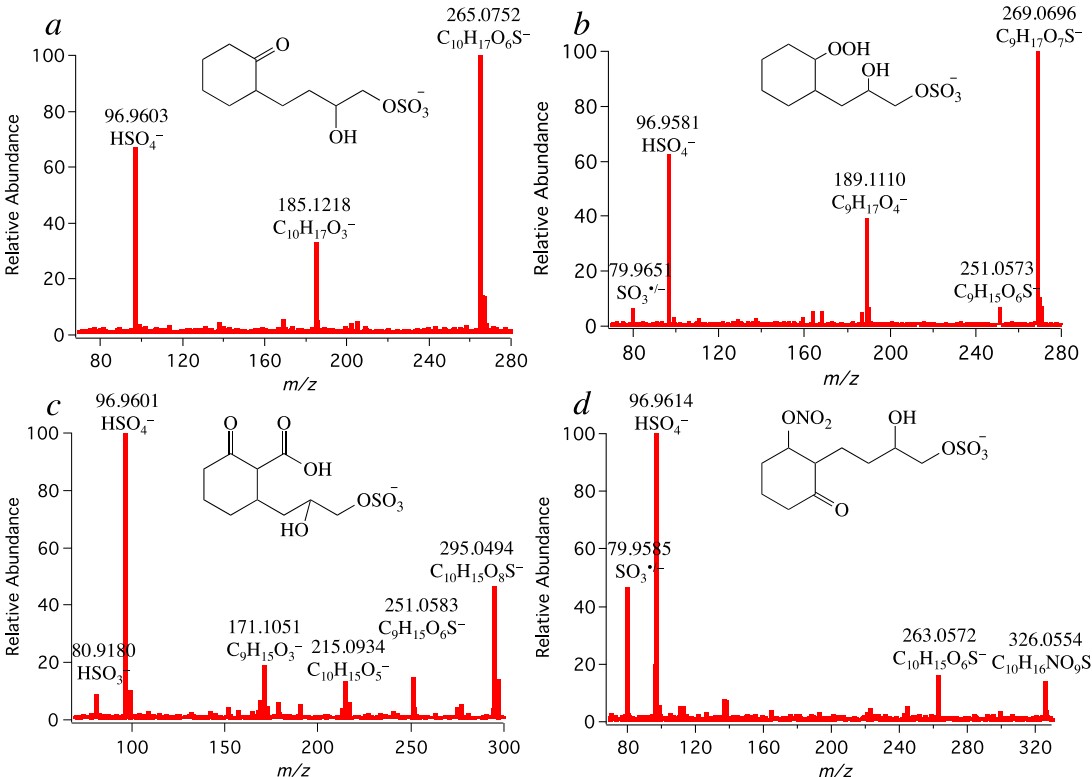

**Figure 2.** MS$^2$ spectra obtained for selected decalin-derived OSs: (*a*) *m/z* 265.0752 (C$_{10}$H$_{17}$O$_6$S$^-$), (*b*) *m/z* 269.0696 (C$_9$H$_{17}$O$_7$S$^-$), (*c*) *m/z* 295.0494 (C$_{10}$H$_{15}$O$_8$S$^-$) and (*d*) *m/z* 326.0554 (C$_{10}$H$_{16}$NO$_9$S$^-$). Fragmentation schemes are proposed in Figure S2.




**Figure 3.** Proposed formation pathways of OS-267, OS-269 and OS-285 from the oxidation of decalin in presence of sulfate aerosol. ISO = isomerization reaction either through H shift (1,5- or 1,7-) or through hyderoperoxide isomerization with an R radical.



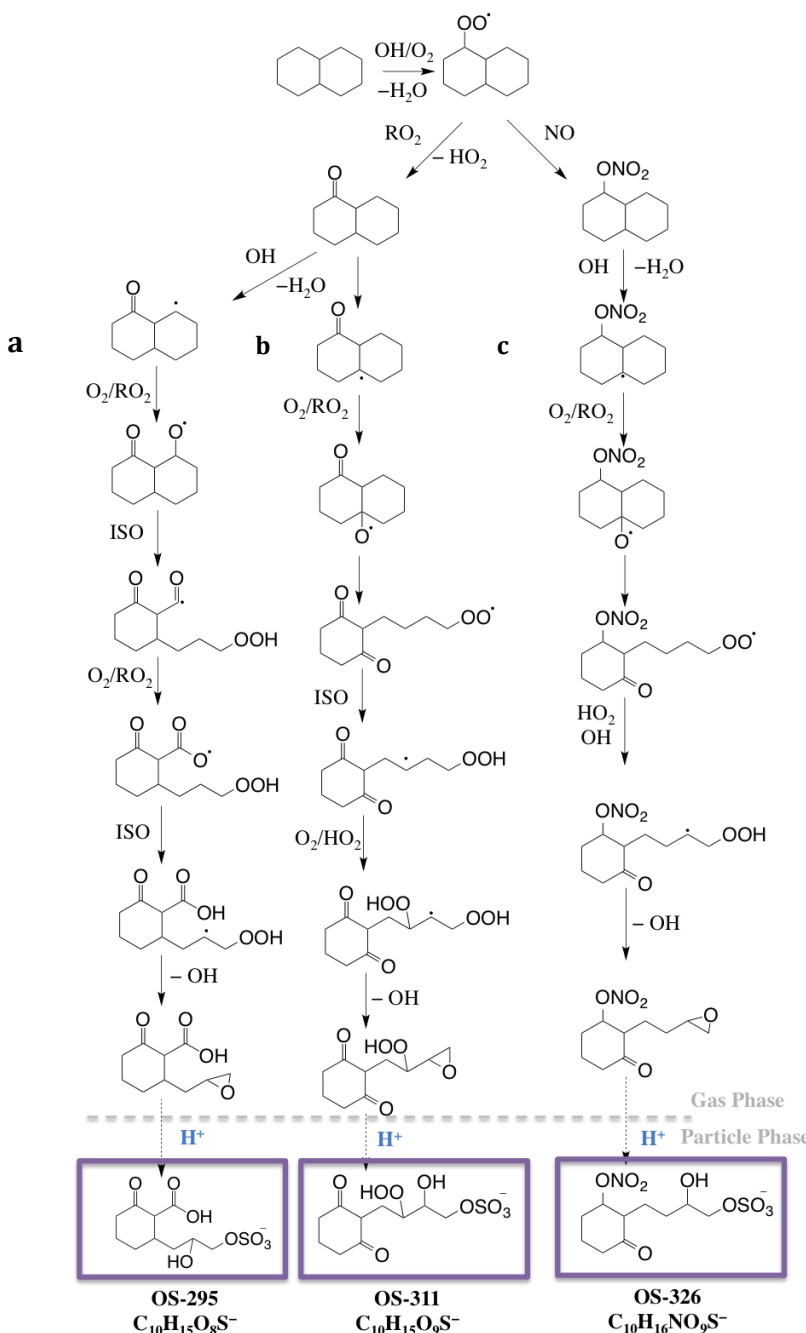

**Figure 4.** Proposed formation pathways of OS-295, OS-311 and OS-326 from the oxidation of decalin in the presence of sulfate aerosol. ISO = isomerization reaction either through H shift (1,5- or 1,7-) or through hyderoperoxide isomerization with an R radical.




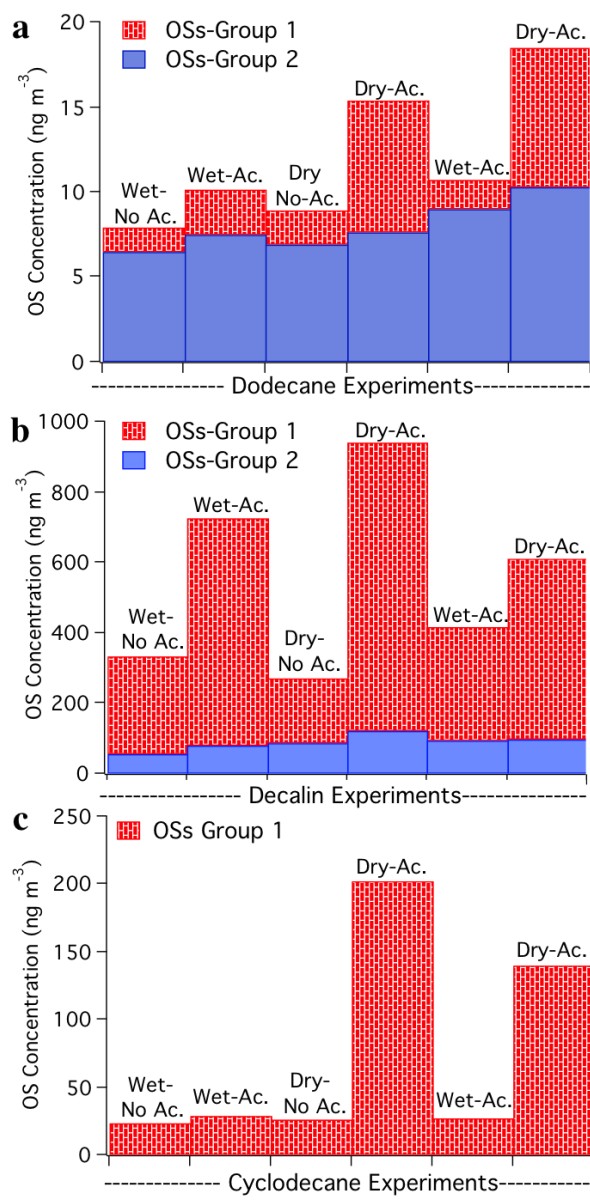

**Figure 5.** Impact of acidity on OS formation from gas-phase oxidation of (a) dodecane, (b) decalin, and (c) cyclodecane. OSs from Group-1 corresponds to compounds strongly impacted by aerosol acidity, while OSs from Group-2 appeared to have less dependency on aerosol acidity.



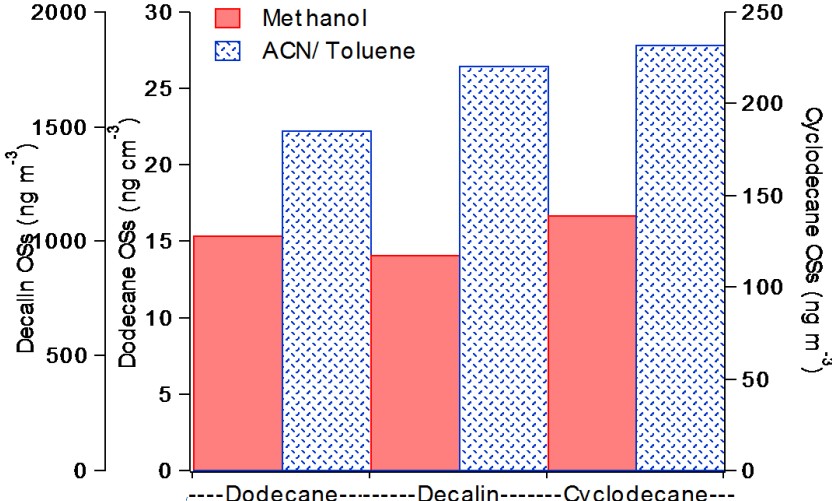

**Figure 6.** Impact of extraction solvent composition on quantification of identified OSs from gas-phase oxidation of alkanes.



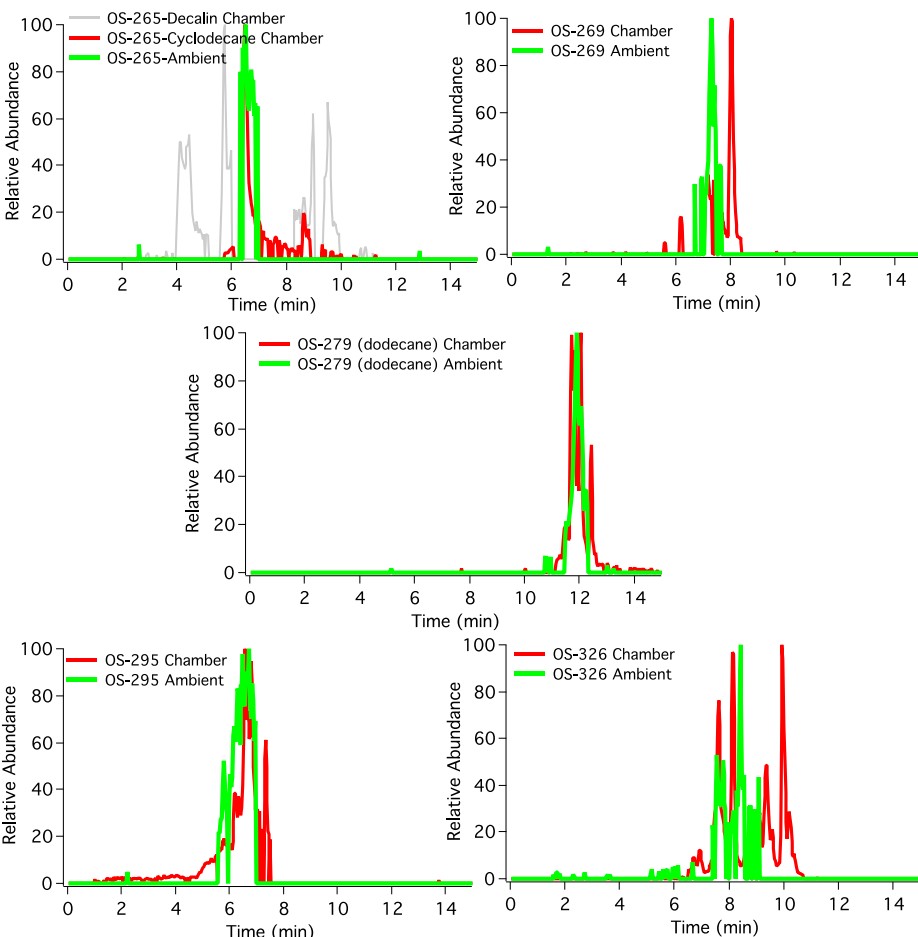

**Figure 7.** Extracted ion chromatograms (EICs) for selected alkane OSs identified in both smog chamber experiments (in red) and ambient samples (in green).