# Peer review of "Chemical Characterization of Organosulfates in Secondary Organic Aerosol Derived from the Photooxidation of Alkanes"

_Atmospheric Chemistry and Physics, 2016_

## Referee Comment (RC1) · Anonymous Referee #1 · 18 Feb 2016

This is an interesting study about the formation of organosulfates from the oxidation of aliphatic alkanes. As the authors point out in the manuscript, earlier studies mostly presented the formation of organosulfates from the oxidation of biogenic VOCs (isoprene, monoterpenes, and sesquiterpenes) or anthropogenic aromatic hydrocarbons, and this is one of the first studies to report the aliphatic alkane organosulfates. The authors conducted a series of well-designed chamber experiments, and filter sample analysis to elucidate the formation mechanisms and the structures of these organosulfates. While tandem MS experiments may not provide conclusive evidence for the structures, proposed formation mechanisms and resulting organosulfate structures are consistent with our current knowledge about atmospheric organosulfate formation pathways. In

addition, the authors support the importance of the aliphatic alkane organosulfates by providing evidence for their presence in ambient PM filter samples. The manuscript is concise and very well written. I suggest the manuscript be published as is.

---

## Referee Comment (RC2) · Anonymous Referee #2 · 21 Feb 2016

General Comments

In this manuscript the authors report results of an experimental study of the formation of organosulfates in secondary organic aerosol (SOA) formed from photochemical reactions of three alkanes: decane, dodecane, and decalin, conducted in an outdoor smog chamber. The SOA was collected on filters and analyzed using liquid chromatography-mass spectrometry to determine elemental formulas for organosulfates and to quantify the compounds. Products observed in experiments were also observed in samples collected in Pakistan and Pasadena, indicating that they can be used as tracers for SOA formation from these alkanes in ambient air. The study is technically well done and the paper is well written. I think it will eventually be suitable for publication in ACP, but I

have a number of comments that should first be addressed. Most importantly, I think the proposed reaction mechanisms for forming the identified organosulfate products are highly implausible, and that other mechanisms are much more likely.

Specific Comments

1. Lines 58-60: I am not aware that it is known that the primary source of SOA model-measurement discrepancies is IVOCs. I consider this to be an ongoing debate, and that there are other sources, such as the effects of vapor wall loss on measured SOA yields, effects of multiphase chemistry, and others. The authors seem to have picked a couple references to support their particular view.

2. Lines 115-118: Why were these compounds chosen? Decane makes sense based on potential abundance, but what about decalin and dodecane? The latter two are interesting from a structural point of view, but I was under the impression that this study was interested in compounds likely to contribute significantly to ambient SOA formation. A little more discussion of the choice of these compounds is warranted.

3. Line 126: Experimental. Were any blank chamber experiments conducted to determine the effect of background air components on SOA formation?

4. Were background VOCs identified/quantified?

5. Line 226-230: The authors have mistakenly assumed that the results of a condensed phase oxidation study can be applied to the gas phase. Ruehl et al. observed enhanced OH reaction at the ends of alkane molecules in drops because of the orientation of the molecules with respect to the liquid surface. In the gas phase no such preference occurs, as that study also showed. In fact, it is well established that the primary H atoms on terminal carbons are about 10 times less reactive with OH radicals than the secondary H atoms on internal carbon atoms (Kwok and Atkinson, Atmos. Environ. (1995), 29, 1685-1695). Reaction occurs preferentially on internal carbons.

6. Line 239-240: Raff and Finlayson-Pitts do not show that RO2-RO2 chemistry dominates in isopropyl nitrite photolysis, only that it contributes to the chemistry. Because $NO_2$ is photolyzed in these systems NO is recycled and so available for reaction with $RO_2$ radicals, even when $O_3$ is present. 7. My major criticism of this paper is that the proposed mechanisms for forming organosulfates with the same elemental composition as the observed products are highly implausible. The authors have ignored much of what is known about the rates of competing reaction pathways and assumed that because under some set of conditions a certain reaction can occur, that it is plausible for the conditions of these experiments. In doing so, it is assumed that essentially all reactions are possible here, ranging from auto-oxidation (which requires pristine conditions), to $RO_2$-$RO_2$ reactions (which require low NO and high VOC concentrations), to $RO_2$-$HO_2$ reactions (which require low NO and low VOC concentrations), to $RO_2$-NO reactions (which require high NO concentrations. The authors do not present any information on the conditions with regards to NO, $NO_2$, $O_3$, etc., and so no constraints are placed on the proposed mechanisms. Regardless, it is difficult to believe that all these conditions were encountered in these experiments.

Besides the problems outlined above, I have listed a few more detailed aspects of the mechanisms that are problematic.

Figure 3, Pathway A. The proposed $RO_2$ isomerization is much too slow to compete with other pathways ($RO_2$, $HO_2$, and NO reactions). See Crounse et al., J. Phys. Chem. Lett. (2013), 4, 3513-3520.

Figure 3, Pathway B. The proposed $RO_2$ isomerization through a 5-member ring, if even possible, would be much too slow to compete with isomerization through a 6-member ring to abstract a tertiary H-atom from the ring, though even this is much too slow to compete with other pathways ($RO_2$, $HO_2$, and NO reactions). See Crounse et al.

Figure 4, Pathway A. The proposed RC(O)O isomerization is much too slow (by about a factor of 106) to compete with decomposition to R + $CO_2$. See Vereecken and Peeters,

PCCP (2009), 11, 9062-9074; PCCP (2010), 12, 12608-12620.

Figure 4, Pathway B. The proposed RO2 isomerization is much too slow to compete with other pathways (RO2, HO2, and NO reactions). See Crounse et al. It is also not clear how the alkyl radical site adjacent to the –OOH group is formed.

Figure 4, Pathway C. Reaction involves three H-atom abstractions by OH radicals, the last two of which must occur at specific H-atoms, and with the last one occurring for a compound that would be expected to be in the particle phase where such reactions are negligibly slow.

These pathways are not only implausible, but if they did occur then there should be many other products that are much more likely to be present. If the authors insist on sticking with these mechanisms, then they should also address this issue. When presenting this kind of analysis it is not enough to show that there is a mechanism that could possibly explain the products, but also that other products predicted by such a mechanism are also present.

In my opinion, a much more plausible mechanism for explaining these products is that a series of compounds containing C–OH, C=O, and C–ONO2 groups were formed from well-established reactions of alkanes with OH radicals under high NO conditions, and that the sulfates were formed by nucleophilic substitution of the –ONO2 group by a –OSO3 group, a reaction that is known to occur in particles.

One simple test for the mechanisms proposed by the authors is to conduct an experiment with added NO, such that the NO concentration remains significant throughout the experiment. Under these conditions no organosulfates should be formed, since the presence of NO will prevent the formation of hydroperoxides, which are proposed precursors to organosulfate formation. However, if the organosulfates observed in the original experiments were formed through the suggested high NO chemistry, then the addition of NO will have no effect.

Technical Comments

None.

---

## Referee Comment (RC3) · Anonymous Referee #3 · 25 Feb 2016

**Review of "Characterization of Organosulfates in Secondary Organic Aerosol Derived from the Photooxidation of Long-Chain Alkanes"**

Reviewer's Summary:

The authors characterize organosulfates (OSs) from the laboratory oxidation of dodecane, decalin, and cyclodecane under varying conditions of humidity and two different seed types (non-acidified, acidified). They observe overlapping organosulfates in the laboratory experiments and on filters from Pasadena, USA and Lahore, Pakistan, concluding that OSs from the oxidation of anthropogenic precursors may contribute to urban SOA.  The results are novel and would be of interest to the readers of ACP; however, I would not recommend this manuscript for publication because it is not well-written and the conclusions are highly speculative.  In particular, the proposed chemical mechanisms from the laboratory experiments are not substantiated by a fundamental knowledge of the chemistry occurring in the reaction chambers used.  The authors inconsistently address the fate of the RO2 radical within their laboratory experiments throughout the text and within the proposed mechanisms.  There seems to be a mix of RO2 reacting with RO2, HO2, and NO, though they claim different regimes depending on what mechanism they are proposing to explain the OSs formed.  For example, they state that reaction with NO is insignificant, yet they report a nitrate containing OS in the decalin system.  They propose the formation of hydroperoxides in the case of dodecane experiments with high initial precursor concentrations and do not propose RO2 + RO2 chemistry, but for the C10 systems RO2 + RO2 reactions are proposed with some RO2 + HO2 reactions.  They propose epoxide precursors in the C10 systems to OS formation, but not in the C12 system.  In general, the proposed mechanisms are arbitrary and do not demonstrate careful control in the design of the experiments or understanding of the chemistry.  This lack of understanding becomes clear because there are several areas where citations are used to support the current work, but the citations are used imprecisely and out of context.  The manuscript would benefit from more clearly stated organizational structure (e.g. why some mechanisms are proposed in the main text versus the supplemental information).  The authors should also clarify motivation in the experimental selection of two C10 cyclic alkane structures and one C12 straight chain structure.  The brevity of the discussion of results on the OSs from dodecane photooxidation are quite brief relative to the other sections interpreting the results from decalin and cylodecane, and the effects of chemical structure are glossed over in brevity.  Further, it is unclear if the conclusion that enhancement of OS yields are due to increased acidity of the seed aerosol is really due to acidity, rather than an effect of seeding the experiments with an atomized solution containing more sulfate.  These concerns are outlined in detail below.

Major Comments:

1.  Lines 58-61: These lines are specious in the use of citations and misleading.  First, as written, these lines assert that the underestimate of global SOA is equivalent to an underestimate in urban SOA.  Second, the references cited (Pye and Pouliot, 2012; Tkacik et al., 2012) do not specifically argue that the underestimate in predicted SOA is due to the omission of IVOCs.  A better reference here based on the lines as written would be (Robinson et al., 2007).  Pye and Pouliot, 2012 can be cited for exploring additional mechanisms (oligomer formation) from alkane and PAH in SOA formation, and Tkacik et al., 2012 can be cited for providing additional evidence that IVOCs may be a missing source in modeling urban SOA, but the authors need to reword these lines carefully and be more precise.

2. Line 65: References here should include Yee et al., 2013 which more specifically addresses analogous to Lim and Ziemann, 2005 the products and mechanisms of C12 alkanes of varying structures.
3. Line 66: Tkacik et al., 2012 should be included here for presenting yields from several alkane systems.
4. Line 71: For this discussion on structure and fragmentation, additional reference should be cited (Lambe et al., 2012).
5. Line 107: The authors assert that acid-catalyzed reactive uptake has not been reported for the oxidation of alkanes. This is not true. Atkinson, Lim, and Ziemann have shown that alkane oxidation leading to 1,4-hydroxy carbonyls convert to cyclic hemiacetals in an acid-catalyzed multi-phase process (Dibble, 2007; Atkinson et al., 2008; Lim and Ziemann, 2009a, 2009b). Schilling Fahnestock et al., 2015 also report the effect of acidity on SOA formation from C12 alkanes.
6. Lines 123-124: Can the authors give more background on these two sites to orient the reader also with the motivation/purpose of this study? What types of sites are these—urban with what type of emissions profiles and surrounded by vegetation, etc.?
7. Line 222-223: It would be helpful to label the 1,3-dodecanone sulfate in Figure 1 to aid the reader. The authors should be careful with their naming convention here (i.e. 1,3-), as this particular isomer certainly is not the only potential isomer and is not the only isomer specified in Yee et al., 2012 and Yee et al, 2013.
8. Lines 226-230: The reference cited, Ruehl et al., 2013, is improperly used here. Ruehl et al., 2013 describes the heterogenous oxidation of octacosane and finds a strong preference for OH attack at the terminal carbons. The current work, however, is gas-phase oxidation, so the specificity of the isomers as listed in lines 228-229 should be rethought. Further, the naming convention for these isomers are inconsistent with the naming convention in line 222-223. It seems as though the 1, 3-dodecanone denotes the 1 position as the ketone, whereas here the reference to 2, 4-, 3,5-, and 4,6- and other isomers suggests the 1 position is likely the carbon at the end of the dodecane chain.
9. Lines 234-237: This is a poorly worded sentence. It is unclear in relation to the context of the current work, and there is imprecise use of citations. Hydroperoxides can undergo further oxidation by reaction with OH, but to generate alkoxy radicals from hydroperoxides, that would likely include photolysis. The authors need to address the extent of photolysis in the experiments then. Or are the authors referring to reactions of RO2 + NO to generate alkoxy radicals? If so, then the authors need to address the extent of RO2 + NO occurring in the experiments. If the former, rewrite as, "First-generation hydroperoxides can undergo further oxidation by reaction with OH to form low-volatility, more highly oxidized products, or can be photolyzed to alkoxy radicals (RO) to form more highly volatile products." The use of Carasquillo et al., 2014 here is inappropriate to discuss the oxidation of hydroperoxides as written. Carasquillo et al., 2014 describe the fate of differing alkoxy radical structures and how it affects SOA yield. The authors need to clarify what they are trying to say here and how it relates to the mechanism proposed in Figure 1.
10. Lines 237-243: The authors can cite Raff and Finlayson-Pitts, 2010 for IPN as an OH radical source, but it cannot be cited to fully account for the chemical conditions (i.e. the fate of $RO_2$) in the current experiments without considering the differences between their experiment and that

of Raff and Finlayson-Pitts, 2010. The authors should report $NO_x$ levels in these experiments to verify the claim that $RO_2$ + NO reactions are minimal. Also, how is $O_3$ formed in these experiments? The authors need to calculate (considering the relatively high levels of initial hydrocarbon), the relative fate of $RO_2$ between reaction with $RO_2$, $HO_2$, and NO. The proposal that OS-279 stems from hydroperoxide species in Figure 1 seems least inappropriate if $RO_2$ fate is really dominated by reaction with $RO_2$ and/or NO.

11. Table 2: The authors never describe the origin of the C7 (OS-209) and C9 (OS-237) organosulfates observed in the dodecane system and also observed in the ambient samples. This is another indication that fragmentation pathways are at play, potentially through RO2 + NO reactions in the system. The authors need to be careful in explaining the fate of the RO2 radical in their experiments and whether the ambient observation of these OSs can really be attributed to dodecane chemistry in the atmosphere when they may clearly originate from other precursors. The authors need to also describe the potential influence of monoterpenes at the sites they have taken samples from to preclude OS origin from biogenic precursors, as they say themselves that C10 monoterpene OSs are isomeric to some proposed in the C10 alkane systems. How good is RT matching/SICs for confirming that the laboratory generated OSs are really the same as those in ambient data? What measurements in these locations suggest that decalin, cylodecane, and dodecane are prevalent here?

12. Lines 246-247: The citation of Claeys et al., 2004 is inappropriate here. The authors propose that "heterogenous chemistry of gas-phase organic peroxide" is a mechanism for OS and tetrol formation, citing Claeys et al., 2004. Yet, Claeys et al., 2004 state, "The mechanism we suggest, reaction with hydrogen peroxide under acidic conditions in the aerosol liquid phase…," which is not consistent with the heterogeneous mechanism proposed in the current work and in Riva et al., 2015b. The difference in humidity should also affect the distribution of hydroperoxide compounds in the gas/particle phase. The authors should address this in the context of mechanistic explanations for their observations.

13. Lines 256: Several citations should be added here. Include reference to works by Lim and Ziemann, Lambe et al., Yee et al., Loza et al., Tkacik et al.

14. Lines 284-286: The mechanism described in text corresponds with the pathway in Figure S8, pathway c, not pathway a. Authors should rewrite these lines to describe pathway a. It also becomes clear here that the authors are not consistent with description of the chemistry proceeding in the chamber. In Figure S8, the fate of RO2 is initially reaction with RO2, but then in pathway a, it shifts to RO2+HO2. The selected pathways seem arbitrary to explain the proposed structure in Figure 2a.

15. Line 290: It is unclear whether the analytical technique is sufficient for seeing hydroperoxide moieties on molecules as they are included in the proposed structures. Were hydroperoxide standards such as t-buytylhydroperoxide or cumene hydroperoxide run using this method to verify that the hydroperoxide moiety can be retained on the column? Or is there something about the organosulfates that allow for this? The authors should address this in the experimental methods section as well.

16. Lines 295-296: The work of Yee et al., 2013 and Schilling Fahnestock et al., 2015 do not test decalin, so they should not be cited here to support proposed formation of a 1-hydroperoxy radical in the decalin system used in the current work. While the mechanisms laid out in Atkinson, 2000 can apply here, as worded it seems as if the authors are proposing the particular

alkoxyl radical ($C_{10}H_{17}O^{\cdot}$) was discussed in the reference.  Reword the sentence/omit the reference.

17. Line 298/Figure 3: To guide the reader it would be beneficial to update the mechanism in Figure 3 with the same label of tCARBROOH next to intermediate the authors are referring to

18. Lines 298/Figure 3: Do the authors see evidence of an analogous product in the case of decalin, as the OS-279 that was observed in the dodecane case?  The proposed mechanism of carbonyl hydroperoxide heterogenous reactive uptake followed by OS formation should also be considered for the decalin tCARBROOH as well and supported by the measurements/compared on the basis of volatility differences due to carbon number/ring structure and the impact of reactive uptake versus partitioning to the particle phase.

19. Lines 299-301: The mechanism of $RO_2$ radical isomerization/auto-oxidation is a special case in which the $RO_2$ radical lifetime is long enough for this to take place (e.g. remote areas).  Please also refer to and reference (Peeters et al., 2009; Crounse et al., 2011; Orlando and Tyndall, 2012).  The authors need to justify why the basis of their proposed mechanisms for OS formation rely on this pathway when the initial fate of $RO_2$ radicals in their systems are presented as $RO_2 + RO_2$ (e.g. Figures 3, 4, and S8). Is it RO2 + RO2, RO2 + HO2, RO2 + NO, RO2 isomerization?

20. Lines 301-305: The authors should address the extent of photolysis reactions affecting the fate of the proposed hydroperoxides and aldehydes in the system.

21. Line 306: "previously unreported" is unclear.  Do the authors mean previously unreported in ambient data or previously unreported from similar experiments?

22. Line 311: As worded, OS-267, is proposed to originate from further oxidation of OS-269, but the arrows drawn in Figure 3 are inconsistent suggesting origin from the epoxide.  Please clarify.

23. Line 327/Figure 4:  The description of Ring cleavage of the C10 alkoxy radical is not consistent with the "ISO"/isomerization descriptor in Figure 4.  Please clarify that pathway.

24. Line 340: Figure 1d does not exist.  Clarify the reference.

25. Lines 343-345: Sentence is awkward beginning with "Pathway c", and from what figure?  Clarify that it is Figure 4.  Again, citation of Atkinson, 2000 seems inappropriate as the sentence is written.

26. Lines 343-345/Figure 4, pathway c: Here the authors propose that RO2 + NO chemistry is occurring to form a nitrate containing OS.  This contradicts the authors' earlier statement in lines 237-243 stating that the RO2 + NO reactions are not significant in their experimental setup.  The authors need to handle in more detail the fate of RO2 under the unclear experimental conditions.

27. Lines 349-364: Why is discussion of OS-281 and OS-297 featured here, when discussion of OS-265 is discussed near the beginning of Section 3.2?  Since they are all referenced in Figure S8, their chemistry should be discussed together from the same mechanistic precursors.

28. Section 3.2: Authors should clarify the main mechanistic differences and relative importance between Figure S8, Figure 3, and Figure 4, and the flow of products to be discussed at the beginning of Section 3.2.  Currently as written, the flow of Section 3.2 is very arbitrary when choosing different OS products to discuss.

29. Lines 377-380: Incorrect use of citations here.  Yee et al., 2012 do not propose RO2 + RO2 chemistry and therefore a "precedent" has not been established.  The authors should not be citing Atkinson, 2000; Yee et al., 2012 and Raff and Finlayson-Pitts, 2010 to speak for the

experimental conditions in the current work.  The mechanism of RO2 + RO2 → RO can be supported by work on general atmospheric chemistry mechanisms including Atkinson, 2000 and many other works, and so if this mechanistic pathway is to be cited, than many other works should be cited as well.

30. Line 379: Inconsistent citation here compared to line 300 for similar mechanistic argument.

31. Figure S14: Why do the authors propose in the case of cyclodecane formation of the hydroperoxide from RO2 + HO2 pathways and subsequent chemistry thereof, but not in the case of decalin in any of Figures 3, 4, and S8?  Further, Figure S14 outlines a mechanism from further reaction of the hydroperoxide to get to an epoxide that then enters the particle phase to produce OS-251 and OS-249.  This seems like a plausible analogous mechanism to propose for the case of dodecane rather than reactive uptake of a carbonyl hydroperoxide.  Why do the authors propose different mechanisms between dodecane and cyclodecane to generate the similar analogs (OS-279, OS-249)?

32. Section 3.4: This section is weak and little effort is made to really describe the chemical differences between the systems to interpret the findings.  There should be comparisons of vapor pressures of the precursors and carbon numbers and discussion of previously published yields from these compounds to support the discussion.  How does quantification using the available OS surrogate standards potentially affect the OS quantification across these systems/factoring in different sensitivities?

33. Line 400: The authors claim that "the presence of acidic aerosols significantly increase OS formation in most cases".  However, is this just an effect of using an atomized solution with more sulfate (0.06M ammonium sulfate + 0.06M sulfuric acid) in the acidic case versus only 0.06M ammonium sulfate in the non-acidified case?  It may be a concerted effect of more available sulfate in the "acidic" case as well as acidity.

34. Line 484: The authors return to claim that the experiments are conducted under dominant "RO2/HO2" chemistry—this is contradictory to the formation of OS-326 containing a nitrate group.

35. Lines 484-486: Enhancement of OS due to acidified ammonium sulfate seed needs to be addressed with regard to the effect of just having introduced more sulfate into the experiments compared to the non-acidified case.  See earlier Major Comment, 33.

36. Lines 491-496: The "novel pathway" involving reactive uptake of hydroperoxides is not well-substantiated in the current work and is mostly speculation.  The vapor pressure alone of the carbonyl hydroperoxide makes it a potential candidate to partition to the particle phase, not via reactive uptake.  There are no direct measurements of hydroperoxides in the gas phase, and insufficient discussion on if hydroperoxides are detected in the particle phase using the UPLC technique.  Further, if reactive uptake is at play, why have the authors not seen the corresponding decalin analog of carbonyl hydroperoxide?

Minor Comments:

1. Line 30: "Both studies strongly support formation of OSs" is awkward.  Reword, for example, "Both studies strongly support that OSs can form from the gas-phase oxidation of anthropogenic precursors…"

2. Line 48: Change "aerosol" to "particles", as aerosol is technically defined as both gas + particle.

3. Line 76: Insert comma after "2015)".
4. Line 82: Insert comma after "$NO_3$".
5. Line 83: Change comma to semi-colon after "2007,".
6. Line 90: Delete "of".
7. Line 103: Change "reduce" to "reduces".
8. Line 136: Check misprint on the high humidity range listed as "(4-60%)".
9. Line 220: Insert after ")", ", hereafter referred to as OS-279, ".
10. Line 268: "ion at *m/z* 265.0749" should be "ion at *m/z* 265.0752" according to Figure 2.
11. Line 315: Add in "Figure 3, pathway a" to be clear.
12. Line 345: Change "identical" to "analogous" as the sequence of reactions are certainly not identical as shown in Figure 4.
13. Line 350: Change chemical formula to include S for OS-281.
14. Line 352: Rewrite the sentence. The radical reacts with O2, followed by 1,6 H shift, etc.
15. Lines 362-363: Rewrite the awkward phrasing, "which be reactively taken up to give a sulfate ester".
16. Lines 371-374: Poor grammar. Rewrite sentence.
17. Line 379: "hydroperoxydes" is spelled wrong.
18. Line 459: Add "of" after "oxidation".
19. Line 482: Add "," after "cyclododecane".

Additional References:

Atkinson, R.; Arey, J.; Aschmann, S. M. Atmospheric Chemistry of Alkanes: Review and Recent Developments. *Atmos. Environ.* **2008**, *42* (23), 5859–5871.

Crounse, J. D.; Paulot, F.; Kjaergaard, H. G.; Wennberg, P. O. Peroxy Radical Isomerization in the Oxidation of Isoprene. *Phys. Chem. Chem. Phys.* **2011**, *13* (30), 13607–13613.

Dibble, T. S. Cyclization of 1,4-Hydroxycarbonyls Is Not a Homogenous Gas Phase Process. *Chem. Phys. Lett.* **2007**, *447* (1-3), 5–9.

Lambe, A. T.; Onasch, T. B.; Croasdale, D. R.; Wright, J. P.; Martin, A. T.; Franklin, J. P.; Massoli, P.; Kroll, J. H.; Canagaratna, M. R.; Brune, W. H.; et al. Transitions from Functionalization to Fragmentation Reactions of Laboratory Secondary Organic Aerosol (SOA) Generated from the OH Oxidation of Alkane Precursors. *Environ. Sci. Technol.* **2012**, *46* (10), 5430–5437.

Lim, Y. Bin; Ziemann, P. J. Chemistry of Secondary Organic Aerosol Formation from OH Radical-Initiated Reactions of Linear, Branched, and Cyclic Alkanes in the Presence of NOx. *Aerosol Sci. Technol.* **2009a**, *43*, 604–619.

Lim, Y. Bin; Ziemann, P. J. Kinetics of the Heterogeneous Conversion of 1,4-Hydroxycarbonyls to Cyclic Hemiacetals and Dihydrofurans on Organic Aerosol Particles. *Phys. Chem. Chem. Phys.* **2009b**, *11* (36), 8029–8039.

Orlando, J. J.; Tyndall, G. S. Laboratory Studies of Organic Peroxy Radical Chemistry: An Overview with Emphasis on Recent Issues of Atmospheric Significance. *Chem. Soc. Rev.* **2012**, *41* (19), 6294.

Peeters, J.; Nguyen, T. L.; Vereecken, L. HO X Radical Regeneration in the Oxidation of Isoprene W. *Phys. Chem. Chem. Phys.* **2009**, *11*, 5935–5939.

Robinson, A. L.; Donahue, N. M.; Shrivastava, M. K.; Weitkamp, E. a; Sage, A. M.; Grieshop, A. P.; Lane, T. E.; Pierce, J. R.; Pandis, S. N. Rethinking Organic Aerosols: Semivolatile Emissions and Photochemical Aging. *Science* **2007**, *315* (5816), 1259–1262.

---

## Author Comment (AC1) · 29 Jun 2016

This is an interesting study about the formation of organosulfates from the oxidation of aliphatic alkanes. As the authors point out in the manuscript, earlier studies mostly pre- sented the formation of organosulfates from the oxidation of biogenic VOCs (isoprene, monoterpenes, and sesquiterpenes) or anthropogenic aromatic hydrocarbons, and this is one of the first studies to report the aliphatic alkane organosulfates. The authors conducted a series of well-designed chamber experiments, and filter sample analysis to elucidate the formation mechanisms and the structures of these organosulfates. While tandem MS experiments may not provide conclusive evidence for the structures, proposed formation mechanisms and resulting organosulfate structures are consistent with our current knowledge about atmospheric organosulfate formation pathways. In addition, the authors support the importance of the aliphatic alkane organosulfates by providing evidence for their presence in ambient PM filter samples. The manuscript is concise and very well written. I suggest the manuscript be published as is.

We thank Referee # 1 for careful consideration of our article.

---

## Author Comment (AC2) · 29 Jun 2016

General Comments

In this manuscript the authors report results of an experimental study of the formation of organosulfates in secondary organic aerosol (SOA) formed from photochemical reactions of three alkanes: decane, dodecane, and decalin, conducted in an outdoor smog chamber. The SOA was collected on filters and analyzed using liquid chromatography- mass spectrometry to determine elemental formulas for organosulfates and to quantify the compounds. Products observed in experiments were also observed in samples collected in Pakistan and Pasadena, indicating that they can be used as tracers for SOA formation from these alkanes in ambient air. The study is technically well done and the paper is well written. I think it will eventually be suitable for publication in ACP, but I have a number of comments that should first be addressed. Most importantly, I think the proposed reaction mechanisms for forming the identified organosulfate products are highly implausible, and that other mechanisms are much more likely.

**Specific Comments**

1. Lines 58-60: I am not aware that it is known that the primary source of SOA model-measurement discrepancies is IVOCs. I consider this to be an ongoing debate, and that there are other sources, such as the effects of vapor wall loss on measured SOA yields, effects of multiphase chemistry, and others. The authors seem to have picked a couple references to support their particular view.

We agree that it is an ongoing debate to explain the discrepancies between the mass of SOA observed in the atmosphere and the one estimated by the different models. IVOCs have not been traditionally included in chemical transport models and it is this what we wanted to underline. To avoid confusion, we have changed the sentence as follows on lines 59-62:

*"The omission of intermediate volatility organic compounds (IVOC) as SOA precursors, such as alkanes or polycyclic aromatic hydrocarbons (PAHs), could contribute in part to the underestimation of SOA mass observed in urban areas (Robinson et al., 2007; Tkacik et al., 2012)."*

2. Lines 115-118: Why were these compounds chosen? Decane makes sense based on potential abundance, but what about decalin and dodecane? The latter two are interesting from a structural point of view, but I was under the impression that this study was interested in compounds likely to contribute significantly to ambient SOA formation. A little more discussion of the choice of these compounds is warranted.

These compounds have been selected due to their potential contribution to SOA formation in the atmosphere. Recent studies have investigated the SOA formation from decalin and dodecane oxidations and reported large SOA yields (Yee et al., 2013; Hunter et al., 2014). Moreover, Pye and Pauliot (2012) have shown that, even though less emitted into the atmosphere, the cyclic $C_{10}$ alkanes have a greater potential for SOA formation than linear or branched alkanes $< C_{12}$.

A few sentences have been added to better explain our selection of parent VOCs on lines 118-124:

*"These alkanes were selected based on their potential contribution to atmospheric SOA formation (Hunter et al., 2014). Studies have demonstrated that cyclic compounds ($< C_{12}$) are expected to be more efficient SOA precursors than linear or branched alkanes with the same number of carbons (Lim and Ziemann, 2005; Pye and Pauliot, 2012). Alkanes $\geq C_{10}$ are considered as effective SOA precursors, especially when placed in the context of their emission rates (Pye and Pauliot, 2012)."*

3. Line 126: Experimental. Were any blank chamber experiments conducted to determine the effect of background air components on SOA formation?

As already described in the experimental section on lines 151-155, blank filters were collected before each experiment and analyzed following the protocol described in the article to characterize organosulfate (OS) composition. None of the identified OSs was observed in the blanks collected prior to each experiment. It is important to point out that the chambers were cleaned using a clean air generator for at least few hours and the concentration of VOCs are expected to be the same background level as $NO_x$ and $O_3$ (few ppb). Most importantly, due to the concentration of SOA precursors used in this work (> 150-200 ppb) the impact of VOCs present in the background could be considered negligible. Moreover, we did not observe any SOA constituents other than those derived from alkanes used in the chamber experiments.

4. Were background VOCs identified/quantified?

Gas-phase samples were collected before and during each experiment using GC-FID. As explained in the previous comment, the impact of any background VOCs could be considered negligible; especially since OA constituents were not observed from chamber filter blanks (filters collected from the flushed chamber).

5. Line 226-230: The authors have mistakenly assumed that the results of a condensed phase oxidation study can be applied to the gas phase. Ruehl et al. observed enhanced OH reaction at the ends of alkane molecules in drops because of the orientation of the molecules with respect to the liquid surface. In the gas phase no such preference occurs, as that study also showed. In fact, it is well established that the primary H atoms on terminal carbons are about 10 times less reactive with OH radicals than the secondary H atoms on internal carbon atoms (Kwok and Atkinson, Atmos. Environ. (1995), 29, 1685-1695). Reaction occurs preferentially on internal carbons.

We agree with the reviewer 2 and we have removed this sentence.

6. Line 239-240: Raff and Finlayson-Pitts do not show that $RO_2$-$RO_2$ chemistry dominates in isopropyl nitrite photolysis, only that it contributes to the chemistry. Because $NO_2$ is photolyzed in these systems NO is recycled and so available for reaction with $RO_2$ radicals, even when $O_3$ is present.

We did not mention that $RO_2$ chemistry is dominated by $RO_2 + RO_2$ reactions as it seems to be stipulated by referee #2. As it is underlined in Raff and Finlayson-Pitts *"without additional NO injection, the yields drop to 0.70 ± 0.01 and 0.58 ± 0.03, respectively. The dramatic*

*differences in product yields likely reflect differences in RO$_2$ + HO$_2$ and RO$_2$ + RO$_2$ pathways favored in the low-NOx situation versus the RO$_2$ + NO pathways that dominate under high-NO$_x$ conditions"*. Thus, this previous study revealed that without additional injection of NO, RO$_2$ chemistry could be considered as low-NO$_x$, as already stipulated on lines 256-257.

In our experiment we did not add NO (prior to IPN injection) and background NO levels were measured near the detection limit of the NO$_x$ monitor (i.e., 1 ppb). After IPN injection an significant increase of O$_3$ was observed in all experiments (as described lines 258-260) and NO concentration dropped below 1 ppb. We agree that NO$_2$ is photolyzed in these systems and NO is recycled. However, under the conditions described here (and in the article on lines 258-260 and in Table 1) most of the NO is expected to react with O$_3$. Rate constants for the RO$_2$ + NO and NO + O$_3$ reactions were determined to be $4.7 \times 10^{-12}$ cm$^3$ molecule$^{-1}$ s$^{-1}$ ($k_1$) (suggested by MCM; Ehn et al., 2014) and $1.8 \times 10^{-14}$ cm$^3$ molecule$^{-1}$ s$^{-1}$ (Atkinson et al., 2000; IUPAC) ($k_2$), respectively. Gratien et al. (2010, ES&T, 44, 8150-8155) have calculated the OH radical concentration to be $5 \times 10^6$ molecule cm$^{-3}$ from the photolysis of 5 ppm of isopropyl nitrite (IPN). In our experiments, 0.1-0.25 ppm of IPN was injected into the chambers. Using similar OH radical concentration to Gratien et al. (2010, ES&T) as an upper limit, RO$_2$ concentration could be estimated to be at the ppt level. Therefore, under the conditions of our study and assuming an O$_3$ concentration of 0.5 ppm, NO would react primarily with O$_3$: $(k_2[O_3][NO])/(k_1[RO_2][NO]) > 350$.

Lines 258-260 have been changed to:
*"Although RO$_2$ radicals could also react with NO formed by either IPN or NO$_2$ photolysis, formation of ozone under chamber conditions (0.3-0.6 ppm, depending on the concentration of IPN injected, Table 1) would rapidly quench NO (Atkinson et al., 2000). Therefore, RO$_2$ + NO reactions are not expected to be significant."*

7. My major criticism of this paper is that the proposed mechanisms for forming organosulfates with the same elemental composition as the observed products are highly implausible. The authors have ignored much of what is known about the rates of competing reaction pathways and assumed that because under some set of conditions a certain reaction can occur, that it is plausible for the conditions of these experiments. In doing so, it is assumed that essentially all reactions are possible here, ranging from auto-oxidation (which requires pristine conditions), to RO2-RO2 reactions (which require low NO and high VOC

concentrations), to RO2-HO2 reactions (which require low NO and low VOC concentrations), to RO2- NO reactions (which require high NO concentrations. The authors do not present any information on the conditions with regards to NO, $NO_2$, $O_3$, etc., and so no constraints are placed on the proposed mechanisms. Regardless, it is difficult to believe that all these conditions were encountered in these experiments.

We have considered the rates of competing reaction pathways and respectfully disagree with reviewer 2. First, as demonstrated by Ehn et al. (2014), at ppb levels of NO (1-5 ppb) a competition exists between $RO_2$ + NO, $RO_2$ + $HO_2$ and $RO_2$ autoxidation reactions. Indeed, as presented in the Ehn study, ELVOC, even though reduced, still formed at NO concentrations greater than a few ppb and underlines that auto-oxidation does not occur solely under pristine conditions. In other words, different $RO_2$ termination pathways could compete under specific conditions. It is important to point out that the concentrations of VOCs used in this work could lead to $RO_2$ + $RO_2$ chemistry.

In order to give more evidence of our tentatively proposed mechanisms, total peroxide measurements have been performed and added to our study. These results, which are now reported in Table 1, reveal that organic peroxides contribute to ~ 28 % (on average) of the SOA mass formed from the photooxidation of the studied alkanes. These measurements highlight the significant presence of peroxides in aerosol and add support to the proposed mechanisms. In addition to the peroxide measurements, concentrations of $O_3$ and NO were also added in Table 1 to support the low-NO conditions observed in this work. It should be pointed out that based on the GC-FID measurements ~ 85-90% of the VOC remained present when the concentration of NO dropped below the detection limit.

In addition, Crounse et al. (2013) (reference provided by Referee #2) have also shown that $RO_2$ autooxidation and formation of hydroperoxides occurred in experiments using methyl nitrite, similar to IPN, as an OH radical source.

Finally, we would like to stress to this reviewer that the analytical work and interpretation of the $MS^2$ spectra were not questioned by any of the 3 reviewers. The tandem MS data support the proposed structures. However, we have stated in many places throughout the text that the reaction pathways are tentatively proposed and until authentic standards become available these products remain tentatively identified. The main emphasis of our paper is that aliphatic

organosulfates do form from alkane oxidation, and likely offer one explanation as to why many groups have reported aliphatic OSs in urban areas (Ma et al., 2014; Tao et al., 2014)

A few sentences have been added to the revised manuscript to discuss the $RO_2$ chemistry on lines 404-406:

*"Indeed, Ehn et al. (2014) have demonstrated that NO reactions could be competitive at ppb levels. Under our experimental conditions $RO_2 + NO$, $RO_2 + HO_2/RO_2$ and $RO_2$ autoxidation are possible."*

A description of the organic peroxide measurements been added to the manuscript on lines 211-219:

*"**Total Organic Peroxide Analysis.** The total amount of organic peroxides in the SOA was quantified using an iodometric-spectrophotometric method adapted from Docherty et al. (2005). As described in Surratt et al. (2006), the method employed in this work slightly differs in the choice of extraction solvent: we used a 50:50 (v/v) mixture of methanol and ethyl acetate, rather than pure ethyl acetate. Calibrations and measurements were performed at 470 nm using a Hitachi U-3300 dual beam spectrophotometer. Benzoyl peroxide was used as the standard for quantification of organic peroxides formed from alkane oxidations. The molar absorptivity measured from the calibration curve was ~ 825, which is in excellent agreement with previously reported values (Docherty et al., 2005; Surratt et al., 2006)."*

A description of the organic peroxide results has been added on lines 261-265:

*"In addition, total organic peroxide aerosol concentrations, presented in Table 1, reveal that organic peroxides account (on average) for 28 % of the SOA mass measured in the different experiments in support of a significant contribution of $RO_2 + RO_2/HO_2$ and/or $RO_2$ autoxidation to SOA formation from alkane oxidations."*

Besides the problems outlined above, I have listed a few more detailed aspects of the mechanisms that are problematic.

Figure 3, Pathway A. The proposed $RO_2$ isomerization is much too slow to compete with other pathways (RO2, HO2, and NO reactions). See Crounse et al., J. Phys. Chem. Lett. (2013), 4, 3513-3520.

It is important to mention that we do not claim that the proposed mechanisms represent the major reaction pathways of the photooxidation of the studied alkanes, but are tentatively proposed to explain the formation of the OSs identified in this work. Indeed, concentrations reported in Tables S1-S3 underline that identified compounds are in low abundance regardless of the mass of SOA measured in all experiments. In the figure captions of each mechanism, we have added explicit statements that these are "tentatively proposed" mechanisms.

It is not clear how referee 2 could use the cited reference to point out that $RO_2$ isomerization is "much too slow to compete with other pathways". Here are listed some conclusions of the cited reference (the authors have calculated that "*the reactivity of $RO_2$ was 0.005 $s^{-1}$ with NO and 0.006 $s^{-1}$ with $HO_2$*" under the conditions of this study):

- *"Importantly, there is no evidence for the formation of $C_5$-hydroxy carbonyl nitrate (Scheme 2, HCN) or $C_5$-hydroxy carbonyl hydroperoxide (Scheme 2, HHPC). This indicates that the second H-shift involving the transfer of a H-atom α to a hydroperoxide group (Scheme 1) or a hydroxy group (Scheme 2) is at least 10 times faster than the competing reactions with NO or $HO_2$ (i.e., >0.1 s−1)."*

- *"It appears that 1,4- (aldehydic H), 1,5-, 1,6-, 1,7-, and 1,8-H-shifts to peroxy radicals (as well as intermolecular H-shifts in the condensed phase) may all be important in the oxidation of organic species in the environment and deserving of additional study. "*

- *"Examining Table 2, it is clear that for many oxygenated hydrocarbons, autoxidation will be competitive with other peroxy radical chemistry."*

As mentioned by Crounse et al. 2013, 1,4-H – 1.8H shift may all be important in the oxidation of organic species and in Figure 3, pathway **a** proposed a 1,6-H shift suggests that this reaction pathway could occur. Recent studies from Rissanen et al. (2014; 2015) have also reported that 1,4 – 1.8-H shift reactions could contribute in formation of highly oxidized products.

Figure 3, Pathway B. The proposed RO2 isomerization through a 5-member ring, if even possible, would be much too slow to compete with isomerization through a 6- member ring to

abstract a tertiary H-atom from the ring, though even this is much too slow to compete with other pathways (RO2, HO2, and NO reactions). See Crounse et al.

Formation of OS-285 (Pathway B, Figure 3) is not explained by $RO_2$ isomerization (even though as discussed by Crounse et al., isomerization through a 5-member ring may be important) but from RO (formed from $RO_2 + RO_2$ reaction) isomerization through a 1,4-H-shift (Rissanen et al., 2014, 2015).

One sentence has been added on lines 363-370:

*"The structure proposed for OS-285 is based on the formation of reaction of the hydroperoxyperoxyl radical intermediate in pathway **b** with $RO_2$ followed by a 1,4-H shift (Rissanen et al., 2015) and addition of $O_2$ to give a hydroxyhydroperoxyperoxyl radical ($C_9H_{17}O_5^•$). $C_9H_{17}O_5^•$ could then lead to an epoxide by isomerization (Iinuma et al., 2009; Surratt et al., 2010; Jacobs et al., 2013; Mael et al., 2015) and form OS-285. $C_9H_{17}O_5^•$ could also react with $HO_2$ and form the corresponding $C_9$-hydroxydihydroperoxide ($C_9H_{18}O_5$), which could then undergo heterogeneous reaction and lead to OS-269 (Figure 3, pathway **b**)."*

Figure 4, Pathway A. The proposed RC(O)O isomerization is much too slow (by about a factor of 106) to compete with decomposition to R + CO2. See Vereecken and Peeters, PCCP (2009), 11, 9062-9074; PCCP (2010), 12, 12608-12620.

We agree that $CO_2$ elimination is likely the dominant pathway as it was suggested by previous work. However, similar reaction pathways have been proposed in recent studies (Yao et al., 2014; Sato et al., 2015; Zhang et al., 2015), suggesting that, even though not dominant, isomerization of acyloxy radical could occur.

We do not claim that that the tentatively proposed mechanisms represent the dominant reaction pathways from alkane photooxidation, but likely explain the formation of OSs identified in this work. All mechanisms are based on known gas-phase reactions, which have been proposed in previous studies.

One sentence has been added on lines 382-388:

*"The salient features of pathway a include oxidation of the $RO_2$ to 2-decalinone, formation of a $C_{10}$ alkoxy radical followed by ring cleavage of the $C_9−C_{10}$ decalin bond and further $RO_2$ isomerization (1,8-H shift) leading to a 4-(carboxy cyclohexyl)-1-hydroperoxybut-2-yl radical*

*via RO₂ chemistry. Although considered as a minor reaction pathway (Crounse et al., 2013), the acyloxy radical could lead to the epoxide from the isomerization of the O₂ adduct (Paulot et al., 2009; Yao et al., 2014; Zhang et al., 2015). Further acid-catalyzed ring opening of the epoxide leads to OS-295 ($C_{10}H_{15}O_8S^-$).”*

Figure 4, Pathway B. The proposed RO2 isomerization is much too slow to compete with other pathways (RO2, HO2, and NO reactions). See Crounse et al. It is also not clear how the alkyl radical site adjacent to the –OOH group is formed.

As discussed previously it is not clear how the referee could use Crounse et al. to argue that proposed isomerization pathways is “much too slow”.

An error was made in the mechanism and it has been corrected in the new version. Two 1,6 and 1,5-H-shift reactions lead to epoxide proposed in Figure 4. As mentioned by Crounse et al. (2013) the intramolecular barrier for H-shift to the RO₂ is minimal for 1,5- or 1,6-H-shift reactions.

Figure 4, Pathway C. Reaction involves three H-atom abstractions by OH radicals, the last two of which must occur at specific H-atoms, and with the last one occurring for a compound that would be expected to be in the particle phase where such reactions are negligibly slow.

We agree with reviewer that such oxidation processes are expected to be slow in particle phase. However, we would like to point out that it has been previously reported that highly oxidized multifunctional compounds were identified in the gas phase (Ehn et al., 2014; Rissanen et al., 2014; 2015). Therefore, gas-phase oxidation of $C_{10}H_{17}NO_6$, even though minor, could occur, leading to OS-326, which has been quantified in low abundance in the decalin-derived SOA.

These pathways are not only implausible, but if they did occur then there should be many other products that are much more likely to be present. If the authors insist on sticking with these mechanisms, then they should also address this issue. When presenting this kind of analysis it is not enough to show that there is a mechanism that could possibly explain the products, but also that other products predicted by such a mechanism are also present.

It is not clear how the referee #2 could conclude by using reference (Crounse et al., 2013) that the proposed pathways are “implausible”. Indeed this cited reference highlights “*that 1,4-(aldehydic H), 1,5-, 1,6-, 1,7-, and 1,8-H-shifts to peroxy radicals (as well as intermolecular*

*H-shifts in the condensed phase) may all be important in the oxidation of organic species in the environment and deserving of additional study."* Moreover, similar reaction pathways have been recently proposed from the oxidation of different VOCs and the reaction mechanisms proposed in this study are consistent with the previous studies.

It is important to point out again that, we do not argue that the proposed mechanisms represent the major pathways of photooxidation of the studied alkanes. If it was the case, the OSs identified in this work will represent most of the SOA mass measured during the different experiments, which is not the case. It should be pointed out that analyses are still ongoing to identify non-OSs reaction products in both gas and particle phases, but the main focus of this article is on the chemical characterization of OSs. Liquid chromatography coupled with electrospray ionization mass spectrometry is not a sensitive technique for detection of the intermediate reaction products proposed in the mechanisms (e.g. ketone, epoxide). As demonstrated previously, epoxides quickly react in the presence of acidified particles and are not observed in SOA (Minerath et al., 2009; Surratt et al., 2010; Mael et al., 2015). Hence, it appears difficult with the techniques used in this work to identify the primary/secondary products proposed in the mechanisms. As mentioned in the conclusion on lines 527-528 "*more work is required to validate pathway(s) leading to the formation of gaseous epoxy-products*".

In my opinion, a much more plausible mechanism for explaining these products is that a series of compounds containing C–OH, C=O, and C–ONO2 groups were formed from well-established reactions of alkanes with OH radicals under high NO conditions, and that the sulfates were formed by nucleophilic substitution of the –ONO2 group by a –OSO3 group, a reaction that is known to occur in particles.

We can't rule out potential nucleophilic substitution of the –ONO2 group by a –OSO3 group as proposed in previous work (Darer et al., 2011; Hu et al., 2011). However, gas-phase chemistry cannot explain formation of identified compounds only by C–OH, C=O, and C–ONO2 groups. As discussed above, organic peroxide compounds represent 28 % (on average) of the SOA mass, which supports the proposed alternative mechanisms.

One sentence has been added in the conclusion, lines 527-530:

*"However, more work is required to validate pathway(s) leading to the formation of gaseous epoxy-products, since OS formation from other chemical pathways such as nuclophilic*

*substitution of the –ONO2 group by a –OSO3 group cannot be ruled out (Darer et al., 2011; Hu et al., 2011)."*

One simple test for the mechanisms proposed by the authors is to conduct an experiment with added NO, such that the NO concentration remains significant throughout the experiment. Under these conditions no organosulfates should be formed, since the presence of NO will prevent the formation of hydroperoxides, which are proposed precursors to organosulfate formation. However, if the organosulfates observed in the original experiments were formed through the suggested high NO chemistry, then the addition of NO will have no effect. Photooxidation of dodecane has been investigated using an additional injection of NO (200 ppb) prior to IPN injection. NO concentration dropped below ppb levels in less than 1 hour and OS concentrations were significantly reduced (factor of 3-4) compared to other experiments, underlying that NO concentration does have an impact on OS formation.

One sentence has been added to discuss the findings of this experiment, lines 274-277:
*"It should be mentioned that photooxidation of dodecane has also been investigated using an additional injection of NO (200 ppb) prior IPN injection. In this experiment SOA formation was significantly reduced as well as the OS concentrations (factor of 3-4), confirming that NO strongly impacts the formation of OS, such as OS-279."*

Technical Comments None.

---

## Author Comment (AC3) · 29 Jun 2016

Response to Anonymous Referee #3

We thank Referee # 3 for the comments and address each below. Our responses are denoted in blue texts.

Review of "Characterization of Organosulfates in Secondary Organic Aerosol Derived from the Photooxidation of Long-Chain Alkanes"

Reviewer's Summary:

The authors characterize organosulfates (OSs) from the laboratory oxidation of dodecane, decalin, and cyclodecane under varying conditions of humidity and two different seed types (non-acidified, acidified). They observe overlapping organosulfates in the laboratory experiments and on filters from Pasadena, USA and Lahore, Pakistan, concluding that OSs from the oxidation of anthropogenic precursors may contribute to urban SOA. The results are novel and would be of interest to the readers of ACP; however, I would not recommend this manuscript for publication because it is not well-written and the conclusions are highly speculative. In particular, the proposed chemical mechanisms from the laboratory experiments are not substantiated by a fundamental knowledge of the chemistry occurring in the reaction chambers used. The authors inconsistently address the fate of the RO2 radical within their laboratory experiments throughout the text and within the proposed mechanisms. There seems to be a mix of RO2 reacting with RO2, HO2, and NO, though they claim different regimes depending on what mechanism they are proposing to explain the OSs formed. For example, they state that reaction with NO is insignificant, yet they report a nitrate containing OS in the decalin system.

First, with regard to $RO_2$ chemistry: as demonstrated by Ehn et al. (2014), at ppb levels of NO (1-5 ppb; NO concentration in our study, < 1 ppb based on NO measurement) competition exists between $RO_2 + NO$, $RO_2 + HO_2$ and $RO_2$ autoxidation reactions. Nevertheless, ELVOC, though reduced, still formed, indicating that auto-oxidation does not occur solely under pristine conditions. It is important to point out that the high concentrations of VOCs used in this work favor involvement of $RO_2 + RO_2$ chemistry. In addition, previous work (Crounse et al., 2013), has also reported different $RO_2$ regimes, such as autooxidation or $RO_2 + HO_2$ reactions, in experiments using methyl nitrite as an OH radical source, similar to isopropyl

nitrite used in our study. As discussed below in response to reviewer's comment # 10, $RO_2$ + NO reactions are minimal; however, the nitrated OSs at *m/z* 326 are also measured in low concentrations (ng/m$^3$, Table S3).

Second: we do not claim that the proposed mechanisms represent the major reaction pathways of the photooxidation of the studied alkanes, but are tentatively proposed to explain the formation of the OSs identified in this study. Mechanisms have been clearly indicated as *proposed* branching of pathways of the alkane photooxidations presented to explain formation of specific OSs products consistent with MS$^2$ data. This approach to rationalizing OH oxidation products is universally applied in oxidation studies (Yee et al 2013; Bugler et al., 2015). Furthermore, the concentrations reported in Tables S1-S3 emphasize the fact that identified compounds are in low abundance regardless of the mass of SOA measured in all experiments.

Finally: we stress to Reviewer #3 that neither the analytical work nor the interpretation of the MS$^2$ data were questioned. The tandem MS data are consistent with the structures proposed for products observed in both the lab-generated and ambient aerosol samples and we repeat that we clearly indicated in the text that the mechanisms presented are suggested as pathways leading to ions consistent with those observed and until authentic standards become available both the product structures and mechanisms of formation remain tentative. An additional and crucial comment we make is that the major objective of our study is to demonstrate that aliphatic organosulfates form via alkane oxidation, and offer one explanation for reports of aliphatic OSs in urban areas (Ma et al., 2014; Tao et al., 2014).

They propose the formation of hydroperoxides in the case of dodecane experiments with high initial precursor concentrations and do not propose RO2 + RO2 chemistry, but for the C10 systems RO2 + RO2 reactions are proposed with some RO2 + HO2 reactions. They propose epoxide precursors in the C10 systems to OS formation, but not in the C12 system. In general, the proposed mechanisms are arbitrary and do not demonstrate careful control in the design of the experiments or understanding of the chemistry. This lack of understanding becomes clear because there are several areas where citations are used to support the current work, but the citations are used imprecisely and out of context. The manuscript would benefit from more

clearly stated organizational structure (e.g. why some mechanisms are proposed in the main text versus the supplemental information).

First we would like to point out that pathways have been proposed that lead to structures consistent with the mass spectrometric data acquired for the observed OSs. We have not attempted to hypothesize general mechanism that would be predicted to give the entire array of precursors contributing to the total mass of SOA. Regarding the possibility of an epoxide precursor to OS-279, we considered the possibility of formation of OS-279 from the reactive uptake of the corresponding epoxide ($C_{12}H_{24}O$); however, the composition of OS-279 (1 DBE) is inconsistent with reactive uptake of an epoxide. Therefore, we have used the few studies available in the literature (Yee et al., 2012; 2013) to propose the formation of OS-279 from the heterogeneous chemistry of hydroperoxides. Finally, we have considered the potential heterogeneous chemistry of hydroperoxides formed from the photooxidation of decalin and cyclodecane as discussed above.

The authors should also clarify motivation in the experimental selection of two C10 cyclic alkane structures and one C12 straight chain structure.

These compounds have been selected due to their potential contribution to SOA formation in the atmosphere. Recent studies have investigated the SOA formation from decalin and dodecane oxidations and reported large SOA yields (Yee et al., 2013; Hunter et al., 2014). Moreover, Pye and Pauliot (2012) have shown that, even though less emitted into the atmosphere, the cyclic $C_{10}$ alkanes have a greater potential for SOA formation than linear or branched alkanes $< C_{12}$.

A few sentences have been added to better explain our selection of parent VOCs on lines 118-122:

*"These alkanes were selected based on their potential contribution to atmospheric SOA formation (Hunter et al., 2014). Studies have demonstrated that cyclic compounds ($< C_{12}$) are expected to be more efficient SOA precursors than linear or branched alkanes with the same number of carbons (Lim and Ziemann, 2005; Pye and Pauliot, 2012). Alkanes $\geq C_{10}$ are considered as effective SOA precursors, especially when placed in the context of their emission rates (Pye and Pauliot, 2012)."*

The brevity of the discussion of results on the OSs from dodecane photooxidation are quite brief relative to the other sections interpreting the results from decalin and cylodecane, and the effects of chemical structure are glossed over in brevity.

As mentioned in lines 229-231, low abundances of OS-209 and OS-237 precluded acquisition of high-resolution $MS^2$ data, and thus, structures have not been proposed for the parent ions. Without structural information, discussion of formation pathways is not possible. The criticism of our conclusions as "highly speculative," is not consistent with request for more detail on the chemical structure of products formed in abundances too low to obtain high-resolution $MS^2$ data.

Further, it is unclear if the conclusion that enhancement of OS yields are due to increased acidity of the seed aerosol is really due to acidity, rather than an effect of seeding the experiments with an atomized solution containing more sulfate. These concerns are outlined in detail below.

This point has been previously discussed and published work demonstrates that acidity, rather than concentration of sulfate, is the key parameter (reference cited in the article) in the formation of OSs. Chan et al. (2011) have reported that the formation of OSs from the oxidation of $\beta$-caryophyllene is directly correlated with the aerosol acidity ($[H^+]$).

Major Comments:

1. Lines 58-61: These lines are specious in the use of citations and misleading. First, as written, these lines assert that the underestimate of global SOA is equivalent to an underestimate in urban SOA. Second, the references cited (Pye and Pouliot, 2012; Tkacik et al., 2012) do not specifically argue that the underestimate in predicted SOA is due to the omission of IVOCs. A better reference here based on the lines as written would be (Robinson et al., 2007). Pye and Pouliot, 2012 can be cited for exploring additional mechanisms (oligomer formation) from alkane and PAH in SOA formation, and Tkacik et al., 2012 can be cited for providing additional evidence that IVOCs may be a missing source in modeling urban SOA, but the authors need to reword these lines carefully and be more precise.

    Sentence has been modified on lines 59-62 as follows:

    *"The omission of intermediate volatility organic compounds (IVOC) as SOA precursors, such as alkanes or polycyclic aromatic hydrocarbons (PAHs), could contribute in part to the underestimation of SOA mass observed in urban areas (Robinson et al., 2007; Tkacik et al., 2012)."*

2. Line 65: References here should include Yee et al., 2013 which more specifically addresses analogous to Lim and Ziemann, 2005 the products and mechanisms of C12 alkanes of varying structures.

References have been added.

3. Line 66: Tkacik et al., 2012 should be included here for presenting yields from several alkane systems.

Reference has been added.

4. Line 71: For this discussion on structure and fragmentation, additional reference should be cited (Lambe et al., 2012).

Reference has been added.

5. Line 107: The authors assert that acid-catalyzed reactive uptake has not been reported for the oxidation of alkanes. This is not true. Atkinson, Lim, and Ziemann have shown that alkane oxidation leading to 1,4-hydroxy carbonyls convert to cyclic hemiacetals in an acid-catalyzed multi-phase process (Dibble, 2007; Atkinson et al., 2008; Lim and Ziemann, 2009a, 2009b). Schilling Fahnestock et al., 2015 also report the effect of acidity on SOA formation from C12 alkanes.

This statement has been removed from the revised version of the manuscript.

6. Lines 123-124: Can the authors give more background on these two sites to orient the reader also with the motivation/purpose of this study? What types of sites are these—urban with what type of emissions profiles and surrounded by vegetation, etc.?

As it is highlighted in the abstract, the motivation for our study was to demonstrate the formation of OSs from the oxidation of alkanes, which has been inferred from previous field studies. Description of both sites is presented in the experimental section and references, which have already characterized both sites, are cited.

The following sentences have been added:

Lines 185-188: *"As stipulated previously at both urban sites, anthropogenic activities (e.g., vehicular exhaust, industrial sources, cooking, etc.) likely dominated the organic aerosol mass fraction of $PM_{2.5}$ (Stone et al., 2010; Hayes et al., 2013). In addition,*

*Gentner et al. (2012) have reported significant emission of long-chain alkanes during the CalNex field study."*

7. Line 222-223: It would be helpful to label the 1,3-dodecanone sulfate in Figure 1 to aid the reader. The authors should be careful with their naming convention here (i.e. 1,3-), as this particular isomer certainly is not the only potential isomer and is not the only isomer specified in Yee et al., 2012 and Yee et al, 2013.

The OS has been labeled in Figure 1. We agree with referee #3 that other isomers are present since we identified at least 3 isomers as mentioned on line 236 and reported in Table S1.

The sentence has been modified on lines 239-242 to*:*

*"Kwok and Atkinson (1995) have reported that OH oxidation of long-chain alkanes preferentially occurred at an internal carbon and thus multiple isomers may be proposed. Based on Yee et al. (2012; 2013) one isomer may be, however, assigned as 6-dodecanone-4-sulfate."*

8. Lines 226-230: The reference cited, Ruehl et al., 2013, is improperly used here. Ruehl et al., 2013 describes the heterogenous oxidation of octacosane and finds a strong preference for OH attack at the terminal carbons. The current work, however, is gas-phase oxidation, so the specificity of the isomers as listed in lines 228-229 should be rethought. Further, the naming convention for these isomers are inconsistent with the naming convention in line 222-223. It seems as though the 1, 3-dodecanone denotes the 1 position as the ketone, whereas here the reference to 2, 4-, 3,5-, and 4,6- and other isomers suggests the 1 position is likely the carbon at the end of the dodecane chain.

Sentence has been removed and naming of the molecule is now consistent with line 242. As discussed by Kwok and Atkinson (1995), reaction occurs preferentially on internal carbons and the sentence has been changed as proposed in the previous point.

9. Lines 234-237: This is a poorly worded sentence. It is unclear in relation to the context of the current work, and there is imprecise use of citations. Hydroperoxides can undergo further oxidation by reaction with OH, but to generate alkoxy radicals from hydroperoxides, that would likely include photolysis. The authors need to address the extent of photolysis in the experiments then. Or are the authors referring to reactions of

RO2 + NO to generate alkoxy radicals? If so, then the authors need to address the extent of RO2 + NO occurring in the experiments. If the former, rewrite as, "First-generation hydroperoxides can undergo further oxidation by reaction with OH to form low-volatility, more highly oxidized products, or can be photolyzed to alkoxy radicals (RO) to form more highly volatile products." The use of Carasquillo et al., 2014 here is inappropriate to discuss the oxidation of hydroperoxides as written. Carasquillo et al., 2014 describe the fate of differing alkoxy radical structures and how it affects SOA yield. The authors need to clarify what they are trying to say here and how it relates to the mechanism proposed in Figure 1.

We thank the reviewer for its comments, sentences have been changed to simplify and clarify this paragraph.

Lines 249-255: "*First-generation hydroperoxides ($C_{12}H_{26}O_2$) can undergo further oxidation by reaction with OH to form either more highly oxidized products, such as dihydroperoxides ($C_{12}H_{26}O_4$), or semi-volatile products ($C_{12}H_{24}O$) (Yee et al., 2012). In addition, hydroperoxides can be photolyzed to alkoxy radicals (RO) to form more highly oxidized products. Low-volatility products could then condense onto sulfate aerosols and undergo further heterogeneous reactions (Schilling Fahnestock et al., 2015) leading to OSs as discussed below.*"

10. Lines 237-243: The authors can cite Raff and Finlayson-Pitts, 2010 for IPN as an OH radical source, but it cannot be cited to fully account for the chemical conditions (i.e. the fate of $RO_2$) in the current experiments without considering the differences between their experiment and that of Raff and Finlayson-Pitts, 2010. The authors should report NOx levels in these experiments to verify the claim that RO2 + NO reactions are minimal. Also, how is O3 formed in these experiments? The authors need to calculate (considering the relatively high levels of initial hydrocarbon), the relative fate of RO2 between reaction with $RO_2$, $HO_2$, and NO. The proposal that OS-279 stems from hydroperoxide species in Figure 1 seems least inappropriate if $RO_2$ fate is really dominated by reaction with RO2 and/or NO.

In order to provide additional support for the proposed mechanisms, total organic peroxide measurements have been performed. These results which are now reported in Table 1 reveal that organic peroxides (including hydroperoxides) could contribute up to ~ 28 % (on average) of the SOA mass formed from the photooxidation of the precursors

used in this work. These measurements highlight the significant presence of organic peroxides and/or hydroperoxides in aerosol and thus support the proposed mechanisms. In addition to the organic peroxide measurements, concentrations of $O_3$ and NO were also added in Table 1 to confirm the low-NO conditions cited in this work.

The description of the organic peroxide measurements has been added, lines 211-219, revised manuscript:

 *"**Total Organic Peroxide Analysis.** The total organic peroxides in the SOA were quantified using an iodometric-spectrophotometric method adapted from Docherty et al. (2005). As described in Surratt et al. (2006), the method employed in this work differs in the choice of extraction solvent: we used a 50:50 (v/v) mixture of methanol and ethyl acetate, rather than pure ethyl acetate. Calibrations and measurements were performed at 470 nm using a Hitachi U-3300 dual beam spectrophotometer. Benzoyl peroxide was used as the standard for quantification of organic peroxides formed from alkane oxidations. The molar absorptivity measured from the calibration curve was ~ 825, which is in excellent agreement with reported values (Docherty et al., 2005; Surratt et al., 2006)."*

A discussion of the results of the organic peroxide measurements has been added, lines 261-265, revised version:

*"In addition, total organic peroxide aerosol concentrations, presented in Table 1, reveal that organic peroxides account (on average) for 28 % of the SOA mass measured in the different experiments in support of a significant contribution of $RO_2$ + $RO_2/HO_2$ and/or $RO_2$ autoxidation to SOA formation from alkane oxidations."*

In our experiment we did not add NO (prior to IPN injection) and background NO levels were measured near the detection limit of the $NO_x$ monitor (i.e., 1 ppb). After IPN injection a significant increase of $O_3$ was observed in all experiments (as described lines 258-260) and NO concentration dropped below 1 ppb. We agree that $NO_2$ is photolyzed in these systems and NO is recycled. However, under the conditions described here (and in the article on lines 258-260 and in Table 1) most of the NO is expected to react with $O_3$. Rate constants for the $RO_2$ + NO and NO + $O_3$ reactions were determined to be $4.7 \times 10^{-12}$ cm$^3$ molecule$^{-1}$ s$^{-1}$ ($k_1$) (suggested by MCM; Ehn et al., 2014) and $1.8 \times 10^{-14}$ cm$^3$ molecule$^{-1}$ s$^{-1}$ (Atkinson et al., 2000; IUPAC) ($k_2$), respectively. Gratien et al. (2010, ES&T, 44, 8150-8155) have calculated the OH radical concentration to be $5 \times 10^6$ molecule cm$^{-3}$ from

the photolysis of 5 ppm of isopropyl nitrite (IPN). In our experiments, 0.1-0.25 ppm of IPN was injected into the chambers. Using similar OH radical concentration to Gratien et al. (2010, ES&T) as an upper limit, $RO_2$ concentration could be estimated to be at ppt levels. Therefore, under the conditions of our study and assuming an $O_3$ concentration of 0.5 ppm, NO would react predominantly with $O_3$: $(k_2[O_3]\,[NO])/(k_1[RO_2]\,[NO]) > 350$.

Lines 258-260 have been modified:

*"Although $RO_2$ radicals could also react with NO formed by either IPN or $NO_2$ photolysis, formation of ozone under chamber conditions (0.3-0.6 ppm, depending on the concentration of IPN injected, Table 1) would rapidly quench NO (Atkinson et al., 2000). Therefore, $RO_2$ + NO reactions are not expected to be significant."*

Finally, the photooxidation of dodecane has been also investigated using an additional injection of NO (200 ppb) prior IPN injection. NO concentration dropped below the ppb level in less than 1 hour and OS concentrations were significantly reduced (factor of 3-4) compared to other experiments, confirming that NO concentration does have an impact on OS formation.

One sentence has been added to describe this experiment, lines 274-277:

*"It should be mentioned that photooxidation of dodecane has also been investigated using an additional injection of NO (200 ppb) prior IPN injection. In this experiment SOA formation was significantly reduced as well as the OS concentrations (factor of 3-4), confirming that NO strongly impacts the formation of OS, such as OS-279."*

11. Table 2: The authors never describe the origin of the C7 (OS-209) and C9 (OS-237) organosulfates observed in the dodecane system and also observed in the ambient samples. This is another indication that fragmentation pathways are at play, potentially through RO2 + NO reactions in the system. The authors need to be careful in explaining the fate of the RO2 radical in their experiments and whether the ambient observation of these OSs can really be attributed to dodecane chemistry in the atmosphere when they may clearly originate from other precursors. The authors need to also describe the potential influence of monoterpenes at the sites they have taken samples from to preclude OS origin from biogenic precursors, as they say themselves that C10 monoterpene OSs are isomeric to some proposed in the C10 alkane systems. How good is RT matching/SICs

for confirming that the laboratory generated OSs are really the same as those in ambient data? What measurements in these locations suggest that decalin, cylodecane, and dodecane are prevalent here?

As mentioned in lines 229-231, low abundances of OS-209 and OS-237 precluded acquisition of high-resolution $MS^2$ data, and thus structures have not been proposed for these parent ions. Without any compositional information, reaction pathways cannot be discussed. As shown by Yee et al. (2012), hydroperoxides can be photolyzed to RO radicals, which fragment to smaller carbonyls. The potential formation of RO radicals from photolysis of hydroperoxides has been added in the revised version of the manuscript.

Isobaric compounds could likely be formed in the atmosphere, however, structures would be significantly different and isomers could be distinguished in most of the cases. Although we cannot completely rule out co-elution of some isobars, Figure 7 illustrates the most likely typical situation, in which isobars from the photooxidation of cyclodecane and decalin have different retention times (R.T.), allowing differentiation. OSs are known to form from the oxidation of monoterpenes and several isobaric OSs have been identified: OS-249, -251, -267, -279 and -326. Structures proposed in previous work are significantly different from structures proposed in this work and thus should be separated by liquid chromatography. We have analyzed ambient filters collected during SOAS campaign in rural areas (Centerville, Alabama, US) and find that the R.T.s of monoterpene-derived OSs are different from those of the OSs identified from the oxidation of the alkanes studied in this work.

For example:

OS-249: from monoterpenes: 10.3 min; cylodecane: 8.5/9.3

OS-279: from monoterpenes: 6.2 min; cylodecane: 5.8/6.8

We do not have access to potential collocated measurements during both field measurements, however, results proposed by Gentner et al (2012) tend to support significant emissions of long-chain alkanes in California and especially during CalNex.

The following sentences have been added:

Lines 185-188:

*"As stipulated previously at both urban sites, anthropogenic activities (e.g., vehicular exhaust, industrial sources, cooking, etc.) likely dominated the organic aerosol mass fraction of $PM_{2.5}$ (Stone et al., 2010; Hayes et al., 2013). In addition, Gentner et al. (2012) have reported significant emission of long-chain alkanes during the CalNex field study."*

11. Lines 246-247: The citation of Claeys et al., 2004 is inappropriate here. The authors propose that "heterogenous chemistry of gas-phase organic peroxide" is a mechanism for OS and tetrol formation, citing Claeys et al., 2004. Yet, Claeys et al., 2004 state, "The mechanism we suggest, reaction with hydrogen peroxide under acidic conditions in the aerosol liquid phase...," which is not consistent with the heterogeneous mechanism proposed in the current work and in Riva et al., 2015b. The difference in humidity should also affect the distribution of hydroperoxide compounds in the gas/particle phase. The authors should address this in the context of mechanistic explanations for their observations.

    We agree with reviewer that Claeys et al. is an inappropriate reference to be used here and we have removed it. We have shown in previous work and in another manuscript currently under review that organic peroxides could lead to OSs and polyols from aerosol-phase acid-catalyzed reactions. It is not clear how the RH could directly impact the distribution of the hydroperoxides as suggested by the reviewer. However, we have reported that the liquid water content of the aerosol plays an important role, but the acidity has a stronger impact on OS formations.

12. Lines 256: Several citations should be added here. Include reference to works by Lim and Ziemann, Lambe et al., Yee et al., Loza et al., Tkacik et al.

    References have been added.

13. Lines 284-286: The mechanism described in text corresponds with the pathway in Figure S8, pathway c, not pathway a. Authors should rewrite these lines to describe pathway a. It also becomes clear here that the authors are not consistent with description of the chemistry proceeding in the chamber. In Figure S8, the fate of RO2 is initially reaction with RO2, but then in pathway a, it shifts to RO2+HO2. The selected pathways seem arbitrary to explain the proposed structure in Figure 2a.

    Figure S8 describes sequential reactions via $RO_2$ leading to a ring opened ketoperoxy transient. Three branching reactions are available to transient: reaction with $HO_2$ leading

to the structure proposed for OS-265, pathway **a**; further reaction with an $RO_2$ species leading to the structure proposed for OS-265 and OS-281, pathway **b**; or isomerization and reaction with $O_2$ eventually leading to OS-281 and OS-297, pathway **c**. Figure S8 does not therefore represent a "shift" in chemistry, but branching reactions leading to three observed product ions. We note that the RO radical precursor to pathways a, b and c may also result from an $RO_2 + HO_2$ reaction (Kautzman et al., 2010; Birdsall et al., 2011). Since it is not possible to distinguish whether RO originates from $RO_2 + RO_2$ or $RO_2 + HO_2$ reactions the alternative $RO_2 + HO_2$ reaction has been added to all mechanisms.

This paragraph has been changed to be consistent with proposed mechanism.

Lines 310-314: *"A scheme leading to the structure proposed in Figure 2a is based on the cleavage of the $C_1 - C_2$ decalin bond, followed by reaction with a second $O_2$ molecule and $HO_2$ leads to a terminal carbonyl hydroperoxide ($C_{10}H_{18}O_3$) (Yee et al., 2013). $C_{10}H_{18}O_3$ could then further react with OH radicals and lead to an epoxide and sulfate ester by reactive uptake/heterogeneous chemistry (Paulot et al., 2009)."*

14. Line 290: It is unclear whether the analytical technique is sufficient for seeing hydroperoxide moieties on molecules as they are included in the proposed structures. Were hydroperoxide standards such as t-buytylhydroperoxide or cumene hydroperoxide run using this method to verify that the hydroperoxide moiety can be retained on the column? Or is there something about the organosulfates that allow for this? The authors should address this in the experimental methods section as well.

Such compounds can be retained on the column used in this project and we have demonstrated this for another project with a synthetic isoprene hydroxyhydroperoxide standard (ISOPOOH). As an example, please see the chromatogram below:

[Figure]

Moreover, Witkowski and Gierczak (2012) have recently developed a method to quantify hydroperoxide compounds formed from the ozonolysis of cyclohexene. The authors used a column similar ($C_{18}$) to that used in the present work. Electrospray ionization mass spectrometry operated in negative mode (Cech and Enke, 2001; Witkowski and Gierczak, 2012) is not highly sensitive to detection of non-acidic compounds, such as pure hydroperoxides or alcohols. However, LC/ESI(-)-MS provides excellent sensitivity for multifunctional compounds (like hydroperoxides and alcohols) containing the OS functional group, since the OS functional group yields an intense [M - H]$^-$ ion, as reported in many studies (Surratt et al., 2008; Kristensen et al., 2011; Kundu et al., 2013; Hansen et al., 2014).

As mentioned line 340-342, we expect to detect the presence of hydroxyl or hydroperoxide functional groups when the OS group is present in the multifunctional compounds analyzed by LC/ESI(-)-MS.

The following sentences has been added:

Lines 340-346: *"As a result, the presence of hydroperoxide and/or hydroxyl substituents is expected in order to satisfy the molecular formulas obtained by the accurate mass measurement. Although ESI-MS in the negative ion mode is not sensitive to multifunctional hydroperoxides and alcohols (Cech and Enke, 2001; Witkowski and Gierczak, 2012), this technique is highly sensitive to hydroperoxides and alcohols which also contain OS groups and give [M − H]$^-$ ions (Surratt et al., 2008; Kristensen et al., 2011; Kundu et al., 2013; Hansen et al., 2014)."*

15. Lines 295-296: The work of Yee et al., 2013 and Schilling Fahnestock et al., 2015 do not test decalin, so they should not be cited here to support proposed formation of a 1-hydroperoxy radical in the decalin system used in the current work. While the mechanisms laid out in Atkinson, 2000 can apply here, as worded it seems as if the authors are proposing the particular alkoxyl reference.

In Yee et al. 2013 they studied the oxidation of hexylcyclohexane and cyclododecane and they proposed (Figure 1, sidebar, Yee et al. 2013) a ring scission and formation of a terminal carbonyl hydroperoxide as proposed in Figure 3. We do not claim that both studies have investigated the oxidation of decalin but have used the analogous ring scission sequence to explain our products. To avoid confusion regarding the content of the

Yee citation, we have moved the citation in the text to follow the description of ring scission. Sentence has been modified in the revised manuscript.

Lines 348-352: *"Under low-NO$_x$ conditions, abstraction of a proton $\alpha$ to the ring scission of decalin followed by reaction with O$_2$ leads to the 1-hydroperoxy radical, which in turn can react with another RO$_2$ radical to yield the corresponding alkoxyl radical (C$_{10}$H$_{17}$O$^{\bullet}$) (Atkinson, 2000). Cleavage of the C$_1$−C$_2$ decalin bond, followed by reaction with a second O$_2$ molecule and HO$_2$ leads to a terminal carbonyl hydroperoxide (C$_{10}$H$_{18}$O$_3$) (Yee et al., 2013)."*

17. Line 298/Figure 3: To guide the reader it would be beneficial to update the mechanism in Figure 3 with the same label of tCARBROOH next to intermediate the authors are referring to

We have now added the formula of the different primary products.

18. Lines 298/Figure 3: Do the authors see evidence of an analogous product in the case of decalin, as the OS-279 that was observed in the dodecane case? The proposed mechanism of carbonyl hydroperoxide heterogenous reactive uptake followed by OS formation should also be considered for the decalin tCARBROOH as well and supported by the measurements/compared on the basis of volatility differences due to carbon number/ring structure and the impact of reactive uptake versus partitioning to the particle phase.

We thank the reviewer for its comment. We have revised the pathways proposed for decalin oxidation products OS-265; -267; -269 and -285 and cyclodecane oxidation products OS-249; -251; -265 and -267 to include reactive uptake of the hydroperoxide on wet acidic aerosols.

The appropriate mechanisms have been updated as well as the manuscript:

Lines 314-316: *"OS-265 (C$_{10}$H$_{17}$O$_6$S$^-$) could also arise from the acid-catalyzed perhydrolysis of the hydroperoxide (C$_{10}$H$_{18}$O$_4$) generated from the reaction of C$_{10}$H$_{17}$O$_4^{\bullet}$ + HO$_2$ (Figure S8, pathway b)."*

Lines 319-325: *"The pathway proposed in Figure S8 pathway **b** is based on gas-phase oxidation of a 4-(cyclohexan-2-one)but-1-yl radical followed by reaction with $O_2$ and a 1,5-H shift (Crounse et al., 2011; Orlando and Tyndall, 2012) and lead to a $C_{10}$-carbonyl-hydroxyhydroperoxide ($C_{10}H_{18}O_4$). $C_{10}H_{18}O_4$ could then further react with OH radical and by elimination of OH lead to an epoxide (Figure S8, pathway **b**). In addition, OS-281 could arise from acid-catalyzed perhydrolysis of $C_{10}$-carbonyl dihydroperoxides ($C_{10}H_{18}O_5$) as proposed in Figure S8, pathway **c**."*

Lines 357-374: *"$C_9H_{17}O_4{}^{•}$ can react via pathway **a** (Figure 3) through a 1,6-H shift (Crounse et al., 2011; Orlando and Tyndall, 2012) followed by elimination of OH resulting in a formation of an epoxide analogous to the formation of isoprene epoxydiol (IEPOX) (Paulot et al., 2009; Mael et al., 2015). The epoxide can then undergo acid-catalyzed ring opening to give OS-269 ($C_9H_{17}O_7S^-$). The $MS^2$ spectrum of OS-285 ($C_9H_{17}O_8S^-$; Figure S5) shows product ions corresponding to $HSO_3^-$, $HSO_4^-$ and loss of neutral $SO_3$, in accord with a sulfate ester β to a labile proton, but yields no further structural information. The structure proposed for OS-285 is based on the formation of reaction of the hydroperoxyperoxyl radical intermediate in pathway **b** with $RO_2$ followed by a 1,4-H shift (Rissanen et al., 2015) and addition of $O_2$ to give a hydroxyhydroperoxyperoxyl radical ($C_9H_{17}O_5{}^{•}$). $C_9H_{17}O_5{}^{•}$ could then lead to an epoxide by isomerization (Iinuma et al., 2009; Surratt et al., 2010; Jacobs et al., 2013; Mael et al., 2015) and form OS-285. $C_9H_{17}O_5{}^{•}$ could also react with $HO_2$ and form the corresponding $C_9$-hydroxydihydroperoxide ($C_9H_{18}O_5$), which could then undergo heterogeneous reaction and lead to OS-269 (Figure 3, pathway **b**). Finally, a $C_9$-carbonyl hydroperoxide ($C_9H_{16}O_3$) could also be formed from the $RO + O_2$ reaction (Figure 3, pathway **c**), which could then further react with OH radicals and lead to a $C_9$-carbonyl dihydroperoxide ($C_9H_{16}O_5$). Hence, $C_9H_{16}O_5$ could form OS-267 ($C_9H_{15}O_7S^-$) from heterogeneous reaction on acidic aerosols."*

19. Lines 299-301: (e.g. remote areas). Please also refer to and reference (Peeters et al., 2009; Crounse et al., 2011; Orlando and Tyndall, 2012). The authors need to justify why the basis of their proposed mechanisms for OS formation rely on this pathway when (e.g. Figures 3, 4, and S8). Is it RO2 + RO2, RO2 + HO2, RO2 + NO, RO2 isomerization?
The suggested references have been added to the revised manuscript.

As discussed previously, different regime of $RO_2$ radicals could exist, either terminal ($RO_2$ + $HO_2$; $RO_2$ + $RO_2$; $RO_2$ + $NO$) or autooxidation reactions. In this study, we do not claim to propose all chemical pathways from the oxidation of the alkanes are examined. In most of the mechanisms we have considered the different potential $RO_2$ reactions ($RO_2$ + $RO_2$; $RO_2$ + $HO_2$; $RO_2$ + $RO_2$, and $RO_2$ autooxidation), which could lead to the identified OSs through multiphase chemistry of the products shown in the tentatively proposed mechanisms. RO radicals might have formed for other minor chemical channels, such as $ROOH$ + $h\nu$ or $RO_2$ + $NO$, which were not initially included in the manuscript. It is important to note that these potential reactions, which are now included in the manuscript, do not change the different mechanisms tentatively proposed in this study. In addition we have proposed reaction sequences based on known/reported reactions that will lead to products consistent with the mass spectrometric data. This is the same approach used by other investigators, such as Yee et al. (2013).

20. Lines 301-305: The authors should address the extent of photolysis reactions affecting the fate of the proposed hydroperoxides and aldehydes in the system.

    Potential photolysis reactions are now discussed in the revised manuscript. We have incorporated the potential photolysis of hydroperoxides leading to RO radicals and also the photolysis of the aldehyde proposed in Figure 3, which could lead to the $RO_2$ radical ($C_{10}H_{17}O_5^{\bullet}$).

    Lines 353-357: *"The aldehydic intermediate in the sequence following $C_1$-$C_2$ ring scission may be oxidized to the corresponding acyl radical either by photolysis (Wang et al., 2006) or by H-abstraction (Kwok and Atkinson 1995) followed by addition of $O_2$, reaction with $RO_2$ or $HO_2$ and decarboxylation of the resulting acyl-oxy radical (R(O)O) (Chacon-Madrid et al., 2013) to a hydroperoxyperoxy radical ($C_9H_{17}O_4^{\bullet}$)."*

21. Line 306: "previously unreported" is unclear. Do the authors mean previously unreported in ambient data or previously unreported from similar experiments?

    OS-267 has been identified in previous smog chamber experiments. Sentence has been modified in the revised manuscript.

    Lines 370-374: *"Finally, a $C_9$-carbonyl hydroperoxide ($C_9H_{16}O_3$) could also be formed from the RO + $O_2$ reaction (Figure 3, pathway **c**), which could then further react with OH*

*radicals and lead to a $C_9$-carbonyl dihydroperoxide ($C_9H_{16}O_5$). Hence, $C_9H_{16}O_5$ could form OS-267 ($C_9H_{15}O_7S^-$) from heterogeneous reaction on acidic aerosols."*

22. Line 311: As worded, OS-267, is proposed to originate from further oxidation of OS-269, but the arrows drawn in Figure 3 are inconsistent suggesting origin from the epoxide.
    We have corrected the revised manuscript.

23. Line 327/Figure 4: The description of Ring cleavage of the C10 alkoxy radical is not consistent with the "ISO"/isomerization descriptor in Figure 4. Please clarify that pathway.
    ISO descriptor in Figure 4 (*pathway a*) indicates the isomerization of the $RO_2$ formed from the ring cleavage and lead to hydroperoxide functional group.

    Sentences have been added to better discuss this pathway:
    Lines 382-388: *"The salient features of pathway a include oxidation of the $RO_2$ to 2-decalinone, formation of a $C_{10}$ alkoxy radical followed by ring cleavage of the $C_9{-}C_{10}$ decalin bond and further $RO_2$ isomerization (1,8-H shift) leading to a 4-(carboxy cyclohexyl)-1-hydroperoxybut-2-yl radical via $RO_2$ chemistry. Although considered as a minor reaction pathway (Crounse et al., 2013), the acyloxy radical could lead to the epoxide from the isomerization of the $O_2$ adduct (Paulot et al., 2009; Yao et al., 2014; Zhang et al., 2015). Further acid-catalyzed ring opening of the epoxide leads to OS-295 ($C_{10}H_{15}O_8S^-$)."*

24. Line 340: Figure 1d does not exist. Clarify the reference.
    The reference was Figure 2d and not 1d, the text has been appropriately modified.

25. Lines 343-345: Sentence is awkward beginning with "Pathway c", and from what figure? Clarify that it is Figure 4. Again, citation of Atkinson, 2000 seems inappropriate as the sentence is written.
    Sentence has been changed, pathway **c** referred to Figure 4. Citation of Atkinson (2000) was used to support formation of an organonitrate from $RO_2$ + NO reaction, since it is a common reaction as discussed in the Atkinson's review.

Lines 402-408: *"Although $RO_2$ + NO reactions are expected to be minor under the conditions used in this work (i.e. NO < 1 ppb, formation of RO radicals or organonitrates cannot be ruled out. Indeed, Ehn et al. (2014) have demonstrated that NO reactions could be competitive at ppb levels. Under our experimental conditions $RO_2$ + NO, $RO_2$ + $HO_2$ and $RO_2$ autoxidation are possible. Therefore, the parent ion at m/z 326 could arise from the reaction of the decalin-2-peroxy radical with NO to form decalin-2-nitrate ($C_{10}H_{17}NO_3$) with subsequent reactions shown in Figure 4, pathway c"*

26. Lines 343-345/Figure 4, pathway c: Here the authors propose that RO2 + NO chemistry is occurring to form a nitrate containing OS. This contradicts the authors' earlier statement in lines 237-243 stating that the RO2 + NO reactions are not significant in their experimental setup. The authors need to handle in more detail the fate of RO2 under the unclear experimental conditions.

    As demonstrated and discussed by Ehn et al. (2014) at ppb levels of NO (which is even higher than the conditions of our study) a competition exists between $RO_2$ + NO; $RO_2$ + $HO_2$ and $RO_2$ autoxidation reactions. The Ehn et al. (2014) study demonstrates that ELVOC, even though reduced, are still formed at NO concentrations greater than few ppb. It is important to point out that the concentrations of VOCs used in this work could also lead to $RO_2$ + $RO_2$ chemistry. Therefore, not only one $RO_2$ reaction could occur and the different $RO_2$ reactions have to be considered, which has been done in this study.

    $O_3$ and NO concentrations are provided in the revised manuscript (Table 1). In addition a paragraph has been added describing the fate of $RO_2$.

    Lines 404-406: *"Indeed, Ehn et al. (2014) have demonstrated that NO reactions could be competitive at ppb levels. Under our experimental conditions $RO_2$ + NO, $RO_2$ + $HO_2$/$RO_2$ and $RO_2$ autoxidation are possible."*

27. Lines 349-364: Why is discussion of OS-281 and OS-297 featured here, when discussion of OS-265 is discussed near the beginning of Section 3.2? Since they are all referenced in Figure S8, their chemistry should be discussed together from the same mechanistic precursors.
    We chose to describe the formation and tentative structural assignments of ions observed on ambient filters at the beginning of the discussion, as explained in the manuscript on

lines 293 to 295 ("*Figures 2 and S2 present MS² spectra and fragmentation schemes of selected parent ions at m/z 265.0749 (OS-265), 269.0696 (OS-269), 295.0494 (OS-295) and 326.0554 (OS-326). MS² spectra and fragmentation schemes of other OSs are reported in Figure S3-S7. The selected OSs were, as described in the next section, quantified and characterized in the fine urban aerosol samples.*").

However, section 3.2 has been reorganized as requested by the reviewer #3 and formation pathways of OS-265, -281 and -297 are included in the same paragraph.

28. Section 3.2: Authors should clarify the main mechanistic differences and relative importance between Figure S8, Figure 3, and Figure 4, and the flow of products to be discussed at the beginning of Section 3.2. Currently as written, the flow of Section 3.2 is very arbitrary when choosing different OS products to discuss.

    We decided to separate the different reaction pathways for clarity since it would not have been clear and quite difficult/confusing to propose in one figure the formation pathways of all OSs. The different reaction pathways are separated based on OSs that are generated from branching reactions of a common transient. This section has been modified as discussed in the previous point. The importance of the proposed pathways cannot be evaluated based on this study and this was not the study objective.

    We have added a sentence to clarify this point.
    Lines 297-299: "*The different reaction pathways presented below, are separated based on OSs that are generated from branching reactions of a common transient.*"

29. Lines 377-380: Incorrect use of citations here. Yee et al., 2012 do not propose RO2 + RO2 chemistry and therefore a "precedent" has not been established. The authors should not be citing Atkinson, 2000; Yee et al., 2012 and Raff and Finlayson-Pitts, 2010 to speak for the experimental conditions in the current work. The mechanism of RO2 + RO2→RO can be supported by work on general atmospheric chemistry mechanisms including Atkinson, 2000 and many other works, and so if this mechanistic pathway is to be cited, than many other works should be cited as well.

    We agree with reviewer that Yee et al. is an inappropriate reference and have removed it. General references on atmospheric chemistry have been added: Atkinson and Arey (2003,

Chem Rev, 103, 4605-4638) and Ziemann and Atkinson (2012, Chem Soc. Rev., 41, 6582-6605).

30. Line 379: Inconsistent citation here compared to line 300 for similar mechanistic argument.

We have cited Ehn et al., 2014 and Jokinen et al., 2014 and Mentel et al., 2015 for both transformations.

31. Figure S14: Why do the authors propose in the case of cyclodecane formation of the hydroperoxide from RO2 + HO2 pathways and subsequent chemistry thereof, but not in the case of decalin in any of Figures 3, 4, and S8? Further, Figure S14 outlines a mechanism from further reaction of the hydroperoxide to get to an epoxide that then enters the particle phase to produce OS-251 and OS-249. This seems like a plausible analogous mechanism to propose for the case of dodecane rather than reactive uptake of a carbonyl hydroperoxide. Why do the authors propose different mechanisms between dodecane and cyclodecane to generate the similar analogs (OS-279, OS-249)?

As mentioned in the point #18 above, cyclodecane-derived OSs might be formed from the heterogeneous chemistry of hydroperoxides. We have added these different pathways in the different Figures as well as in the manuscript. Please note that we have now combined Figures S14 and S15 to present the OS formation from cyclodecane in one Figure.

Lines 428-435: *"The formation of compounds such as cyclodecanone ($C_{10}H_{18}O$), cyclodecane hydroperoxide ($C_{10}H_{20}O_2$) or cyclodecane hydroxyhydroperoxide ($C_{10}H_{20}O_3$) are proposed as intermediate products leading to epoxy-compounds after additional oxidation/isomerization processes, as presented in Figure S14. In addition $C_{10}H_{20}O_3$, cyclodecane hydroperoxide ketone ($C_{10}H_{18}O_3$) and cyclodecane hydroxyoxohydroperoxide ($C_{10}H_{18}O_4$), proposed as intermediate products, could condense onto wet acidic aerosols and lead to the corresponding OSs through acid-catalyzed perhydrolysis reactions (Figure S14)."*

As described above we proposed different fates for the $RO_2$ radicals: $RO_2 + RO_2$, $RO_2 + HO_2$ and $RO_2$ autoxidation reactions in the different schemes proposed for the photooxidation of decalin and cyclodecane. $RO_2 + HO_2$ reactions are also proposed in the

case of the photooxidation of decalin, which likely explain the formations of hydroperoxides as discussed above.

32. Section 3.4: This section is weak and little effort is made to really describe the chemical differences between the systems to interpret the findings. There should be comparisons of vapor pressures of the precursors and carbon numbers and discussion of previously published yields from these compounds to support the discussion. How does quantification using the available OS surrogate standards potentially affect the OS quantification across these systems/factoring in different sensitivities?

The objective of this study is to establish that OSs may be products of the photooxidation of anthropogenic precursors, such as the alkanes examined here, and thus to demonstrate the relevance of this chemistry observations of aliphatic OSs in urban areas (Mao et al., 2014; Tao et al., 2014). Since the reaction pathways leading to the products observed in this study and in ambient samples are tentative, we feel that discussion at the level of thermodynamics is not justified and have deleted Section 3.4.

33. Line 400: The authors claim that "the presence of acidic aerosols significantly increase OS formation in most cases". However, is this just an effect of using an atomized solution with more sulfate (0.06M ammonium sulfate + 0.06M sulfuric acid) in the acidic case versus only 0.06M ammonium sulfate in the non-acidified case? It may be a concerted effect of more available sulfate in the "acidic" case as well as acidity.

It has been demonstrated in previous studies (cited references) that acidity rather than the concentration of sulfate is a key parameter in the formation of OS. Chan et al. (2011) demonstrate that the formation of OSs from the oxidation of $\beta$-caryophyllene is directly correlated with the aerosol acidity ([H$^+$]).

34. Line 484: The authors return to claim that the experiments are conducted under dominant "RO2/HO2" chemistry—this is contradictory to the formation of OS-326 containing a nitrate group.

This issue has been discussed previously in response to reviewer comments # 10, 13, 19, 26, and 31 above.

35. Lines 484-486: Enhancement of OS due to acidified ammonium sulfate seed needs to be addressed with regard to the effect of just having introduced more sulfate into the experiments compared to the non-acidified case. See earlier Major Comment, 33.

As discussed in response to Reviewer comment # 33, enhancement of OSs has been demonstrated to result from an increase of the aerosol acidity (cited references) and our work is consistent with these studies.

36. Lines 491-496: The "novel pathway" involving reactive uptake of hydroperoxides is not well- substantiated in the current work and is mostly speculation. The vapor pressure alone of the carbonyl hydroperoxide makes it a potential candidate to partition to the particle phase, not via reactive uptake. There are no direct measurements of hydroperoxides in the gas phase, and insufficient discussion on if hydroperoxides are detected in the particle phase using the UPLC technique. Further, if reactive uptake is at play, why have the authors not seen the corresponding decalin analog of carbonyl hydroperoxide?

We agree with reviewer #3 that "reactive uptake" of hydroperoxides is currently not well substantiated, and have clearly indicated that this pathway is tentative. With regard to reactive uptake and perhydrolysis of carbonyl hydroperoxides generally as an alternative pathway the revised manuscript cites this route as a possibility in the formation of 8 OSs, as indicated in response to earlier comments. Also as discussed above in response to Reviewer comment #14, LC-EI/MS in the negative ion mode, used to identify the OS products, will not detect analytes (such as hydroperoxides) not containing substituents readily yielding negative ions.

**Minor Comments:**

1. Line 30: "Both studies strongly support formation of OSs" is awkward. Reword, for example, "Both studies strongly support that OSs can form from the gas-phase oxidation of anthropogenic precursors..."

The wording has been revised on lines 30-33 as follows:

*"Both studies strongly support the formation of OSs from the gas-phase oxidation of anthropogenic precursors, as hypothesized on the basis of recent field studies in which aromatic and aliphatic OSs were detected in fine aerosol collected from several major urban locations."*

2. Line 48: Change "aerosol" to "particles", as aerosol is technically defined as both gas + particle.

   The wording has been revised as suggested.

3. Line 76: Insert comma after "2015)".

   A comma has been inserted.

4. Line 83: Change comma to semi-colon after "2007,".

   A semi-colon has been inserted.

5. Line 90: Delete "of".

   Use of "of" is appropriate and we have not made this change.

6. Line 103: Change "reduce" to "reduces".

   The revision has been made as suggested.

7. Line 136: Check misprint on the high humidity range listed as "(4-60%)".

   The correction has been made (i.e. 40-60%).

8. Line 220: Insert after ")", ", hereafter referred to as OS-279, ".

   The change has been made as suggested.

9. Line 268: "ion at m/z 265.0749" should be "ion at m/z 265.0752" according to Figure 2.

   "at" has been inserted.

10. Line 315: Add in "Figure 3, pathway a" to be clear.

    "Figure 3, pathway a" has been inserted.

11. Line 345: Change "identical" to "analogous" as the sequence of reactions are certainly not identical as shown in Figure 4.

    "Analogous" has been substituted.

12. Line 350: Change chemical formula to include S for OS-281.

    The formula has been corrected.

13. Line 352: Rewrite the sentence. The radical reacts with O2, followed by 1,6 H shift, etc.

    Sentence has been modified on lines 319-325 to:

    *"The pathway proposed in Figure S8 pathway **b** is based on gas-phase oxidation of a 4-(cyclohexan-2-one)but-1-yl radical followed by reaction with $O_2$ and a 1,5-H shift (Crounse et al., 2011; Orlando and Tyndall, 2012) and lead to a $C_{10}$-carbonyl-hydroxyhydroperoxide ($C_{10}H_{18}O_4$). $C_{10}H_{18}O_4$ could then further react with OH radical and by elimination of OH lead to an epoxide (Figure S8, pathway **b**). In addition, OS-281 could arise from acid-catalyzed perhydrolysis of $C_{10}$-carbonyl dihydroperoxides ($C_{10}H_{18}O_5$) as proposed in Figure S8, pathway **c**."*

14. Lines 362-363: Rewrite the awkward phrasing, "which be reactively taken up to give a sulfate ester".

    Sentence has been changed.

    Lines 331-334: *"However, in contrast to pathway **b**, $RO_2$ formed by the addition of $O_2$ undergoes a 1,6-H shift (Crounse et al., 2011; Orlando and Tyndall, 2012) followed by addition of a second $O_2$ molecule, a 1,5-H shift and elimination of OH to yield an epoxide, which leads to a sulfate ester by reactive uptake onto acidified aerosols."*

15. Lines 371-374: Poor grammar. Rewrite sentence.

    The sentence has been changed on lines 419-422 to:

    *"None of the fragment ions observed in the $MS^2$ spectrum suggests the presence of a terminal carbonyl or a carboxyl functional group in the cyclodecane-OSs, which is consistent with conservation of the cyclodecane ring."*

16. Line 379: "hydroperoxydes" is spelled wrong.

    The spelling has been corrected.

17. Line 459: Add "of" after "oxidation".

    "of" has been added.

18. Line 482: Add "," after "cyclododecane".

    The comma has been added.

**Additional References:**

Atkinson, R.; Arey, J.; Aschmann, S. M. Atmospheric Chemistry of Alkanes: Review and Recent Developments. Atmos. Environ. 2008, 42 (23), 5859–5871.

Crounse, J. D.; Paulot, F.; Kjaergaard, H. G.; Wennberg, P. O. Peroxy Radical Isomerization in the Oxidation of Isoprene. Phys. Chem. Chem. Phys. 2011, 13 (30), 13607–13613.

Dibble, T. S. Cyclization of 1,4-Hydroxycarbonyls Is Not a Homogenous Gas Phase Process. Chem. Phys. Lett. 2007, 447 (1-3), 5–9.

Lambe, A. T.; Onasch, T. B.; Croasdale, D. R.; Wright, J. P.; Martin, A. T.; Franklin, J. P.; Massoli, P.; Kroll, J. H.; Canagaratna, M. R.; Brune, W. H.; et al. Transitions from Functionalization to Fragmentation Reactions of Laboratory Secondary Organic Aerosol (SOA) Generated from the OH Oxidation of Alkane Precursors. Environ. Sci. Technol. 2012, 46 (10), 5430–5437.

Lim, Y. Bin; Ziemann, P. J. Chemistry of Secondary Organic Aerosol Formation from OH Radical-Initiated Reactions of Linear, Branched, and Cyclic Alkanes in the Presence of NOx. Aerosol Sci. Technol. 2009a, 43, 604–619.

Lim, Y. Bin; Ziemann, P. J. Kinetics of the Heterogeneous Conversion of 1,4-Hydroxycarbonyls to Cyclic Hemiacetals and Dihydrofurans on Organic Aerosol Particles. Phys. Chem. Chem. Phys. 2009b, 11 (36), 8029–8039.

Orlando, J. J.; Tyndall, G. S. Laboratory Studies of Organic Peroxy Radical Chemistry: An Overview with Emphasis on Recent Issues of Atmospheric Significance. Chem. Soc. Rev. 2012, 41 (19), 6294.

Peeters, J.; Nguyen, T. L.; Vereecken, L. HO X Radical Regeneration in the Oxidation of Isoprene W. Phys. Chem. Chem. Phys. 2009, 11, 5935–5939.

Robinson, A. L.; Donahue, N. M.; Shrivastava, M. K.; Weitkamp, E. a; Sage, A. M.; Grieshop, A. P.; Lane, T. E.; Pierce, J. R.; Pandis, S. N. Rethinking Organic Aerosols: Semivolatile Emissions and Photochemical Aging. Science 2007, 315 (5816), 1259–1262.

---

## Referee Report (RR1)

Response to Anonymous Referee #3

We thank Referee # 3 for the comments and address each below. Our responses are denoted in blue texts.

Review of "Characterization of Organosulfates in Secondary Organic Aerosol Derived from the Photooxidation of Long-Chain Alkanes"

Reviewer's Summary:

The authors characterize organosulfates (OSs) from the laboratory oxidation of dodecane, decalin, and cyclodecane under varying conditions of humidity and two different seed types (non-acidified, acidified). They observe overlapping organosulfates in the laboratory experiments and on filters from Pasadena, USA and Lahore, Pakistan, concluding that OSs from the oxidation of anthropogenic precursors may contribute to urban SOA. The results are novel and would be of interest to the readers of ACP; however, I would not recommend this manuscript for publication because it is not well-written and the conclusions are highly speculative. In particular, the proposed chemical mechanisms from the laboratory experiments are not substantiated by a fundamental knowledge of the chemistry occurring in the reaction chambers used. The authors inconsistently address the fate of the $RO_2$ radical within their laboratory experiments throughout the text and within the proposed mechanisms. There seems to be a mix of $RO_2$ reacting with $RO_2$, $HO_2$, and NO, though they claim different regimes depending on what mechanism they are proposing to explain the OSs formed. For example, they state that reaction with NO is insignificant, yet they report a nitrate containing OS in the decalin system.

First, with regard to $RO_2$ chemistry: as demonstrated by Ehn et al. (2014), at ppb levels of NO

(1-5 ppb; NO concentration in our study, $< 1$ ppb based on NO measurement) competition exists between $RO_2 + NO$, $RO_2 + HO_2$ and $RO_2$ autoxidation reactions. Nevertheless, ELVOC, though reduced, still formed, indicating that auto-oxidation does not occur solely under pristine conditions. It is important to point out that the high concentrations of VOCs used in this work favor involvement of $RO_2 + RO_2$ chemistry. In addition, previous work (Crounse et al., 2013), has also reported different $RO_2$ regimes, such as autooxidation or $RO_2 + HO_2$ reactions, in experiments using methyl nitrite as an OH radical source, similar to isopropyl nitrite used in our study. As discussed below in response to reviewer's comment # 10, $RO_2 +$ NO reactions are minimal; however, the nitrated OSs at *m/z* 326 are also measured in low concentrations (ng/m$^3$, Table S3).

Thank you for clarifying the conditions of the experiment. It is still interesting that while the OSs do not make up the majority of the OA produced, the OS-326 can make up anywhere from 3% to 14% of the total OS mass though.

Second: we do not claim that the proposed mechanisms represent the major reaction pathways of the photooxidation of the studied alkanes, but are tentatively proposed to explain the formation of the OSs identified in this study. Mechanisms have been clearly indicated as *proposed* branching of pathways of the alkane photooxidations presented to explain formation of specific OSs products consistent with MS$^2$ data. This approach to rationalizing OH oxidation products is universally applied in oxidation studies (Yee et al 2013; Bugler et al., 2015). Furthermore, the concentrations reported in Tables S1-S3 emphasize the fact that identified compounds are in low abundance regardless of the mass of SOA measured in all experiments.

It is certainly not expected that a fully exhaustive list of mechanistic pathways are presented for each studied alkane, but the proposed pathways to OS formation should be self-consistent with the conditions of the experiment and between precursors. Stated more broadly, the presence of analog OSs between two HC systems should not be proposed to form under different mechanistic pathways since the authors state similar chemical conditions for the systems studied. Further, the absence of an analog OS in one system and presence in another system should also be considered when proposing a mechanism that would likely happen in both systems. With the fate of the RO2 radical now clearly communicated and consideration of more analogous pathways between the precursor systems, the proposed pathways have more credence.

Finally: we stress to Reviewer #3 that neither the analytical work nor the interpretation of the MS$^2$ data were questioned. The tandem MS data are consistent with the structures proposed for products observed in both the lab-generated and ambient aerosol samples and we repeat that we clearly indicated in the text that the mechanisms presented are suggested as pathways leading to ions consistent with those observed and until authentic standards become available both the product structures and mechanisms of formation remain tentative. An additional and crucial comment we make is that the major objective of our study is to demonstrate that aliphatic organosulfates form via alkane oxidation, and offer one explanation for reports of aliphatic OSs in urban areas (Ma et al., 2014; Tao et al., 2014).

They propose the formation of hydroperoxides in the case of dodecane experiments with high initial precursor concentrations and do not propose RO2 + RO2 chemistry, but for the C10 systems RO2 + RO2 reactions are proposed with some RO2 + HO2 reactions. They propose epoxide precursors in the C10 systems to OS formation, but not in the C12 system. In general, the proposed mechanisms are arbitrary and do not demonstrate careful control in the design of the experiments or understanding of the chemistry. This lack of understanding becomes clear because there are several areas where citations are used to support the current work, but the citations are used imprecisely and out of context. The manuscript would benefit from more clearly stated organizational structure (e.g. why some mechanisms are proposed in the main text versus the supplemental information).

First we would like to point out that pathways have been proposed that lead to structures consistent with the mass spectrometric data acquired for the observed OSs. We have not attempted to hypothesize general mechanism that would be predicted to give the entire array of precursors contributing to the total mass of SOA. Regarding the possibility of an epoxide precursor to OS-279, we considered the possibility of formation of OS-279 from the reactive uptake of the corresponding epoxide (C$_{12}$H$_{24}$O); however, the composition of OS-279 (1 DBE) is inconsistent with reactive uptake of an epoxide. Therefore, we have used the few studies available in the literature (Yee et al., 2012; 2013) to propose the formation of OS-279 from the heterogeneous chemistry of hydroperoxides. Finally, we have considered the potential heterogeneous chemistry of hydroperoxides formed from the photooxidation of decalin and cyclodecane as discussed above.

This is an interesting chemical feature that is different between the proposed mechanism for OS formation in dodecane compared to the C10 systems, not included in textual description on OS-

279 proposed pathway. This point also is lost in general textual descriptions of epoxide route to OS from these HCs. The authors might consider highlighting this finding as a nuance between the systems and possibility of enhanced epoxide formation from C10 cyclic systems and subsequent OS formation. This fits better in the context of atmospheric relevance and motivation (confusing in the first version) as the authors cite the potential for SOA formation from C10 cyclic alkanes to be greater than linear or branched C12 alkanes.

The authors should also clarify motivation in the experimental selection of two C10 cyclic alkane structures and one C12 straight chain structure.

These compounds have been selected due to their potential contribution to SOA formation in the atmosphere. Recent studies have investigated the SOA formation from decalin and dodecane oxidations and reported large SOA yields (Yee et al., 2013; Hunter et al., 2014). Moreover, Pye and Pauliot (2012) have shown that, even though less emitted into the atmosphere, the cyclic $C_{10}$ alkanes have a greater potential for SOA formation than linear or branched alkanes $< C_{12}$.

Check all misspellings on reference to Pauliot. Should be Pouliot.

A few sentences have been added to better explain our selection of parent VOCs on lines 118-122:

*"These alkanes were selected based on their potential contribution to atmospheric SOA formation (Hunter et al., 2014). Studies have demonstrated that cyclic compounds ($< C_{12}$) are expected to be more efficient SOA precursors than linear or branched alkanes with the same number of carbons (Lim and Ziemann, 2005; Pye and Pauliot, 2012). Alkanes $\geq C_{10}$ are considered as effective SOA precursors, especially when placed in the context of their emission rates (Pye and Pauliot, 2012)."*

The brevity of the discussion of results on the OSs from dodecane photooxidation are quite brief relative to the other sections interpreting the results from decalin and cylodecane, and the effects of chemical structure are glossed over in brevity.

As mentioned in lines 229-231, low abundances of OS-209 and OS-237 precluded acquisition of high-resolution $MS^2$ data, and thus, structures have not been proposed for the parent ions. Without structural information, discussion of formation pathways is not possible. The criticism of our conclusions as "highly speculative," is not consistent with request for more detail on the chemical structure of products formed in abundances too low to obtain high-resolution $MS^2$ data.

Thank you for adding this clarification in the revised version.

Further, it is unclear if the conclusion that enhancement of OS yields are due to increased acidity of the seed aerosol is really due to acidity, rather than an effect of seeding the experiments with an atomized solution containing more sulfate. These concerns are outlined in detail below.

This point has been previously discussed and published work demonstrates that acidity, rather than concentration of sulfate, is the key parameter (reference cited in the article) in the formation of OSs. Chan et al. (2011) have reported that the formation of OSs from the oxidation of $\beta$-caryophyllene is directly correlated with the aerosol acidity ([H$^+$]).

Thank you for clarifying this; the authors should consider adding such discussion to the manuscript to clarify the experimental methods and design as in Major Comment 33.

Major Comments:

1. Lines 58-61: These lines are specious in the use of citations and misleading. First, as written, these lines assert that the underestimate of global SOA is equivalent to an underestimate in urban SOA. Second, the references cited (Pye and Pouliot, 2012; Tkacik et al., 2012) do not specifically argue that the underestimate in predicted SOA is due to the omission of IVOCs. A better reference here based on the lines as written would be (Robinson et al., 2007). Pye and Pouliot, 2012 can be cited for exploring additional mechanisms (oligomer formation) from alkane and PAH in SOA formation, and Tkacik et al., 2012 can be cited for providing additional evidence that IVOCs may be a missing source in modeling urban SOA, but the authors need to reword these lines carefully and be more precise.

   Sentence has been modified on lines 59-62 as follows:

   *"The omission of intermediate volatility organic compounds (IVOC) as SOA precursors, such as alkanes or polycyclic aromatic hydrocarbons (PAHs), could contribute in part to the underestimation of SOA mass observed in urban areas (Robinson et al., 2007; Tkacik et al., 2012)."*

2. Line 65: References here should include Yee et al., 2013 which more specifically addresses analogous to Lim and Ziemann, 2005 the products and mechanisms of C12 alkanes of varying structures.

   References have been added.

3. Line 66: Tkacik et al., 2012 should be included here for presenting yields from several alkane systems.

Reference has been added.

4. Line 71: For this discussion on structure and fragmentation, additional reference should be cited (Lambe et al., 2012).

Reference has been added.

5. Line 107: The authors assert that acid-catalyzed reactive uptake has not been reported for the oxidation of alkanes. This is not true. Atkinson, Lim, and Ziemann have shown that alkane oxidation leading to 1,4-hydroxy carbonyls convert to cyclic hemiacetals in an acid-catalyzed multi-phase process (Dibble, 2007; Atkinson et al., 2008; Lim and Ziemann, 2009a, 2009b). Schilling Fahnestock et al., 2015 also report the effect of acidity on SOA formation from C12 alkanes.

This statement has been removed from the revised version of the manuscript.

6. Lines 123-124: Can the authors give more background on these two sites to orient the reader also with the motivation/purpose of this study? What types of sites are these— urban with what type of emissions profiles and surrounded by vegetation, etc.?

As it is highlighted in the abstract, the motivation for our study was to demonstrate the formation of OSs from the oxidation of alkanes, which has been inferred from previous field studies. Description of both sites is presented in the experimental section and references, which have already characterized both sites, are cited.

The following sentences have been added:

Lines 185-188:*"As stipulated previously at both urban sites, anthropogenic activities (e.g., vehicular exhaust, industrial sources, cooking, etc.) likely dominated the organic aerosol mass fraction of PM$_{2.5}$ (Stone et al., 2010; Hayes et al., 2013). In addition, Gentner et al. (2012) have reported significant emission of long-chain alkanes during the CalNex field study."*

7. Line 222-223: It would be helpful to label the 1,3-dodecanone sulfate in Figure 1 to aid the reader. The authors should be careful with their naming convention here (i.e. 1,3-), as

this particular isomer certainly is not the only potential isomer and is not the only isomer specified in Yee et al., 2012 and Yee et al, 2013.

The OS has been labeled in Figure 1. We agree with referee #3 that other isomers are present since we identified at least 3 isomers as mentioned on line 236 and reported in Table S1.

The OS has not been labeled accordingly in Figure 1.  Please make the change so it is easier for reader to follow in-text and figure.

The sentence has been modified on lines 239-242 to*:*

*"Kwok and Atkinson (1995) have reported that OH oxidation of long-chain alkanes preferentially occurred at an internal carbon and thus multiple isomers may be proposed. Based on Yee et al. (2012; 2013) one isomer may be, however, assigned as 6-dodecanone-4-sulfate."*

As currently drawn in Figure 1 for OS-279, the naming should be changed from 6-dodecanone-4-sulfate to 6-dodecanone-8-sulfate, no?  Also, the particular isomer chosen here should just be an example isomer for the purpose of drawing the mechanism.  It is unnecessary and would be incorrect to cite Yee et al., (2012; 2013) for this isomer as that work also does not "assign" a specific isomer or isolate a specific isomer from the measurements.

The sentence should be modified to:

*"Kwok and Atkinson (1995) have reported that OH oxidation of long-chain alkanes preferentially occurred at an internal carbon and thus multiple isomers may be proposed. One such isomer, 6-dodecanone-8-sulfate, is drawn in Figure 1 to represent a proposed structure for OS-279."*

8. Lines 226-230: The reference cited, Ruehl et al., 2013, is improperly used here. Ruehl et al., 2013 describes the heterogenous oxidation of octacosane and finds a strong preference for OH attack at the terminal carbons. The current work, however, is gas-phase oxidation, so the specificity of the isomers as listed in lines 228-229 should be rethought. Further, the naming convention for these isomers are inconsistent with the naming convention in line 222-223. It seems as though the 1, 3-dodecanone denotes the 1 position as the ketone, whereas here the reference to 2, 4-, 3,5-, and 4,6- and other isomers suggests the 1 position is likely the carbon at the end of the dodecane chain.

9. Lines 234-237: This is a poorly worded sentence. It is unclear in relation to the context of the current work, and there is imprecise use of citations. Hydroperoxides can undergo further oxidation by reaction with OH, but to generate alkoxy radicals from hydroperoxides, that would likely include photolysis. The authors need to address the extent of photolysis in the experiments then. Or are the authors referring to reactions of RO2 + NO to generate alkoxy radicals? If so, then the authors need to address the extent of RO2 + NO occurring in the experiments. If the former, rewrite as, "First-generation hydroperoxides can undergo further oxidation by reaction with OH to form low-volatility, more highly oxidized products, or can be photolyzed to alkoxy radicals (RO) to form more highly volatile products." The use of Carasquillo et al., 2014 here is inappropriate to discuss the oxidation of hydroperoxides as written. Carasquillo et al., 2014 describe the fate of differing alkoxy radical structures and how it affects SOA yield. The authors need to clarify what they are trying to say here and how it relates to the mechanism proposed in Figure 1.

We thank the reviewer for its comments, sentences have been changed to simplify and clarify this paragraph.

Insertion should be made to the sentences below as indicated in red.

Lines 249-255: "*First-generation hydroperoxides ($C_{12}H_{26}O_2$) can undergo further oxidation by reaction with OH to form either more highly oxidized products, such as dihydroperoxides ($C_{12}H_{26}O_4$), or semi-volatile products ($C_{12}H_{24}O$) (Yee et al., 2012). In addition, hydroperoxides can be photolyzed to alkoxy radicals (RO) undergoing additional transformation to form more highly oxidized products. Low-volatility products could then condense onto sulfate aerosols and undergo further heterogeneous reactions (Schilling Fahnestock et al., 2015) leading to OSs as discussed below.*"

10. Lines 237-243: The authors can cite Raff and Finlayson-Pitts, 2010 for IPN as an OH radical source, but it cannot be cited to fully account for the chemical conditions (i.e. the fate of $RO_2$) in the current experiments without considering the differences between their experiment and that of Raff and Finlayson-Pitts, 2010. The authors should report NOx levels in these experiments to verify the claim that RO2 + NO reactions are minimal.

Also, how is O3 formed in these experiments? The authors need to calculate (considering the relatively high levels of initial hydrocarbon), the relative fate of RO2 between reaction with $RO_2$, $HO_2$, and NO. The proposal that OS-279 stems from hydroperoxide species in Figure 1 seems least inappropriate if $RO_2$ fate is really dominated by reaction with RO2 and/or NO.

In order to provide additional support for the proposed mechanisms, total organic peroxide measurements have been performed. These results which are now reported in Table 1 reveal that organic peroxides (including hydroperoxides) could contribute up to ~ 28 % (on average) of the SOA mass formed from the photooxidation of the precursors used in this work. These measurements highlight the significant presence of organic peroxides and/or hydroperoxides in aerosol and thus support the proposed mechanisms. In addition to the organic peroxide measurements, concentrations of $O_3$ and NO were also added in Table 1 to confirm the low-NO conditions cited in this work.

The description of the organic peroxide measurements has been added, lines 211-219, revised manuscript:

*"**Total Organic Peroxide Analysis.** The total organic peroxides in the SOA were quantified using an iodometric-spectrophotometric method adapted from Docherty et al. (2005). As described in Surratt et al. (2006), the method employed in this work differs in the choice of extraction solvent: we used a 50:50 (v/v) mixture of methanol and ethyl acetate, rather than pure ethyl acetate. Calibrations and measurements were performed at 470 nm using a Hitachi U-3300 dual beam spectrophotometer. Benzoyl peroxide was used as the standard for quantification of organic peroxides formed from alkane oxidations. The molar absorptivity measured from the calibration curve was ~ 825, which is in excellent agreement with reported values (Docherty et al., 2005; Surratt et al., 2006)."*

A discussion of the results of the organic peroxide measurements has been added, lines 261-265, revised version*:*

*"In addition, total organic peroxide aerosol concentrations, presented in Table 1, reveal that organic peroxides account (on average) for 28 % of the SOA mass measured in the different experiments in support of a significant contribution of $RO_2 + RO_2/HO_2$ and/or $RO_2$ autoxidation to SOA formation from alkane oxidations."*

In our experiment we did not add NO (prior to IPN injection) and background NO levels

were measured near the detection limit of the $NO_x$ monitor (i.e., 1 ppb). After IPN injection a significant increase of $O_3$ was observed in all experiments (as described lines 258-260) and NO concentration dropped below 1 ppb. We agree that $NO_2$ is photolyzed in these systems and NO is recycled. However, under the conditions described here (and in the article on lines 258-260 and in Table 1) most of the NO is expected to react with $O_3$. Rate constants for the $RO_2 + NO$ and $NO + O_3$ reactions were determined to be $4.7 \times 10^{-12}$ $cm^3$ molecule$^{-1}$ s$^{-1}$ ($k_1$) (suggested by MCM; Ehn et al., 2014) and $1.8 \times 10^{-14}$ $cm^3$ molecule$^{-1}$ s$^{-1}$ (Atkinson et al., 2000; IUPAC) ($k_2$), respectively. Gratien et al. (2010, ES&T, 44, 8150-8155) have calculated the OH radical concentration to be $5 \times 10^6$ molecule cm$^{-3}$ from the photolysis of 5 ppm of isopropyl nitrite (IPN). In our experiments, 0.1-0.25 ppm of IPN was injected into the chambers. Using similar OH radical concentration to Gratien et al. (2010, ES&T) as an upper limit, $RO_2$ concentration could be estimated to be at ppt levels. Therefore, under the conditions of our study and assuming an $O_3$ concentration of 0.5 ppm, NO would react predominantly with $O_3$: $(k_2[O_3] [NO])/(k_1[RO_2] [NO]) > 350$.

Lines 258-260 have been modified:

*"Although $RO_2$ radicals could also react with NO formed by either IPN or $NO_2$ photolysis, formation of ozone under chamber conditions (0.3-0.6 ppm, depending on the concentration of IPN injected, Table 1) would rapidly quench NO (Atkinson et al., 2000). Therefore, $RO_2$ + NO reactions are not expected to be significant."*

Finally, the photooxidation of dodecane has been also investigated using an additional injection of NO (200 ppb) prior IPN injection. NO concentration dropped below the ppb level in less than 1 hour and OS concentrations were significantly reduced (factor of 3-4) compared to other experiments, confirming that NO concentration does have an impact on OS formation.

One sentence has been added to describe this experiment, lines 274-277:

*"It should be mentioned that photooxidation of dodecane has also been investigated using an additional injection of NO (200 ppb) prior IPN injection. In this experiment SOA formation was significantly reduced as well as the OS concentrations (factor of 3-4), confirming that NO strongly impacts the formation of OS, such as OS-279."*

These clarifications on the chemical regimes taking place during the experiments greatly

enhance the quality of the manuscript and better substantiate the mechanisms proposed and conclusions reached. Thank you.

11. Table 2: The authors never describe the origin of the C7 (OS-209) and C9 (OS-237) organosulfates observed in the dodecane system and also observed in the ambient samples. This is another indication that fragmentation pathways are at play, potentially through RO2 + NO reactions in the system. The authors need to be careful in explaining the fate of the RO2 radical in their experiments and whether the ambient observation of these OSs can really be attributed to dodecane chemistry in the atmosphere when they may clearly originate from other precursors. The authors need to also describe the potential influence of monoterpenes at the sites they have taken samples from to preclude OS origin from biogenic precursors, as they say themselves that C10 monoterpene OSs are isomeric to some proposed in the C10 alkane systems. How good is RT matching/SICs for confirming that the laboratory generated OSs are really the same as those in ambient data? What measurements in these locations suggest that decalin, cylodecane, and dodecane are prevalent here?

As mentioned in lines 229-231, low abundances of OS-209 and OS-237 precluded acquisition of high-resolution MS$^2$ data, and thus structures have not been proposed for these parent ions. Without any compositional information, reaction pathways cannot be discussed. As shown by Yee et al. (2012), hydroperoxides can be photolyzed to RO radicals, which fragment to smaller carbonyls. The potential formation of RO radicals from photolysis of hydroperoxides has been added in the revised version of the manuscript.

Isobaric compounds could likely be formed in the atmosphere, however, structures would be significantly different and isomers could be distinguished in most of the cases. Although we cannot completely rule out co-elution of some isobars, Figure 7 illustrates the most likely typical situation, in which isobars from the photooxidation of cyclodecane and decalin have different retention times (R.T.), allowing differentiation. OSs are known to form from the oxidation of monoterpenes and several isobaric OSs have been identified: OS-249, -251, -267, -279 and -326. Structures proposed in previous work are significantly different from structures proposed in this work and thus should be separated by liquid chromatography. We have analyzed ambient filters collected during SOAS campaign in rural areas (Centerville, Alabama, US) and find that the R.T.s of

monoterpene-derived OSs are different from those of the OSs identified from the oxidation of the alkanes studied in this work.

For example:

   OS-249: from monoterpenes: 10.3 min; cylodecane: 8.5/9.3

   OS-279: from monoterpenes: 6.2 min; cylodecane: 5.8/6.8

Thank you for this clarification.

We do not have access to potential collocated measurements during both field measurements, however, results proposed by Gentner et al (2012) tend to support significant emissions of long-chain alkanes in California and especially during CalNex.

The following sentences have been
added: Lines 185-188:

*"As stipulated previously at both urban sites, anthropogenic activities (e.g., vehicular exhaust, industrial sources, cooking, etc.) likely dominated the organic aerosol mass fraction of $PM_{2.5}$ (Stone et al., 2010; Hayes et al., 2013). In addition, Gentner et al. (2012) have reported significant emission of long-chain alkanes during the CalNex field study."*

11. Lines 246-247: The citation of Claeys et al., 2004 is inappropriate here. The authors propose that "heterogenous chemistry of gas-phase organic peroxide" is a mechanism for OS and tetrol formation, citing Claeys et al., 2004. Yet, Claeys et al., 2004 state, "The mechanism we suggest, reaction with hydrogen peroxide under acidic conditions in the aerosol liquid phase...," which is not consistent with the heterogeneous mechanism proposed in the current work and in Riva et al., 2015b. The difference in humidity should also affect the distribution of hydroperoxide compounds in the gas/particle phase. The authors should address this in the context of mechanistic explanations for their observations.

We agree with reviewer that Claeys et al. is an inappropriate reference to be used here and we have removed it. We have shown in previous work and in another manuscript currently under review that organic peroxides could lead to OSs and polyols from aerosol-phase acid-catalyzed reactions. It is not clear how the RH could directly impact the distribution of the hydroperoxides as suggested by the reviewer. However, we have reported that the liquid water content of the aerosol plays an important role, but the acidity

has a stronger impact on OS formations.
Thank you for clarifying.

12. Lines 256: Several citations should be added here. Include reference to works by Lim and Ziemann, Lambe et al., Yee et al., Loza et al., Tkacik et al.
References have been added.

13. Lines 284-286: The mechanism described in text corresponds with the pathway in Figure S8, pathway c, not pathway a. Authors should rewrite these lines to describe pathway a. It also becomes clear here that the authors are not consistent with description of the chemistry proceeding in the chamber. In Figure S8, the fate of RO2 is initially reaction with RO2, but then in pathway a, it shifts to RO2+HO2. The selected pathways seem arbitrary to explain the proposed structure in Figure 2a.

Figure S8 describes sequential reactions via $RO_2$ leading to a ring opened ketoperoxy transient. Three branching reactions are available to transient: reaction with $HO_2$ leading to the structure proposed for OS-265, pathway **a**; further reaction with an $RO_2$ species leading to the structure proposed for OS-265 and OS-281, pathway **b**; or isomerization and reaction with $O_2$ eventually leading to OS-281 and OS-297, pathway **c**. Figure S8 does not therefore represent a "shift" in chemistry, but branching reactions leading to three observed product ions. We note that the RO radical precursor to pathways a, b and c may also result from an $RO_2 + HO_2$ reaction (Kautzman et al., 2010; Birdsall et al., 2011). Since it is not possible to distinguish whether RO originates from $RO_2 + RO_2$ or $RO_2 + HO_2$ reactions the alternative $RO_2 + HO_2$ reaction has been added to all mechanisms.

This paragraph has been changed to be consistent with proposed mechanism.

Lines 310-314: *"A scheme leading to the structure proposed in Figure 2a is based on the cleavage of the $C_1 - C_2$ decalin bond, followed by reaction with a second $O_2$ molecule and $HO_2$ leads to a terminal carbonyl hydroperoxide ($C_{10}H_{18}O_3$) (Yee et al., 2013). $C_{10}H_{18}O_3$ could then further react with OH radicals and lead to an epoxide and sulfate ester by reactive uptake/heterogeneous chemistry (Paulot et al., 2009)."*

14. Line 290: It is unclear whether the analytical technique is sufficient for seeing

hydroperoxide moieties on molecules as they are included in the proposed structures. Were hydroperoxide standards such as t-buytylhydroperoxide or cumene hydroperoxide run using this method to verify that the hydroperoxide moiety can be retained on the column? Or is there something about the organosulfates that allow for this? The authors should address this in the experimental methods section as well.

Such compounds can be retained on the column used in this project and we have demonstrated this for another project with a synthetic isoprene hydroxyhydroperoxide standard (ISOPOOH). As an example, please see the chromatogram below:

[Figure]

Moreover, Witkowski and Gierczak (2012) have recently developed a method to quantify hydroperoxide compounds formed from the ozonolysis of cyclohexene. The authors used a column similar ($C_{18}$) to that used in the present work. Electrospray ionization mass spectrometry operated in negative mode (Cech and Enke, 2001; Witkowski and Gierczak, 2012) is not highly sensitive to detection of non-acidic compounds, such as pure hydroperoxides or alcohols. However, LC/ESI(-)-MS provides excellent sensitivity for multifunctional compounds (like hydroperoxides and alcohols) containing the OS functional group, since the OS functional group yields an intense $[M - H]^-$ ion, as reported in many studies (Surratt et al., 2008; Kristensen et al., 2011; Kundu et al., 2013; Hansen et al., 2014).

As mentioned line 340-342, we expect to detect the presence of hydroxyl or hydroperoxide functional groups when the OS group is present in the multifunctional compounds analyzed by LC/ESI(-)-MS.

The following sentences has been added:

Lines 340-346: *"As a result, the presence of hydroperoxide and/or hydroxyl substituents is expected in order to satisfy the molecular formulas obtained by the accurate mass measurement. Although ESI-MS in the negative ion mode is not sensitive to multifunctional hydroperoxides and alcohols (Cech and Enke, 2001; Witkowski and Gierczak, 2012), this technique is highly sensitive to hydroperoxides and alcohols which also contain OS groups and give $[M – H]^-$ ions (Surratt et al., 2008; Kristensen et al., 2011; Kundu et al., 2013; Hansen et al., 2014)."*

Thank you for this excellent clarification.

15. Lines 295-296: The work of Yee et al., 2013 and Schilling Fahnestock et al., 2015 do not test decalin, so they should not be cited here to support proposed formation of a 1-hydroperoxy radical in the decalin system used in the current work. While the mechanisms laid out in Atkinson, 2000 can apply here, as worded it seems as if the authors are proposing the particular alkoxyl reference.

In Yee et al. 2013 they studied the oxidation of hexylcyclohexane and cyclododecane and they proposed (Figure 1, sidebar, Yee et al. 2013) a ring scission and formation of a terminal carbonyl hydroperoxide as proposed in Figure 3. We do not claim that both studies have investigated the oxidation of decalin but have used the analogous ring

scission sequence to explain our products. To avoid confusion regarding the content of the Yee citation, we have moved the citation in the text to follow the description of ring scission. Sentence has been modified in the revised manuscript.

Lines 348-352:*"Following analogous mechanisms Under low-NO$_x$ conditions, abstraction of a proton $\alpha$ to the ring scission of decalin followed by reaction with O$_2$ leads to the 1-hydroperoxy radical, which in turn can react with another RO$_2$ radical to yield the corresponding alkoxyl radical (C$_{10}$H$_{17}$O ) (Atkinson, 2000). Cleavage of the C$_1$−C$_2$ decalin bond, followed by reaction with a second O$_2$ molecule and HO$_2$ leads to a terminal carbonyl hydroperoxide (C$_{10}$H$_{18}$O$_3$) (Yee et al., 2013)."*

Please revise Lines 348-352 to be more precise as below.  The reference placement is still confusing, implying that specific C10 compounds were observed in the citations provided, though they are just being used for analogous mechanistic pathways.

Lines 348-352:*"Following analogous mechanisms for low-NO$_x$ conditions (Atkinson, 2000; Yee et al., 2013), abstraction of a proton $\alpha$ to the ring scission of decalin followed by reaction with O$_2$ leads to the 1-hydroperoxy radical, which in turn can react with another RO$_2$ radical to yield the corresponding alkoxyl radical (C$_{10}$H$_{17}$O). Cleavage of the C$_1$−C$_2$ decalin bond, followed by reaction with a second O$_2$ molecule and HO$_2$ leads to a terminal carbonyl hydroperoxide (C$_{10}$H$_{18}$O$_3$)."*

17. Line 298/Figure 3: To guide the reader it would be beneficial to update the mechanism in Figure 3 with the same label of tCARBROOH next to intermediate the authors are referring to

We have now added the formula of the different primary products.

Please add the labels to Figure 3 for clarity.

18. Lines 298/Figure 3: Do the authors see evidence of an analogous product in the case of decalin, as the OS-279 that was observed in the dodecane case? The proposed mechanism of carbonyl hydroperoxide heterogenous reactive uptake followed by OS formation should also be considered for the decalin tCARBROOH as well and

supported by the measurements/compared on the basis of volatility differences due to carbon number/ring structure and the impact of reactive uptake versus partitioning to the particle phase.

We thank the reviewer for its comment. We have revised the pathways proposed for decalin oxidation products OS-265; -267; -269 and -285 and cyclodecane oxidation products OS-249; -251; -265 and -267 to include reactive uptake of the hydroperoxide on wet acidic aerosols.

The appropriate mechanisms have been updated as well as the manuscript:

Lines 314-316: *"OS-265 ($C_{10}H_{17}O_6S^-$) could also arise from the acid-catalyzed perhydrolysis of the hydroperoxide ($C_{10}H_{18}O_4$) generated from the reaction of $C_{10}H_{17}O_4^{\cdot}$ + $HO_2$ (Figure S8, pathway b)."*

Lines 319-325: *"The pathway proposed in Figure S8 pathway **b** is based on gas-phase oxidation of a 4-(cyclohexan-2-one)but-1-yl radical followed by reaction with $O_2$ and a 1,5-H shift (Crounse et al., 2011; Orlando and Tyndall, 2012) and lead to a $C_{10}$-carbonyl- hydroxyhydroperoxide ($C_{10}H_{18}O_4$). $C_{10}H_{18}O_4$ could then further react with OH radical and by elimination of OH lead to an epoxide (Figure S8, pathway **b**). In addition, OS-281 could arise from acid-catalyzed perhydrolysis of $C_{10}$-carbonyl dihydroperoxides ($C_{10}H_{18}O_5$) as proposed in Figure S8, pathway **c**."*

Lines 357---374: *"$C_9H_{17}O_4^{\cdot}$ can react via pathway **a** (Figure 3) through a 1,6--H shift (Crounse et al., 2011; Orlando and Tyndall, 2012) followed by elimination of OH resulting in a formation of an epoxide analogous to the formation of isoprene epoxydiol (IEPOX) (Paulot et al., 2009; Mael et al., 2015). The epoxide can then undergo acid--catalyzed ring opening to give OS--269 ($C_9H_{17}O_7S^-$). The $MS^2$ spectrum of OS--285 ($C_9H_{17}O_8S^-$; Figure S5) shows product ions corresponding to $HSO_3^-$, $HSO_4^-$ and loss of neutral $SO_3$, in accord with a sulfate ester β to a labile proton, but yields no further structural information. The structure proposed for OS-285 is based on the formation of reaction of the hydroperoxyperoxyl radical intermediate in pathway **b** with $RO_2$ followed by a 1,4-H shift (Rissanen et al., 2015) and addition of $O_2$ to give a hydroxyhydroperoxyperoxyl radical ($C_9H_{17}O_5^{\cdot}$). $C_9H_{17}O_5^{\cdot}$ could then lead to an epoxide by isomerization (Iinuma et al., 2009; Surratt et al., 2010; Jacobs et al., 2013; Mael et al., 2015) and form OS-285. $C_9H_{17}O_5^{\cdot}$ could also react with $HO_2$ and form the corresponding*

*$C_9$-hydroxydihydroperoxide ($C_9H_{18}O_5$), which could then undergo heterogeneous reaction and lead to OS-269 (Figure 3, pathway **b**). Finally, a $C_9$--carbonyl hydroperoxide ($C_9H_{16}O_3$) could also be formed from the RO + $O_2$ reaction (Figure 3, pathway **c**), which could then further react with OH radicals and lead to a $C_9$--carbonyl dihydroperoxide ($C_9H_{16}O_5$). Hence, $C_9H_{16}O_5$ could form OS--267 ($C_9H_{15}O_7S^-$) from heterogeneous reaction on acidic aerosols."*

This analysis strengthens the findings of the work by making more clear in the text self-consistent mechanisms.

19. Lines 299-301: (e.g. remote areas). Please also refer to and reference (Peeters et al., 2009; Crounse et al., 2011; Orlando and Tyndall, 2012). The authors need to justify why the basis of their proposed mechanisms for OS formation rely on this pathway when (e.g. Figures 3, 4, and S8). Is it RO2 + RO2, RO2 + HO2, RO2 + NO, RO2 isomerization?

    The suggested references have been added to the revised manuscript. As discussed previously, different regime of $RO_2$ radicals could exist, either terminal ($RO_2$+ $HO_2$; $RO_2$ + $RO_2$; $RO_2$ + NO) or autooxidation reactions. In this study, we do not claim to propose all chemical pathways from the oxidation of the alkanes are examined. In most of the mechanisms we have considered the different potential $RO_2$ reactions ($RO_2$ + $RO_2$; $RO_2$ + $HO_2$; $RO_2$ + $RO_2$, and $RO_2$ autooxidation), which could lead to the identified OSs through multiphase chemistry of the products shown in the tentatively proposed mechanisms. RO radicals might have formed for other minor chemical channels, such as ROOH + h$\nu$ or $RO_2$ + NO, which were not initially included in the manuscript. It is important to note that these potential reactions, which are now included in the manuscript, do not change the different mechanisms tentatively proposed in this study. In addition we have proposed reaction sequences based on known/reported reactions that will lead to products consistent with the mass spectrometric data. This is the same approach used by other investigators, such as Yee et al. (2013).

20. Lines 301-305: The authors should address the extent of photolysis reactions affecting the fate of the proposed hydroperoxides and aldehydes in the system.

    Potential photolysis reactions are now discussed in the revised manuscript. We have incorporated the potential photolysis of hydroperoxides leading to RO radicals and

').

also  the photolysis of the aldehyde proposed in Figure 3, which could lead to the $RO_2$ radical ($C_{10}H_{17}O_5$

This version of the manuscript does not show the changes made to Figure 3 including photolysis.

Lines 353-357: *"The aldehydic intermediate in the sequence following $C_1$-$C_2$ ring scission may be oxidized to the corresponding acyl radical either by photolysis (Wang et al., 2006) or by H-abstraction (Kwok and Atkinson 1995) followed by addition of $O_2$, reaction with $RO_2$ or $HO_2$ and decarboxylation of the resulting acyl-oxy radical $(R(O)O)$ (Chacon- Madrid et al., 2013) to a hydroperoxyperoxy radical ($C_9H_{17}O_4$ )."*

21. Line 306: "previously unreported" is unclear. Do the authors mean previously unreported in ambient data or previously unreported from similar experiments?
OS-267 has been identified in previous smog chamber experiments. Sentence has been modified in the revised manuscript.

Lines 370-374: *"Finally, a $C_9$-carbonyl hydroperoxide ($C_9H_{16}O_3$) could also be formed from the $\overline{RO}$ + $O_2$ reaction (Figure 3, pathway **c**), which could then further react with OH radicals and lead to a $C_9$-carbonyl dihydroperoxide ($C_9H_{16}O_5$). Hence, $C_9H_{16}O_5$ could   form OS-267 ($C_9H_{15}O_7S$  ) from heterogeneous reaction on acidic aerosols."*

22. Line 311: As worded, OS-267, is proposed to originate from further oxidation of OS-269, but the arrows drawn in Figure 3 are inconsistent suggesting origin from the epoxide.
We have corrected the revised manuscript.

23. Line 327/Figure 4: The description of Ring cleavage of the C10 alkoxy radical is not consistent with the "ISO"/isomerization descriptor in Figure 4. Please clarify that pathway.
ISO descriptor in Figure 4 (*pathway a*) indicates the isomerization of the $RO_2$ formed from the ring cleavage and lead to hydroperoxide functional group.

Sentences have been added to better discuss this pathway:

Lines 382-388: *"The salient features of pathway a include oxidation of the $RO_2$ to 2-decalinone, formation of a $C_{10}$ alkoxy radical followed by ring cleavage of the $C_9-C_{10}$ decalin bond and further $RO_2$ isomerization (1,8-H shift) leading to a 4-(carboxy cyclohexyl)-1-hydroperoxybut-2-yl radical via $RO_2$ chemistry. Although considered as a minor reaction pathway (Crounse et al., 2013), the acyloxy radical could lead to the epoxide from the isomerization of the $O_2$ adduct (Paulot et al., 2009; Yao et al., 2014; Zhang et al., 2015). Further acid-catalyzed ring opening of the epoxide leads to OS-295 ($C_{10}H_{15}O_8S$ )."*

24. Line 340: Figure 1d does not exist. Clarify the reference.

    The reference was Figure 2d and not 1d, the text has been appropriately modified.

25. Lines 343-345: Sentence is awkward beginning with "Pathway c", and from what figure? Clarify that it is Figure 4. Again, citation of Atkinson, 2000 seems inappropriate as the sentence is written.

    Sentence has been changed, pathway **c** referred to Figure 4. Citation of Atkinson (2000) was used to support formation of an organonitrate from $RO_2$ + NO reaction, since it is a common reaction as discussed in the Atkinson's review.

    Lines 402-408: *"Although $RO_2$ + NO reactions are expected to be minor under the conditions used in this work (i.e. NO < 1 ppb, formation of RO radicals or organonitrates cannot be ruled out. Indeed, Ehn et al. (2014) have demonstrated that NO reactions could be competitive at ppb levels. Under our experimental conditions $RO_2$ + NO, $RO_2$ + $HO_2$ and $RO_2$ autoxidation are possible. Therefore, the parent ion at m/z 326 could arise from the reaction of the decalin-2-peroxy radical with NO to form decalin-2-nitrate ($C_{10}H_{17}NO_3$) with subsequent reactions shown in Figure 4, pathway c"*

    Thank you for clarifying the experimental conditions.

26. Lines 343-345/Figure 4, pathway c: Here the authors propose that RO2 + NO chemistry is occurring to form a nitrate containing OS. This contradicts the authors' earlier statement in lines 237-243 stating that the RO2 + NO reactions are not significant in their experimental setup. The authors need to handle in more detail the fate of RO2 under the unclear experimental conditions.

    As demonstrated and discussed by Ehn et al. (2014) at ppb levels of NO (which is

even higher than the conditions of our study) a competition exists between $RO_2 + NO$; $RO_2 + HO_2$ and $RO_2$ autoxidation reactions. The Ehn et al. (2014) study demonstrates that ELVOC, even though reduced, are still formed at NO concentrations greater than few ppb. It is important to point out that the concentrations of VOCs used in this work could also lead to $RO_2 + RO_2$ chemistry. Therefore, not only one $RO_2$ reaction could occur and the different $RO_2$ reactions have to be considered, which has been done in this study.

$O_3$ and NO concentrations are provided in the revised manuscript (Table 1). In addition a paragraph has been added describing the fate of $RO_2$.

Lines 404-406: *"Indeed, Ehn et al. (2014) have demonstrated that NO reactions could be competitive at ppb levels. Under our experimental conditions $RO_2 + NO$, $RO_2 + HO_2/RO_2$ and $RO_2$ autoxidation are possible."*

27. Lines 349-364: Why is discussion of OS-281 and OS-297 featured here, when discussion of OS- 265 is discussed near the beginning of Section 3.2? Since they are all referenced in Figure S8, their chemistry should be discussed together from the same mechanistic precursors.

We chose to describe the formation and tentative structural assignments of ions observed on ambient filters at the beginning of the discussion, as explained in the manuscript on lines 293 to 295 ("*Figures 2 and S2 present $MS^2$ spectra and fragmentation schemes of selected parent ions at m/z 265.0749 (OS-265), 269.0696 (OS-269), 295.0494 (OS-295) and 326.0554 (OS-326). $MS^2$ spectra and fragmentation schemes of other OSs are reported in Figure S3-S7. The selected OSs were, as described in the next section, quantified and characterized in the fine urban aerosol samples."*).

However, section 3.2 has been reorganized as requested by the reviewer #3 and formation pathways of OS-265, -281 and -297 are included in the same paragraph.

Thank you.  This is more clear.

28. Section 3.2: Authors should clarify the main mechanistic differences and relative

importance between Figure S8, Figure 3, and Figure 4, and the flow of products to be discussed at the beginning of Section 3.2. Currently as written, the flow of Section 3.2 is very arbitrary when choosing different OS products to discuss.

We decided to separate the different reaction pathways for clarity since it would not have been clear and quite difficult/confusing to propose in one figure the formation pathways of all OSs. The different reaction pathways are separated based on OSs that are generated from branching reactions of a common transient. This section has been modified as discussed in the previous point. The importance of the proposed pathways cannot be evaluated based on this study and this was not the study objective.

We have added a sentence to clarify this point.
Lines 297-299: "*The different reaction pathways presented below, are separated based on OSs that are generated from branching reactions of a common transient.*"

Great clarification. Thank you.

29. Lines 377-380: Incorrect use of citations here. Yee et al., 2012 do not propose RO2 + RO2 chemistry and therefore a "precedent" has not been established. The authors should not be citing Atkinson, 2000; Yee et al., 2012 and Raff and Finlayson-Pitts, 2010 to speak for the experimental conditions in the current work. The mechanism of $RO2 + RO2 \rightarrow RO$ can be supported by work on general atmospheric chemistry mechanisms including Atkinson, 2000 and many other works, and so if this mechanistic pathway is to be cited, than many other works should be cited as well.

We agree with reviewer that Yee et al. is an inappropriate reference and have removed it. General references on atmospheric chemistry have been added: Atkinson and Arey (2003, Chem Rev, 103, 4605-4638) and Ziemann and Atkinson (2012, Chem Soc. Rev., 41,

6582-6605).

30. Line 379: Inconsistent citation here compared to line 300 for similar mechanistic argument.

We have cited Ehn et al., 2014 and Jokinen et al., 2014 and Mentel et al., 2015 for both transformations.

31. Figure S14: Why do the authors propose in the case of cyclodecane formation of the hydroperoxide from RO2 + HO2 pathways and subsequent chemistry thereof, but not in the case of decalin in any of Figures 3, 4, and S8? Further, Figure S14 outlines a mechanism from further reaction of the hydroperoxide to get to an epoxide that then enters the particle phase to produce OS-251 and OS-249. This seems like a plausible analogous mechanism to propose for the case of dodecane rather than reactive uptake of a carbonyl hydroperoxide. Why do the authors propose different mechanisms between dodecane and cyclodecane to generate the similar analogs (OS-279, OS-249)?

As mentioned in the point #18 above, cyclodecane-derived OSs might be formed from the heterogeneous chemistry of hydroperoxides. We have added these different pathways in the different Figures as well as in the manuscript. Please note that we have now combined Figures S14 and S15 to present the OS formation from cyclodecane in one Figure.

These changes make the mechanisms proposed for all systems more consistent, though please make it also clear in the discussion on dodecane why analogous pathways (epoxide formation and subsequent uptake, isomerization, etc.) as proposed in the C10 system are also likely/unlikely for the conditions of the experiment. It would be good to discuss the DBE determination in a non-epoxide route to OS-279. It is not mentioned in text of section 3.1 or in Figure 1

Lines 428-435: *"The formation of compounds such as cyclodecanone ($C_{10}H_{18}O$), cyclodecane hydroperoxide ($C_{10}H_{20}O_2$) or cyclodecane hydroxyhydroperoxide ($C_{10}H_{20}O_3$) are proposed as intermediate products leading to epoxy-compounds after additional oxidation/isomerization processes, as presented in Figure S14. In addition $C_{10}H_{20}O_3$, cyclodecane hydroperoxide ketone ($C_{10}H_{18}O_3$) and cyclodecane hydroxyoxohydroperoxide ($C_{10}H_{18}O_4$), proposed as intermediate products, could condense onto wet acidic aerosols and lead to the corresponding OSs through acid-catalyzed perhydrolysis reactions (Figure S14)."*

As described above we proposed different fates for the $RO_2$ radicals: $RO_2 + RO_2$, $RO_2 + HO_2$ and $RO_2$ autoxidation reactions in the different schemes proposed for the photooxidation of decalin and cyclodecane. $RO_2 + HO_2$ reactions are also proposed in the

case of the photooxidation of decalin, which likely explain the formations of

hydroperoxides as discussed above.

32. Section 3.4: This section is weak and little effort is made to really describe the chemical differences between the systems to interpret the findings. There should be comparisons of vapor pressures of the precursors and carbon numbers and discussion of previously published yields from these compounds to support the discussion. How does quantification using the available OS surrogate standards potentially affect the OS quantification across these systems/factoring in different sensitivities?

The objective of this study is to establish that OSs may be products of the photooxidation of anthropogenic precursors, such as the alkanes examined here, and thus to demonstrate the relevance of this chemistry observations of aliphatic OSs in urban areas (Mao et al., 2014; Tao et al., 2014). Since the reaction pathways leading to the products observed in this study and in ambient samples are tentative, we feel that discussion at the level of thermodynamics is not justified and have deleted Section 3.4.

33. Line 400: The authors claim that "the presence of acidic aerosols significantly increase OS formation in most cases". However, is this just an effect of using an atomized solution with more sulfate (0.06M ammonium sulfate + 0.06M sulfuric acid) in the acidic case versus only 0.06M ammonium sulfate in the non-acidified case? It may be a concerted effect of more available sulfate in the "acidic" case as well as acidity.

It has been demonstrated in previous studies (cited references) that acidity rather than the concentration of sulfate is a key parameter in the formation of OS. Chan et al. (2011) demonstrate that the formation of OSs from the oxidation of $\beta$-caryophyllene is directly correlated with the aerosol acidity ([H$^+$]).

Understood, though please address this point directly in the text in anticipation of the casual reader in the field who may not have as much knowledge of the literature specifically addressing the role and dynamics of sulfate concentration vs aerosol acidity (which in stated reference is still an indirect measurement of the acidity at which the OS actually formed, but an accepted proxy.) As authors state in lines 455-463 and cited literature, there are many ways to form OS, so the concentration of sulfate as a precursor to sulfate anion radical should be a factor in the chemistry.

34. Line 484: The authors return to claim that the experiments are conducted under dominant "RO2/HO2" chemistry—this is contradictory to the formation of OS-326

containing a nitrate group.

This issue has been discussed previously in response to reviewer comments # 10, 13, 19, 26, and 31 above.

35. Lines 484-486: Enhancement of OS due to acidified ammonium sulfate seed needs to be addressed with regard to the effect of just having introduced more sulfate into the experiments compared to the non-acidified case. See earlier Major Comment, 33.

As discussed in response to Reviewer comment # 33, enhancement of OSs has been demonstrated to result from an increase of the aerosol acidity (cited references) and our work is consistent with these studies.

36. Lines 491-496: The "novel pathway" involving reactive uptake of hydroperoxides is not well- substantiated in the current work and is mostly speculation. The vapor pressure alone of the carbonyl hydroperoxide makes it a potential candidate to partition to the particle phase, not via reactive uptake. There are no direct measurements of hydroperoxides in the gas phase, and insufficient discussion on if hydroperoxides are detected in the particle phase using the UPLC technique. Further, if reactive uptake is at play, why have the authors not seen the corresponding decalin analog of carbonyl hydroperoxide?

We agree with reviewer #3 that "reactive uptake" of hydroperoxides is currently not well substantiated, and have clearly indicated that this pathway is tentative. With regard to reactive uptake and perhydrolysis of carbonyl hydroperoxides generally as an alternative pathway the revised manuscript cites this route as a possibility in the formation of 8 OSs, as indicated in response to earlier comments. Also as discussed above in response to Reviewer comment #14, LC-EI/MS in the negative ion mode, used to identify the OS products, will not detect analytes (such as hydroperoxides) not containing substituents readily yielding negative ions.

**Minor Comments:**

1. Line 30: "Both studies strongly support formation of OSs" is awkward. Reword, for example, "Both studies strongly support that OSs can form from the gas-phase oxidation of anthropogenic precursors..."

The wording has been revised on lines 30-33 as follows:

"*Both studies strongly support the formation of OSs from the gas-phase oxidation of anthropogenic precursors, as hypothesized on the basis of recent field studies in which aromatic and aliphatic OSs were detected in fine aerosol collected from several major urban locations.*"

2. Line 48: Change "aerosol" to "particles", as aerosol is technically defined as both  gas + particle.

   The wording has been revised as suggested.

3. Line 76: Insert comma after "2015)".

   A comma has been inserted.

4. Line 83: Change comma to semi-colon after "2007,".

   A semi-colon has been inserted.

5. Line 90: Delete "of".

   Use of "of" is appropriate and we have not made this change.

6. Line 103: Change "reduce" to "reduces".

   The revision has been made as suggested.

7. Line 136: Check misprint on the high humidity range listed as "(4-60%)".

   The correction has been made (i.e. 40-60%).

8. Line 220: Insert after ")", ", hereafter referred to as OS-279, ".

   The change has been made as suggested.

9. Line 268: "ion at m/z 265.0749" should be "ion at m/z 265.0752" according to Figure 2.

   "at" has been inserted.

10. Line 315: Add in "Figure 3, pathway a" to be clear.

    "Figure 3, pathway a" has been inserted.

11. Line 345: Change "identical" to "analogous" as the sequence of reactions are certainly not identical as shown in Figure 4.

   "Analogous" has been substituted.

12. Line 350: Change chemical formula to include S for OS-281.

   The formula has been corrected.

13. Line 352: Rewrite the sentence. The radical reacts with O2, followed by 1,6 H shift, etc.

   Sentence has been modified on lines 319-325 to:

   *"The pathway proposed in Figure S8 pathway **b** is based on gas-phase oxidation of a 4-(cyclohexan-2-one)but-1-yl radical followed by reaction with $O_2$ and a 1,5-H shift (Crounse et al., 2011; Orlando and Tyndall, 2012) and lead to a $C_{10}$-carbonyl-hydroxyhydroperoxide ($C_{10}H_{18}O_4$). $C_{10}H_{18}O_4$ could then further react with OH radical and by elimination of OH lead to an epoxide (Figure S8, pathway **b**). In addition, OS-281 could arise from acid-catalyzed perhydrolysis of $C_{10}$-carbonyl dihydroperoxides ($C_{10}H_{18}O_5$) as proposed in Figure S8, pathway **c**."*

14. Lines 362-363: Rewrite the awkward phrasing, "which be reactively taken up to give a sulfate ester".

   Sentence has been changed.

   Lines 331-334: *"However, in contrast to pathway **b**, $RO_2$ formed by the addition of $O_2$ undergoes a 1,6-H shift (Crounse et al., 2011; Orlando and Tyndall, 2012) followed by addition of a second $O_2$ molecule, a 1,5-H shift and elimination of OH to yield an epoxide, which leads to a sulfate ester by reactive uptake onto acidified aerosols."*

15. Lines 371-374: Poor grammar. Rewrite sentence.

   The sentence has been changed on lines 419-422 to:

   *"None of the fragment ions observed in the $MS^2$ spectrum suggests the presence of a terminal carbonyl or a carboxyl functional group in the cyclodecane-OSs, which is consistent with conservation of the cyclodecane ring."*

16. Line 379: "hydroperoxydes" is spelled wrong.

   The spelling has been corrected.

17. Line 459: Add "of" after "oxidation".

"of" has been added.

18. Line 482: Add "," after "cyclododecane".

The comma has been added.

19. Check Table 1 entry for Decalin Acidified Seed RH Range of 51-49% Initial HC 180 ppb  Is RH range correct?

**Additional References:**

Atkinson, R.; Arey, J.; Aschmann, S. M. Atmospheric Chemistry of Alkanes: Review and Recent Developments. Atmos. Environ. 2008, 42 (23), 5859–5871.

Crounse, J. D.; Paulot, F.; Kjaergaard, H. G.; Wennberg, P. O. Peroxy Radical Isomerization in the Oxidation of Isoprene. Phys. Chem. Chem. Phys. 2011, 13 (30),  13607–13613.

Dibble, T. S. Cyclization of 1,4-Hydroxycarbonyls Is Not a Homogenous Gas Phase Process. Chem. Phys. Lett. 2007, 447 (1-3), 5–9.

Lambe, A. T.; Onasch, T. B.; Croasdale, D. R.; Wright, J. P.; Martin, A. T.; Franklin, J. P.; Massoli, P.; Kroll, J. H.; Canagaratna, M. R.; Brune, W. H.; et al. Transitions from Functionalization to Fragmentation Reactions of Laboratory Secondary Organic Aerosol (SOA) Generated from the OH Oxidation of Alkane Precursors. Environ. Sci. Technol. 2012, 46 (10), 5430–5437.

Lim, Y. Bin; Ziemann, P. J. Chemistry of Secondary Organic Aerosol Formation from OH Radical-Initiated Reactions of Linear, Branched, and Cyclic Alkanes in the Presence of NOx. Aerosol Sci. Technol. 2009a, 43, 604–619.

Lim, Y. Bin; Ziemann, P. J. Kinetics of the Heterogeneous Conversion of 1,4-Hydroxycarbonyls to Cyclic Hemiacetals and Dihydrofurans on Organic Aerosol Particles. Phys. Chem. Chem. Phys. 2009b, 11 (36), 8029–8039.

Orlando, J. J.; Tyndall, G. S. Laboratory Studies of Organic Peroxy Radical Chemistry: An Overview with Emphasis on Recent Issues of Atmospheric Significance. Chem. Soc. Rev. 2012, 41 (19), 6294.

Peeters, J.; Nguyen, T. L.; Vereecken, L. HO X Radical Regeneration in the Oxidation of Isoprene W. Phys. Chem. Chem. Phys. 2009, 11, 5935–5939.

Robinson, A. L.; Donahue, N. M.; Shrivastava, M. K.; Weitkamp, E. a; Sage, A. M.; Grieshop, A. P.; Lane, T. E.; Pierce, J. R.; Pandis, S. N. Rethinking Organic Aerosols: Semivolatile Emissions and Photochemical Aging. Science 2007, 315 (5816), 1259–1262.

---

## Author Response (AR2)

**Response to Anonymous Referee #1**

**We thank Referee # 1 for the comments and address each below. Our responses are denoted in blue texts.**

In this study the authors report the formation of organosulfates from the photooxidation of C10-C12 alkanes. The effect of acidity, humidity, and solvent extraction on the formation and quantification of organosulfates were investigated. The authors evaluated possible mechanisms for the formation of the assigned organosulfates. The manuscript went through the ACPD open discussion process and got mixed reviews, with a number of good suggestions from the more critical reviewers. Based on my examination of the authors' responses to the reviewers' comments, I believe the concerns of the reviewers have been adequately addressed. This revised manuscript is well written and provides evidence for the importance of anthropogenic precursors in the formation of organosulfates that have also been identified in field studies. The manuscript could be published as is, but I have a few minor comments that follow.

**We thank referee #1 for its careful consideration of our article.**

1. The effect of filter solvent extraction on the quantification of organosulfates was investigated by using either methanol or an acetonitrile/toluene mixture. The extracts were dried and then reconstituted in solvents, but not in the same solvents as the ones used in the filter extraction step. Tao et al. 2014 saw a difference in quantified organosulfates between an acetonitrile/toluene mixture and an acetonitrile/water mixture that were used in extraction and direct MS analysis, which suggest very different solubility of organosulfates in these solvents. What was the rationale for using a different solvent for the extraction and re-dissolution? Is there a concern that the initially extracted compounds remained undissolved during the re-dissolution step (i.e., the less polar aliphatic species more soluble in the acetonitrile/toluene solvent that were initially extracted from the filters were not re-dissolved in the acetonitrile/water mixture)?

Filter extractions were done using either methanol or an acetonitrile/toluene (70:30) solvent mixture in order to assess how well the extraction solvents remove the alkane-derived organosulfates from the Teflon filter media. Since we used a reverse-phase chromatography technique that employed methanol as the organic mobile phase, we reconstituted both sets of dried extracts in a 50:50 methanol/water solvent mixture in order to prevent any separation/chromatographic issues. This is consistent with what is usually performed in the literature (Surratt et al., 2008; Kristensen and Glasius, 2011).

However, it is possible that the concentrations of the aliphatic OSs could be underestimated due to the issue pointed out by the reviewer. For example, more work is clearly needed to better investigate/quantify this effect by using internal standards. In order to highlight this

point, we have added few sentences to the revised version as follows:

Lines 485-486: "It is important to note that the concentrations of the aliphatic OSs could be underestimated due to their potential partial re-dissolution in the reconstitution solutions."

Lines 497-499: "However, more work is needed to better characterize and elucidate the impact of solvent mixture on the quantitation of biogenic- and anthropogenic-derived OSs, especially compounds  $> C_{10}$ , by using internal standards."

2. The authors did not appear to directly respond to previous comments from Referee #3 posted during the ACPD open discussion stage. I think more attention needs to be paid to these comments:

"Further, it is unclear if the conclusion that enhancement of OS yields are due to increased acidity of the seed aerosol is really due to acidity, rather than an effect of seeding the experiments with an atomized solution containing more sulfate."

"Line 400: The authors claim that "the presence of acidic aerosols significantly increase OS formation in most cases". However, is this just an effect of using an atomized solution with more sulfate (0.06M ammonium sulfate + 0.06M sulfuric acid) in the acidic case versus only 0.06M ammonium sulfate in the non-acidified case? It may be a concerted effect of more available sulfate in the "acidic" case as well as acidity."

"Lines 484-486: Enhancement of OS due to acidified ammonium sulfate seed needs to be addressed with regard to the effect of just having introduced more sulfate into the experiments compared to the non-acidified case. See earlier Major Comment, 33."

In response to these comments, authors cited additional references, but the main concern is that by adding sulfuric acid to acidify SOA, the effect of varying the acidity is not cleanly separated from the effect in the increase in sulfate concentration. Cited references also use sulfuric acid to change SOA acidity. It is important to note that no control was conducted to show that under the same initial conditions, but with increased sulfate concentration there is not a change in organosulfate formation. There is not a disagreement that acidity can increase organosulfate formation, but a question of whether an increased concentration of sulfate can affect organosulfate formation.

As requested by reviewer #3, we have discussed the impact of the acidity on OS formation. We agreed with reviewer that most of the previous studies have investigated the impact of acidity by varying the concentration of sulfuric acid. However, in the cited reference (i.e. Chan et al., 2011), the authors have changed the acidity of the seed particles by adjusting the ratio of the aqueous  $(NH_4)_2SO_4/H_2SO_4$  solutions to produce a constant aerosol sulfate concentration of 30 µg m-3 across the range of studied acidities. Hence the authors have previously reported that the OS formation is directly correlated with the increase of the

aerosol acidity and not due to the larger presence of sulfate. We agreed with reviewer that more work is needed to better understand the impact of sulfate concentration and aerosol acidity on the OS formation for different systems.

A few sentences have been added to discuss this point:

Lines 453-459: "It is important to point out that the effect of varying the aerosol acidity was not cleanly separated from the potential impact of larger concentrations of aerosol sulfate. However, Chan et al. (2011) have demonstrated that the formation of OSs from the oxidation of  $\beta$ -caryophyllene is directly correlated with aerosol acidity ([H+]). Indeed, the authors have changed the acidity of the seed aerosols by adjusting the ratio of the aqueous (NH4)2SO4/H2SO4 solutions to produce a constant aerosol sulfate concentration of 30 µg m-3 across the range acidities."

Tao, S., Lu, X., Levac, N., Bateman, A.P., Nguyen, T.B., Bones, D.L., Nizkorodov, S.A., Laskin, J., 870 Laskin, A., and Yang, X.: Molecular characterization of organosulfates in organic aerosols from Shanghai and Los Angeles urban areas by nanospray-desorption electrospray ionization high-resolution mass spectrometry, Environ. Sci. Technol., 48 (18), 10993-11001, 2014.

Response to Anonymous Referee #3

We thank Referee # 3 for the comments and address each below. Our responses are denoted in green texts.

**Reviewer's Report on Revised Manuscript**

The manuscript is much improved from the first version, as the experimental conditions are made clearer in this version (in particular the expected fate of RO2 radicals) and the discussion of the chemical mechanisms more complete. The analysis is better communicated and now further substantiates the conclusions reached. In addition, citation of the literature is now more precise. I would recommend it for publication after the remaining revisions requested are addressed. Reviewer's comments and requested revisions are in red text. Perhaps it was the submitted version of the manuscript, but some of the figures are not updated as authors say in response to Major Comments 7, 17, and 20. We thank referee #3 for its careful consideration of our article.

Response to Anonymous Referee #3

We thank Referee # 3 for the comments and address each below. Our responses are denoted in blue texts.

Review of "Characterization of Organosulfates in Secondary Organic Aerosol Derived from the Photooxidation of Long-Chain Alkanes"

Reviewer's Summary:

The authors characterize organosulfates (OSs) from the laboratory oxidation of dodecane, decalin, and cyclodecane under varying conditions of humidity and two different seed types (non-acidified, acidified). They observe overlapping organosulfates in the laboratory experiments and on filters from Pasadena, USA and Lahore, Pakistan, concluding that OSs from the oxidation of anthropogenic precursors may contribute to urban SOA. The results are novel and would be of interest to the readers of ACP; however, I would not recommend this manuscript for publication because it is not well-written and the conclusions are highly speculative. In particular, the proposed chemical mechanisms from the laboratory experiments

are not substantiated by a fundamental knowledge of the chemistry occurring in the reaction chambers used. The authors inconsistently address the fate of the RO2 radical within their laboratory experiments throughout the text and within the proposed mechanisms. There seems to be a mix of RO2 reacting with RO2, HO2, and NO, though they claim different regimes depending on what mechanism they are proposing to explain the OSs formed. For example, they state that reaction with NO is insignificant, yet they report a nitrate containing OS in the decalin system.

First, with regard to RO2 chemistry: as demonstrated by Ehn et al. (2014), at ppb levels of NO (1-5 ppb; NO concentration in our study, < 1 ppb based on NO measurement) competition exists between RO2 + NO, RO2 + HO2 and RO2 autoxidation reactions. Nevertheless, ELVOC, though reduced, still formed, indicating that auto-oxidation does not occur solely under pristine conditions. It is important to point out that the high concentrations of VOCs used in this work favor involvement of RO2 + RO2 chemistry. In addition, previous work (Crounse et al., 2013), has also reported different RO2 regimes, such as autooxidation or RO2 + HO2 reactions, in experiments using methyl nitrite as an OH radical source, similar to isopropyl nitrite used in our study. As discussed below in response to reviewer's comment # 10, RO2 + NO reactions are minimal; however, the nitrated OSs at *m/z* 326 are also measured in low concentrations (ng/m3, Table S3).

Thank you for clarifying the conditions of the experiment. It is still interesting that while the OSs do not make up the majority of the OA produced, the OS-326 can make up anywhere from 3% to 14% of the total OS mass though.

Second: we do not claim that the proposed mechanisms represent the major reaction pathways of the photooxidation of the studied alkanes, but are tentatively proposed to explain the formation of the OSs identified in this study. Mechanisms have been clearly indicated as *proposed* branching of pathways of the alkane photooxidations presented to explain formation of specific OSs products consistent with MS2 data. This approach to rationalizing OH oxidation products is universally applied in oxidation studies (Yee et al 2013; Bugler et al., 2015). Furthermore, the concentrations reported in Tables S1-S3 emphasize the fact that identified compounds are in low abundance regardless of the mass of SOA measured in all experiments.

It is certainly not expected that a fully exhaustive list of mechanistic pathways are presented for each studied alkane, but the proposed pathways to OS formation should be self-consistent with the conditions of the experiment and between precursors. Stated more broadly, the presence of analog OSs between two HC systems should not be proposed to form under different mechanistic pathways since the authors state similar chemical conditions for the systems studied. Further, the absence of an analog OS in one system and presence in another system should also be considered when proposing a mechanism that would likely happen in both systems. With the fate of the RO2 radical now clearly communicated and consideration of more analogous pathways between the precursor systems, the proposed pathways have more credence.

We agreed with reviewer #3 and we have tried to better discuss the different pathway leading to the identified OSs.

Finally: we stress to Reviewer #3 that neither the analytical work nor the interpretation of the MS2 data were questioned. The tandem MS data are consistent with the structures proposed for products observed in both the lab-generated and ambient aerosol samples and we repeat that we clearly indicated in the text that the mechanisms presented are suggested as pathways leading to ions consistent with those observed and until authentic standards become available both the product structures and mechanisms of formation remain tentative. An additional and crucial comment we make is that the major objective of our study is to demonstrate that aliphatic organosulfates form via alkane oxidation, and offer one explanation for reports of aliphatic OSs in urban areas (Ma et al., 2014; Tao et al., 2014).

This is an interesting chemical feature that is different between the proposed mechanism for OS formation in dodecane compared to the C10 systems, not included in textual description on OS-279 proposed pathway. This point also is lost in general textual descriptions of epoxide route to OS from these HCs. The authors might consider highlighting this finding as a nuance between the systems and possibility of enhanced epoxide formation from C10 cyclic systems and subsequent OS formation. This fits better in the context of atmospheric relevance and motivation (confusing in the first version) as the authors cite the potential for SOA formation from C10 cyclic alkanes to be greater than linear or branched C12 alkanes.

We thank the reviewer for its comment and we have added a sentence to highlight this finding as suggested.

Lines 536-538: "It is interesting to note that OS formation through reactive uptake of epoxides have been only observed for cyclic alkanes, which is consistent with the larger concentration of OSs identified from the oxidation of cyclodecane and decalin."

They propose the formation of hydroperoxides in the case of dodecane experiments with high initial precursor concentrations and do not propose RO2 + RO2 chemistry, but for the C10 systems RO2 + RO2 reactions are proposed with some RO2 + HO2 reactions. They propose epoxide precursors in the C10 systems to OS formation, but not in the C12 system. In general, the proposed mechanisms are arbitrary and do not demonstrate careful control in the design of the experiments or understanding of the chemistry. This lack of understanding becomes clear because there are several areas where citations are used to support the current work, but the citations are used imprecisely and out of context. The manuscript would benefit from more clearly stated organizational structure (e.g. why some mechanisms are proposed in the main text versus the supplemental information).

First we would like to point out that pathways have been proposed that lead to structures consistent with the mass spectrometric data acquired for the observed OSs. We have not attempted to hypothesize general mechanism that would be predicted to give the entire array of precursors contributing to the total mass of SOA. Regarding the possibility of an epoxide precursor to OS-279, we considered the possibility of formation of OS-279 from the reactive uptake of the corresponding epoxide ( $C_{12}H_{24}O$ ); however, the composition of OS-279 (1 DBE) is inconsistent with reactive uptake of an epoxide. Therefore, we have used the few studies available in the literature (Yee et al., 2012; 2013) to propose the formation of OS-279 from the heterogeneous chemistry of hydroperoxides. Finally, we have considered the potential heterogeneous chemistry of hydroperoxides formed from the photooxidation of decalin and cyclodecane as discussed above.

The authors should also clarify motivation in the experimental selection of two C10 cyclic alkane structures and one C12 straight chain structure.

These compounds have been selected due to their potential contribution to SOA formation in the atmosphere. Recent studies have investigated the SOA formation from decalin and dodecane oxidations and reported large SOA yields (Yee et al., 2013; Hunter et al., 2014). Moreover, Pye and Pauliot (2012) have shown that, even though less emitted into the atmosphere, the cyclic  $C_{10}$  alkanes have a greater potential for SOA formation than linear or branched alkanes <  $C_{12}$ .

Check all misspellings on reference to Pauliot. Should be Pouliot. The change has been made as suggested.

122:

A few sentences have been added to better explain our selection of parent VOCs on lines 118-

"These alkanes were selected based on their potential contribution to atmospheric SOA formation (Hunter et al., 2014). Studies have demonstrated that cyclic compounds (<  $C_{12}$ ) are expected to be more efficient SOA precursors than linear or branched alkanes with the same number of carbons (Lim and Ziemann, 2005; Pye and Pauliot, 2012). Alkanes  $\geq C_{10}$  are considered as effective SOA precursors, especially when placed in the context of their emission rates (Pye and Pauliot, 2012)."

The brevity of the discussion of results on the OSs from dodecane photooxidation are quite brief relative to the other sections interpreting the results from decalin and cylodecane, and the effects of chemical structure are glossed over in brevity.

As mentioned in lines 229-231, low abundances of OS-209 and OS-237 precluded acquisition of high-resolution  $MS^2$  data, and thus, structures have not been proposed for the parent ions. Without structural information, discussion of formation pathways is not possible. The criticism of our conclusions as "highly speculative," is not consistent with request for more detail on the chemical structure of products formed in abundances too low to obtain high-resolution  $MS^2$  data.

Thank you for adding this clarification in the revised version.

Further, it is unclear if the conclusion that enhancement of OS yields are due to increased acidity of the seed aerosol is really due to acidity, rather than an effect of seeding the experiments with an atomized solution containing more sulfate. These concerns are outlined in detail below.

This point has been previously discussed and published work demonstrates that acidity, rather than concentration of sulfate, is the key parameter (reference cited in the article) in the formation of OSs. Chan et al. (2011) have reported that the formation of OSs from the oxidation of  $\beta$ -caryophyllene is directly correlated with the aerosol acidity ([H+]).

Thank you for clarifying this; the authors should consider adding such discussion to the manuscript to clarify the experimental methods and design as in Major Comment 33.

We have now discussed this point in the manuscript as suggested by the reviewer:

Lines 453-459: "It is important to point out that the effect of varying the aerosol acidity was not cleanly separated from the potential impact of larger concentrations of aerosol sulfate. However, Chan et al. (2011) have demonstrated that the formation of OSs from the oxidation of  $\beta$ -caryophyllene is directly correlated with aerosol acidity ([H+]). Indeed, the authors have changed the acidity of the seed aerosols by adjusting the ratio of the aqueous (NH4)2SO4/H2SO4 solutions to produce a constant aerosol sulfate concentration of 30 µg m-3 across the range acidities."

**Major Comments:**

1. Lines 58-61: These lines are specious in the use of citations and misleading. First, as written, these lines assert that the underestimate of global SOA is equivalent to an underestimate in urban SOA. Second, the references cited (Pye and Pouliot, 2012; Tkacik et al., 2012) do not specifically argue that the underestimate in predicted SOA is due to the omission of IVOCs. A better reference here based on the lines as written would be (Robinson et al., 2007). Pye and Pouliot, 2012 can be cited for exploring additional mechanisms (oligomer formation) from alkane and PAH in SOA formation, and Tkacik et al., 2012 can be cited for providing additional evidence that IVOCs may be a missing source in modeling urban SOA, but the authors need to reword these lines carefully and be more precise.

Sentence has been modified on lines 59-62 as follows:

"The omission of intermediate volatility organic compounds (IVOC) as SOA precursors, such as alkanes or polycyclic aromatic hydrocarbons (PAHs), could contribute in part to the underestimation of SOA mass observed in urban areas (Robinson et al., 2007; Tkacik et al., 2012)."

Line 65: References here should include Yee et al., 2013 which more specifically addresses analogous to Lim and Ziemann, 2005 the products and mechanisms of C12 alkanes of varying structures.

References have been added.

3. Line 66: Tkacik et al., 2012 should be included here for presenting yields from several alkane systems.

Reference has been added.

- Line 71: For this discussion on structure and fragmentation, additional reference should be cited (Lambe et al., 2012).
   Reference has been added.
- 5. Line 107: The authors assert that acid-catalyzed reactive uptake has not been reported for the oxidation of alkanes. This is not true. Atkinson, Lim, and Ziemann have shown that alkane oxidation leading to 1,4-hydroxy carbonyls convert to cyclic hemiacetals in an acid-catalyzed multi-phase process (Dibble, 2007; Atkinson et al., 2008; Lim and Ziemann, 2009a, 2009b). Schilling Fahnestock et al., 2015 also report the effect of acidity on SOA formation from C12 alkanes.

This statement has been removed from the revised version of the manuscript.

references, which have already characterized both sites, are cited.

6. Lines 123-124: Can the authors give more background on these two sites to orient the reader also with the motivation/purpose of this study? What types of sites are these—urban with what type of emissions profiles and surrounded by vegetation, etc.? As it is highlighted in the abstract, the motivation for our study was to demonstrate the formation of OSs from the oxidation of alkanes, which has been inferred from previous field studies. Description of both sites is presented in the experimental section and

The following sentences have been added:

Lines 185-188: "As stipulated previously at both urban sites, anthropogenic activities (e.g., vehicular exhaust, industrial sources, cooking, etc.) likely dominated the organic aerosol mass fraction of  $PM_{2.5}$  (Stone et al., 2010; Hayes et al., 2013). In addition, Gentner et al. (2012) have reported significant emission of long-chain alkanes during the CalNex field study."

7. Line 222-223: It would be helpful to label the 1,3-dodecanone sulfate in Figure 1 to aid the reader. The authors should be careful with their naming convention here (i.e. 1,3-), as this particular isomer certainly is not the only potential isomer and is not the only isomer specified in Yee et al., 2012 and Yee et al, 2013. The OS has been labeled in Figure 1. We agree with referee #3 that other isomers are present since we identified at least 3 isomers as mentioned on line 236 and reported in Table S1.

The OS has not been labeled accordingly in Figure 1. Please make the change so it is easier for reader to follow in-text and figure.

The OS is now labeled (as 6-dodecanone-8-sulfate) in the new version of the manuscript.

The sentence has been modified on lines 239-242 to:

"Kwok and Atkinson (1995) have reported that OH oxidation of long-chain alkanes preferentially occurred at an internal carbon and thus multiple isomers may be proposed. Based on Yee et al. (2012; 2013) one isomer may be, however, assigned as 6-dodecanone-4-sulfate."

As currently drawn in Figure 1 for OS-279, the naming should be changed from 6dodecanone-4-sulfate to 6-dodecanone-8-sulfate, no? Also, the particular isomer chosen here should just be an example isomer for the purpose of drawing the mechanism. It is unnecessary and would be incorrect to cite Yee et al., (2012; 2013) for this isomer as that work also does not "assign" a specific isomer or isolate a specific isomer from the measurements.

The sentence should be modified to:

"Kwok and Atkinson (1995) have reported that OH oxidation of long-chain alkanes preferentially occurred at an internal carbon and thus multiple isomers may be proposed. One such isomer, 6-dodecanone-8-sulfate, is drawn in Figure 1 to represent a proposed structure for OS-279."

We agreed and we have revised the sentence as suggested (Lines 239-242).

8. Lines 226-230: The reference cited, Ruehl et al., 2013, is improperly used here. Ruehl et al., 2013 describes the heterogenous oxidation of octacosane and finds a strong preference for OH attack at the terminal carbons. The current work, however, is gas-phase oxidation, so the specificity of the isomers as listed in lines 228-229 should be rethought. Further, the naming convention for these isomers are inconsistent with the naming convention in line 222-223. It seems as though the 1, 3-dodecanone denotes the 1 position as the ketone,

whereas here the reference to 2, 4-, 3,5-, and 4,6- and other isomers suggests the 1 position is likely the carbon at the end of the dodecane chain.

Sentence has been removed and naming of the molecule is now consistent with line 242. As discussed by Kwok and Atkinson (1995), reaction occurs preferentially on internal carbons and the sentence has been changed as proposed in the previous point.

9. Lines 234-237: This is a poorly worded sentence. It is unclear in relation to the context of the current work, and there is imprecise use of citations. Hydroperoxides can undergo further oxidation by reaction with OH, but to generate alkoxy radicals from hydroperoxides, that would likely include photolysis. The authors need to address the extent of photolysis in the experiments then. Or are the authors referring to reactions of RO2 + NO to generate alkoxy radicals? If so, then the authors need to address the extent of RO2 + NO occurring in the experiments. If the former, rewrite as, "First-generation hydroperoxides can undergo further oxidation by reaction with OH to form low-volatility, more highly oxidized products, or can be photolyzed to alkoxy radicals (RO) to form more highly volatile products." The use of Carasquillo et al., 2014 here is inappropriate to discuss the oxidation of hydroperoxides as written. Carasquillo et al., 2014 describe the fate of differing alkoxy radical structures and how it relates to the mechanism proposed in Figure 1.

We thank the reviewer for its comments, sentences have been changed to simplify and clarify this paragraph.

Insertion should be made to the sentences below as indicated in red. The insertion has been made as suggested (Line 253).

Lines 249-255: "First-generation hydroperoxides  $(C_{12}H_{26}O_2)$  can undergo further oxidation by reaction with OH to form either more highly oxidized products, such as dihydroperoxides  $(C_{12}H_{26}O_4)$ , or semi-volatile products  $(C_{12}H_{24}O)$  (Yee et al., 2012). In addition, hydroperoxides can be photolyzed to alkoxy radicals (RO) undergoing additional transformation to form more highly oxidized products. Low-volatility products could then condense onto sulfate aerosols and undergo further heterogeneous reactions (Schilling Fahnestock et al., 2015) leading to OSs as discussed below." 10. Lines 237-243: The authors can cite Raff and Finlayson-Pitts, 2010 for IPN as an OH radical source, but it cannot be cited to fully account for the chemical conditions (i.e. the fate of RO2) in the current experiments without considering the differences between their experiment and that of Raff and Finlayson-Pitts, 2010. The authors should report NOx levels in these experiments to verify the claim that RO2 + NO reactions are minimal. Also, how is O3 formed in these experiments? The authors need to calculate (considering the relatively high levels of initial hydrocarbon), the relative fate of RO2 between reaction with RO2, HO2, and NO. The proposal that OS-279 stems from hydroperoxide species in Figure 1 seems least inappropriate if RO2 fate is really dominated by reaction with RO2 and/or NO.

In order to provide additional support for the proposed mechanisms, total organic peroxide measurements have been performed. These results which are now reported in Table 1 reveal that organic peroxides (including hydroperoxides) could contribute up to  $\sim$  28 % (on average) of the SOA mass formed from the photooxidation of the precursors used in this work. These measurements highlight the significant presence of organic peroxides and/or hydroperoxides in aerosol and thus support the proposed mechanisms. In addition to the organic peroxide measurements, concentrations of O3 and NO were also added in Table 1 to confirm the low-NO conditions cited in this work.

The description of the organic peroxide measurements has been added, lines 211-219, revised manuscript:

**"Total Organic Peroxide Analysis.** The total organic peroxides in the SOA were quantified using an iodometric-spectrophotometric method adapted from Docherty et al. (2005). As described in Surratt et al. (2006), the method employed in this work differs in the choice of extraction solvent: we used a 50:50 (v/v) mixture of methanol and ethyl acetate, rather than pure ethyl acetate. Calibrations and measurements were performed at 470 nm using a Hitachi U-3300 dual beam spectrophotometer. Benzoyl peroxide was used as the standard for quantification of organic peroxides formed from alkane oxidations. The molar absorptivity measured from the calibration curve was ~ 825, which is in excellent agreement with reported values (Docherty et al., 2005; Surratt et al., 2006)."

A discussion of the results of the organic peroxide measurements has been added, lines 261-265, revised version:

"In addition, total organic peroxide aerosol concentrations, presented in Table 1, reveal that organic peroxides account (on average) for 28 % of the SOA mass measured in the different experiments in support of a significant contribution of  $RO_2 + RO_2/HO_2$  and/or  $RO_2$  autoxidation to SOA formation from alkane oxidations."

In our experiment we did not add NO (prior to IPN injection) and background NO levels were measured near the detection limit of the NOx monitor (i.e., 1 ppb). After IPN injection a significant increase of O3 was observed in all experiments (as described lines 258-260) and NO concentration dropped below 1 ppb. We agree that NO2 is photolyzed in these systems and NO is recycled. However, under the conditions described here (and in the article on lines 258-260 and in Table 1) most of the NO is expected to react with O3. Rate constants for the RO2 + NO and NO + O3 reactions were determined to be  $4.7 \times 10^{-12}$ cm3 molecule-1 s-1 ( $k_l$ ) (suggested by MCM; Ehn et al., 2014) and 1.8 × 10-14 cm3 molecule-1 s-1 (Atkinson et al., 2000; IUPAC) ( $k_2$ ), respectively. Gratien et al. (2010, ES&T, 44, 8150-8155) have calculated the OH radical concentration to be  $5 \times 10^6$ molecule cm-3 from the photolysis of 5 ppm of isopropyl nitrite (IPN). In our experiments, 0.1-0.25 ppm of IPN was injected into the chambers. Using similar OH radical concentration to Gratien et al. (2010, ES&T) as an upper limit, RO2 concentration could be estimated to be at ppt levels. Therefore, under the conditions of our study and assuming an O3 concentration of 0.5 ppm, NO would react predominantly with O3:  $(k_2[O_3])$  $[NO])/(k_1[RO_2][NO]) > 350.$

**Lines 258-260 have been modified:**

"Although  $RO_2$  radicals could also react with NO formed by either IPN or  $NO_2$  photolysis, formation of ozone under chamber conditions (0.3-0.6 ppm, depending on the concentration of IPN injected, Table 1) would rapidly quench NO (Atkinson et al., 2000). Therefore,  $RO_2$  + NO reactions are not expected to be significant."

Finally, the photooxidation of dodecane has been also investigated using an additional injection of NO (200 ppb) prior IPN injection. NO concentration dropped below the ppb level in less than 1 hour and OS concentrations were significantly reduced (factor of 3-4) compared to other experiments, confirming that NO concentration does have an impact on OS formation.

One sentence has been added to describe this experiment, lines 274-277:

"It should be mentioned that photooxidation of dodecane has also been investigated using an additional injection of NO (200 ppb) prior IPN injection. In this experiment SOA formation was significantly reduced as well as the OS concentrations (factor of 3-4), confirming that NO strongly impacts the formation of OS, such as OS-279."

These clarifications on the chemical regimes taking place during the experiments greatly enhance the quality of the manuscript and better substantiate the mechanisms proposed and conclusions reached. Thank you.

11. Table 2: The authors never describe the origin of the C7 (OS-209) and C9 (OS-237) organosulfates observed in the dodecane system and also observed in the ambient samples. This is another indication that fragmentation pathways are at play, potentially through RO2 + NO reactions in the system. The authors need to be careful in explaining the fate of the RO2 radical in their experiments and whether the ambient observation of these OSs can really be attributed to dodecane chemistry in the atmosphere when they may clearly originate from other precursors. The authors need to also describe the potential influence of monoterpenes at the sites they have taken samples from to preclude OS origin from biogenic precursors, as they say themselves that C10 monoterpene OSs are isomeric to some proposed in the C10 alkane systems. How good is RT matching/SICs for confirming that the laboratory generated OSs are really the same as those in ambient data? What measurements in these locations suggest that decalin, cylodecane, and dodecane are prevalent here?

As mentioned in lines 229-231, low abundances of OS-209 and OS-237 precluded acquisition of high-resolution  $MS^2$  data, and thus structures have not been proposed for these parent ions. Without any compositional information, reaction pathways cannot be discussed. As shown by Yee et al. (2012), hydroperoxides can be photolyzed to RO radicals, which fragment to smaller carbonyls. The potential formation of RO radicals from photolysis of hydroperoxides has been added in the revised version of the manuscript.

Isobaric compounds could likely be formed in the atmosphere, however, structures would be significantly different and isomers could be distinguished in most of the cases. Although we cannot completely rule out co-elution of some isobars, Figure 7 illustrates the most likely typical situation, in which isobars from the photooxidation of cyclodecane and decalin have different retention times (R.T.), allowing differentiation. OSs are known to form from the oxidation of monoterpenes and several isobaric OSs have been identified: OS-249, -251, -267, -279 and -326. Structures proposed in previous work are significantly different from structures proposed in this work and thus should be separated by liquid chromatography. We have analyzed ambient filters collected during SOAS campaign in rural areas (Centerville, Alabama, US) and find that the R.T.s of monoterpene-derived OSs are different from those of the OSs identified from the oxidation of the alkanes studied in this work.

For example:

OS-249: from monoterpenes: 10.3 min; cylodecane: 8.5/9.3 OS-279: from monoterpenes: 6.2 min; cylodecane: 5.8/6.8

Thank you for this clarification.

We do not have access to potential collocated measurements during both field measurements, however, results proposed by Gentner et al (2012) tend to support significant emissions of long-chain alkanes in California and especially during CalNex.

The following sentences have been added:

Lines 185-188:

"As stipulated previously at both urban sites, anthropogenic activities (e.g., vehicular exhaust, industrial sources, cooking, etc.) likely dominated the organic aerosol mass fraction of  $PM_{2.5}$  (Stone et al., 2010; Hayes et al., 2013). In addition, Gentner et al. (2012) have reported significant emission of long-chain alkanes during the CalNex field study."

11. Lines 246-247: The citation of Claeys et al., 2004 is inappropriate here. The authors propose that "heterogenous chemistry of gas-phase organic peroxide" is a mechanism for OS and tetrol formation, citing Claeys et al., 2004. Yet, Claeys et al., 2004 state, "The mechanism we suggest, reaction with hydrogen peroxide under acidic conditions in the aerosol liquid phase...," which is not consistent with the heterogeneous mechanism proposed in the current work and in Riva et al., 2015b. The difference in humidity should also affect the distribution of hydroperoxide compounds in the gas/particle phase. The

authors should address this in the context of mechanistic explanations for their observations.

We agree with reviewer that Claeys et al. is an inappropriate reference to be used here and we have removed it. We have shown in previous work and in another manuscript currently under review that organic peroxides could lead to OSs and polyols from aerosol-phase acid-catalyzed reactions. It is not clear how the RH could directly impact the distribution of the hydroperoxides as suggested by the reviewer. However, we have reported that the liquid water content of the aerosol plays an important role, but the acidity has a stronger impact on OS formations.

Thank you for clarifying.

- Lines 256: Several citations should be added here. Include reference to works by Lim and Ziemann, Lambe et al., Yee et al., Loza et al., Tkacik et al.
   References have been added.
- 13. Lines 284-286: The mechanism described in text corresponds with the pathway in Figure S8, pathway c, not pathway a. Authors should rewrite these lines to describe pathway a. It also becomes clear here that the authors are not consistent with description of the chemistry proceeding in the chamber. In Figure S8, the fate of RO2 is initially reaction with RO2, but then in pathway a, it shifts to RO2+HO2. The selected pathways seem arbitrary to explain the proposed structure in Figure 2a.

Figure S8 describes sequential reactions via  $RO_2$  leading to a ring opened ketoperoxy transient. Three branching reactions are available to transient: reaction with HO2 leading to the structure proposed for OS-265, pathway **a**; further reaction with an RO2 species leading to the structure proposed for OS-265 and OS-281, pathway **b**; or isomerization and reaction with O2 eventually leading to OS-281 and OS-297, pathway **c**. Figure S8 does not therefore represent a "shift" in chemistry, but branching reactions leading to three observed product ions. We note that the RO radical precursor to pathways a, b and c may also result from an  $RO_2 + HO_2$  reaction (Kautzman et al., 2010; Birdsall et al., 2011). Since it is not possible to distinguish whether RO originates from  $RO_2 + RO_2$  or  $RO_2 + HO_2$  reaction has been added to all mechanisms.

This paragraph has been changed to be consistent with proposed mechanism.

Lines 310-314: "A scheme leading to the structure proposed in Figure 2a is based on the cleavage of the  $C_1-C_2$  decalin bond, followed by reaction with a second  $O_2$  molecule and  $HO_2$  leads to a terminal carbonyl hydroperoxide ( $C_{10}H_{18}O_3$ ) (Yee et al., 2013).  $C_{10}H_{18}O_3$  could then further react with OH radicals and lead to an epoxide and sulfate ester by reactive uptake/heterogeneous chemistry (Paulot et al., 2009)."

14. Line 290: It is unclear whether the analytical technique is sufficient for seeing hydroperoxide moieties on molecules as they are included in the proposed structures. Were hydroperoxide standards such as t-buytylhydroperoxide or cumene hydroperoxide run using this method to verify that the hydroperoxide moiety can be retained on the column? Or is there something about the organosulfates that allow for this? The authors should address this in the experimental methods section as well.

Such compounds can be retained on the column used in this project and we have demonstrated this for another project with a synthetic isoprene hydroxyhydroperoxide standard (ISOPOOH). As an example, please see the chromatogram below:

Moreover, Witkowski and Gierczak (2012) have recently developed a method to quantify hydroperoxide compounds formed from the ozonolysis of cyclohexene. The authors used a column similar ( $C_{18}$ ) to that used in the present work. Electrospray ionization mass spectrometry operated in negative mode (Cech and Enke, 2001; Witkowski and Gierczak, 2012) is not highly sensitive to detection of non-acidic compounds, such as pure hydroperoxides or alcohols. However, LC/ESI(-)-MS provides excellent sensitivity for multifunctional compounds (like hydroperoxides and alcohols) containing the OS functional group, since the OS functional group yields an intense [M - H]- ion, as reported in many studies (Surratt et al., 2008; Kristensen et al., 2011; Kundu et al., 2013; Hansen et al., 2014).

As mentioned line 340-342, we expect to detect the presence of hydroxyl or hydroperoxide functional groups when the OS group is present in the multifunctional compounds analyzed by LC/ESI(-)-MS.

The following sentences has been added:

Lines 340-346: "As a result, the presence of hydroperoxide and/or hydroxyl substituents is expected in order to satisfy the molecular formulas obtained by the accurate mass measurement. Although ESI-MS in the negative ion mode is not sensitive to multifunctional hydroperoxides and alcohols (Cech and Enke, 2001; Witkowski and Gierczak, 2012), this technique is highly sensitive to hydroperoxides and alcohols which also contain OS groups and give  $[M - H]^-$  ions (Surratt et al., 2008; Kristensen et al., 2011; Kundu et al., 2013; Hansen et al., 2014)."

Thank you for this excellent clarification.

15. Lines 295-296: The work of Yee et al., 2013 and Schilling Fahnestock et al., 2015 do not test decalin, so they should not be cited here to support proposed formation of a 1-hydroperoxy radical in the decalin system used in the current work. While the mechanisms laid out in Atkinson, 2000 can apply here, as worded it seems as if the authors are proposing the particular alkoxyl reference.

In Yee et al. 2013 they studied the oxidation of hexylcyclohexane and cyclododecane and they proposed (Figure 1, sidebar, Yee et al. 2013) a ring scission and formation of a terminal carbonyl hydroperoxide as proposed in Figure 3. We do not claim that both studies have investigated the oxidation of decalin but have used the analogous ring scission sequence to explain our products. To avoid confusion regarding the content of the Yee citation, we have moved the citation in the text to follow the description of ring scission. Sentence has been modified in the revised manuscript.

Lines 348-352: "Under low-NOx conditions, abstraction of a proton  $\alpha$  to the ring scission of decalin followed by reaction with O2 leads to the 1-hydroperoxy radical, which in turn can react with another RO2 radical to yield the corresponding alkoxyl radical (C10H17O') (Atkinson, 2000). Cleavage of the C1-C2 decalin bond, followed by reaction with a second O2 molecule and HO2 leads to a terminal carbonyl hydroperoxide (C10H18O3) (Yee et al., 2013)." Please revise Lines 348-352 to be more precise as below. The reference placement is still confusing, implying that specific C10 compounds were observed in the citations provided, though they are just being used for analogous mechanistic pathways.

Lines 348-352: "Following analogous mechanisms for low-NOx conditions (Atkinson, 2000; Yee et al., 2013), abstraction of a proton  $\alpha$  to the ring scission of decalin followed by reaction with O2 leads to the 1-hydroperoxy radical, which in turn can react with another RO2 radical to yield the corresponding alkoxyl radical (C10H17O). Cleavage of the C1-C2 decalin bond, followed by reaction with a second O2 molecule and HO2 leads to a terminal carbonyl hydroperoxide (C10H18O3)."

The sentences have been revised as suggested (Lines 350-355).

17. Line 298/Figure 3: To guide the reader it would be beneficial to update the mechanism in Figure 3 with the same label of tCARBROOH next to intermediate the authors are referring to

We have now added the formula of the different primary products. Please add the labels to Figure 3 for clarity.

As presented in the revised version we have changed the naming of the different compounds and choose to use the formula ( $C_{10}H_{18}O_3$ ) instead of shorthand chemical formula (c.f. tCARBROOH) for clarity. tCARBROOH is not longer used in the revised manuscript and formula of the different primary products have been already added to the Figures in the revised version.

18. Lines 298/Figure 3: Do the authors see evidence of an analogous product in the case of decalin, as the OS-279 that was observed in the dodecane case? The proposed mechanism of carbonyl hydroperoxide heterogenous reactive uptake followed by OS formation should also be considered for the decalin tCARBROOH as well and supported by the measurements/compared on the basis of volatility differences due to carbon number/ring structure and the impact of reactive uptake versus partitioning to the particle phase.

We thank the reviewer for its comment. We have revised the pathways proposed for decalin oxidation products OS-265; -267; -269 and -285 and cyclodecane oxidation

products OS-249; -251; -265 and -267 to include reactive uptake of the hydroperoxide on wet acidic aerosols.

The appropriate mechanisms have been updated as well as the manuscript:

Lines 314-316: "OS-265 ( $C_{10}H_{17}O_6S^-$ ) could also arise from the acid-catalyzed perhydrolysis of the hydroperoxide ( $C_{10}H_{18}O_4$ ) generated from the reaction of  $C_{10}H_{17}O_4^+$  +  $HO_2$  (Figure S8, pathway b)."

Lines 319-325: "The pathway proposed in Figure S8 pathway **b** is based on gas-phase oxidation of a 4-(cyclohexan-2-one)but-1-yl radical followed by reaction with  $O_2$  and a 1,5-H shift (Crounse et al., 2011; Orlando and Tyndall, 2012) and lead to a  $C_{10}$ -carbonyl-hydroxyhydroperoxide ( $C_{10}H_{18}O_4$ ).  $C_{10}H_{18}O_4$  could then further react with OH radical and by elimination of OH lead to an epoxide (Figure S8, pathway **b**). In addition, OS-281 could arise from acid-catalyzed perhydrolysis of  $C_{10}$ -carbonyl dihydroperoxides ( $C_{10}H_{18}O_5$ ) as proposed in Figure S8, pathway **c**."

Lines 357-374: "C9H17O4' can react via pathway **a** (Figure 3) through a 1,6-H shift (Crounse et al., 2011; Orlando and Tyndall, 2012) followed by elimination of OH resulting in a formation of an epoxide analogous to the formation of isoprene epoxydiol (IEPOX) (Paulot et al., 2009; Mael et al., 2015). The epoxide can then undergo acid-catalyzed ring opening to give OS-269 ( $C_9H_{17}O_7S^-$ ). The MS2 spectrum of OS-285 ( $C_9H_{17}O_8S^-$ ; Figure S5) shows product ions corresponding to HSO3-, HSO4- and loss of neutral SO3, in accord with a sulfate ester  $\beta$  to a labile proton, but yields no further structural information. The structure proposed for OS-285 is based on the formation of reaction of the hydroperoxyperoxyl radical intermediate in pathway **b** with  $RO_2$  followed by a 1,4-H shift (Rissanen et al., 2015) and addition of  $O_2$  to give a hydroxyhydroperoxyperoxyl radical ( $C_9H_{17}O_5$ ).  $C_9H_{17}O_5$  could then lead to an epoxide by isomerization (Iinuma et al., 2009; Surratt et al., 2010; Jacobs et al., 2013; Mael et al., 2015) and form OS-285.  $C_9H_{17}O_5$  could also react with HO2 and form the corresponding  $C_9$ -hydroxydihydroperoxide ( $C_9H_{18}O_5$ ), which could then undergo heterogeneous reaction and lead to OS-269 (Figure 3, pathway b). Finally, a C9-carbonyl hydroperoxide ( $C_9H_{16}O_3$ ) could also be formed from the RO +  $O_2$  reaction (Figure 3, pathway c), which could then further react with OH radicals and lead to a C9-carbonyl

dihydroperoxide ( $C_9H_{16}O_5$ ). Hence,  $C_9H_{16}O_5$  could form OS-267 ( $C_9H_{15}O_7S^-$ ) from heterogeneous reaction on acidic aerosols."

This analysis strengthens the findings of the work by making more clear in the text selfconsistent mechanisms.

19. Lines 299-301: (e.g. remote areas). Please also refer to and reference (Peeters et al., 2009; Crounse et al., 2011; Orlando and Tyndall, 2012). The authors need to justify why the basis of their proposed mechanisms for OS formation rely on this pathway when (e.g. Figures 3, 4, and S8). Is it RO2 + RO2, RO2 + HO2, RO2 + NO, RO2 isomerization? The suggested references have been added to the revised manuscript.

As discussed previously, different regime of RO2 radicals could exist, either terminal (RO2 + HO2; RO2 + RO2; RO2 + NO) or autooxidation reactions. In this study, we do not claim to propose all chemical pathways from the oxidation of the alkanes are examined. In most of the mechanisms we have considered the different potential RO2 reactions (RO2 + RO2; RO2 + HO2; RO2 + RO2, and RO2 autooxidation), which could lead to the identified OSs through multiphase chemistry of the products shown in the tentatively proposed mechanisms. RO radicals might have formed for other minor chemical channels, such as ROOH +  $h\nu$  or RO2 + NO, which were not initially included in the manuscript. It is important to note that these potential reactions, which are now included in the manuscript, do not change the different mechanisms tentatively proposed in this study. In addition we have proposed reaction sequences based on known/reported reactions that will lead to products consistent with the mass spectrometric data. This is the same approach used by other investigators, such as Yee et al. (2013).

20. Lines 301-305: The authors should address the extent of photolysis reactions affecting the fate of the proposed hydroperoxides and aldehydes in the system.

Potential photolysis reactions are now discussed in the revised manuscript. We have incorporated the potential photolysis of hydroperoxides leading to RO radicals and also the photolysis of the aldehyde proposed in Figure 3, which could lead to the  $RO_2$  radical  $(C_{10}H_{17}O_5)$ .

This version of the manuscript does not show the changes made to Figure 3 including photolysis.

We have revised Figure 3 as requested.

Lines 353-357: "The aldehydic intermediate in the sequence following  $C_1$ - $C_2$  ring scission may be oxidized to the corresponding acyl radical either by photolysis (Wang et al., 2006) or by H-abstraction (Kwok and Atkinson 1995) followed by addition of  $O_2$ , reaction with  $RO_2$  or  $HO_2$  and decarboxylation of the resulting acyl-oxy radical (R(O)O) (Chacon-Madrid et al., 2013) to a hydroperoxyperoxy radical ( $C_9H_{17}O_4^*$ )."

21. Line 306: "previously unreported" is unclear. Do the authors mean previously unreported in ambient data or previously unreported from similar experiments? OS-267 has been identified in previous smog chamber experiments. Sentence has been modified in the revised manuscript.

Lines 370-374: "Finally, a C9-carbonyl hydroperoxide ( $C_9H_{16}O_3$ ) could also be formed from the RO + O2 reaction (Figure 3, pathway c), which could then further react with OH radicals and lead to a C9-carbonyl dihydroperoxide ( $C_9H_{16}O_5$ ). Hence,  $C_9H_{16}O_5$  could form OS-267 ( $C_9H_{15}O_7S^-$ ) from heterogeneous reaction on acidic aerosols."

- 22. Line 311: As worded, OS-267, is proposed to originate from further oxidation of OS-269, but the arrows drawn in Figure 3 are inconsistent suggesting origin from the epoxide.We have corrected the revised manuscript.
- 23. Line 327/Figure 4: The description of Ring cleavage of the C10 alkoxy radical is not consistent with the "ISO"/isomerization descriptor in Figure 4. Please clarify that pathway.

ISO descriptor in Figure 4 (*pathway a*) indicates the isomerization of the  $RO_2$  formed from the ring cleavage and lead to hydroperoxide functional group.

Sentences have been added to better discuss this pathway:

Lines 382-388: "The salient features of pathway a include oxidation of the  $RO_2$  to 2decalinone, formation of a  $C_{10}$  alkoxy radical followed by ring cleavage of the  $C_9-C_{10}$ decalin bond and further  $RO_2$  isomerization (1,8-H shift) leading to a 4-(carboxy cyclohexyl)-1-hydroperoxybut-2-yl radical via  $RO_2$  chemistry. Although considered as a minor reaction pathway (Crounse et al., 2013), the acyloxy radical could lead to the epoxide from the isomerization of the  $O_2$  adduct (Paulot et al., 2009; Yao et al., 2014; Zhang et al., 2015). Further acid-catalyzed ring opening of the epoxide leads to OS-295  $(C_{10}H_{15}O_8S^-)$ ."

- 24. Line 340: Figure 1d does not exist. Clarify the reference.The reference was Figure 2d and not 1d, the text has been appropriately modified.
- 25. Lines 343-345: Sentence is awkward beginning with "Pathway c", and from what figure? Clarify that it is Figure 4. Again, citation of Atkinson, 2000 seems inappropriate as the sentence is written.

Sentence has been changed, pathway **c** referred to Figure 4. Citation of Atkinson (2000) was used to support formation of an organonitrate from  $RO_2$  + NO reaction, since it is a common reaction as discussed in the Atkinson's review.

Lines 402-408: "Although  $RO_2 + NO$  reactions are expected to be minor under the conditions used in this work (i.e. NO < 1 ppb, formation of RO radicals or organonitrates cannot be ruled out. Indeed, Ehn et al. (2014) have demonstrated that NO reactions could be competitive at ppb levels. Under our experimental conditions  $RO_2 + NO$ ,  $RO_2 + HO_2$  and  $RO_2$  autoxidation are possible. Therefore, the parent ion at m/z 326 could arise from the reaction of the decalin-2-peroxy radical with NO to form decalin-2-nitrate  $(C_{10}H_{17}NO_3)$  with subsequent reactions shown in Figure 4, pathway c."

26. Lines 343-345/Figure 4, pathway c: Here the authors propose that RO2 + NO chemistry is occurring to form a nitrate containing OS. This contradicts the authors' earlier statement in lines 237-243 stating that the RO2 + NO reactions are not significant in their experimental setup. The authors need to handle in more detail the fate of RO2 under the unclear experimental conditions.

As demonstrated and discussed by Ehn et al. (2014) at ppb levels of NO (which is even higher than the conditions of our study) a competition exists between  $RO_2 + NO$ ;  $RO_2 + HO_2$  and  $RO_2$  autoxidation reactions. The Ehn et al. (2014) study demonstrates that ELVOC, even though reduced, are still formed at NO concentrations greater than few ppb. It is important to point out that the concentrations of VOCs used in this work could also lead to  $RO_2 + RO_2$  chemistry. Therefore, not only one  $RO_2$  reaction could occur and the different  $RO_2$  reactions have to be considered, which has been done in this study.

 $O_3$  and NO concentrations are provided in the revised manuscript (Table 1). In addition a paragraph has been added describing the fate of  $RO_2$ .

Lines 404-406: "Indeed, Ehn et al. (2014) have demonstrated that NO reactions could be competitive at ppb levels. Under our experimental conditions  $RO_2 + NO$ ,  $RO_2 + HO_2/RO_2$  and  $RO_2$  autoxidation are possible."

27. Lines 349-364: Why is discussion of OS-281 and OS-297 featured here, when discussion of OS- 265 is discussed near the beginning of Section 3.2? Since they are all referenced in Figure S8, their chemistry should be discussed together from the same mechanistic precursors.

We chose to describe the formation and tentative structural assignments of ions observed on ambient filters at the beginning of the discussion, as explained in the manuscript on lines 293 to 295 ("Figures 2 and S2 present MS2 spectra and fragmentation schemes of selected parent ions at m/z 265.0749 (OS-265), 269.0696 (OS-269), 295.0494 (OS-295) and 326.0554 (OS-326). MS2 spectra and fragmentation schemes of other OSs are reported in Figure S3-S7. The selected OSs were, as described in the next section, quantified and characterized in the fine urban aerosol samples.").

However, section 3.2 has been reorganized as requested by the reviewer #3 and formation pathways of OS-265, -281 and -297 are included in the same paragraph. Thank you. This is more clear.

28. Section 3.2: Authors should clarify the main mechanistic differences and relative importance between Figure S8, Figure 3, and Figure 4, and the flow of products to be discussed at the beginning of Section 3.2. Currently as written, the flow of Section 3.2 is very arbitrary when choosing different OS products to discuss. We decided to separate the different reaction pathways for clarity since it would not have

been clear and quite difficult/confusing to propose in one figure the formation pathways of all OSs. The different reaction pathways are separated based on OSs that are generated

from branching reactions of a common transient. This section has been modified as discussed in the previous point. The importance of the proposed pathways cannot be evaluated based on this study and this was not the study objective.

We have added a sentence to clarify this point.

Lines 297-299: "The different reaction pathways presented below, are separated based on OSs that are generated from branching reactions of a common transient." Great clarification. Thank you.

29. Lines 377-380: Incorrect use of citations here. Yee et al., 2012 do not propose RO2 + RO2 chemistry and therefore a "precedent" has not been established. The authors should not be citing Atkinson, 2000; Yee et al., 2012 and Raff and Finlayson-Pitts, 2010 to speak for the experimental conditions in the current work. The mechanism of RO2 + RO2→RO can be supported by work on general atmospheric chemistry mechanisms including Atkinson, 2000 and many other works, and so if this mechanistic pathway is to be cited, than many other works should be cited as well.

We agree with reviewer that Yee et al. is an inappropriate reference and have removed it. General references on atmospheric chemistry have been added: Atkinson and Arey (2003, Chem Rev, 103, 4605-4638) and Ziemann and Atkinson (2012, Chem Soc. Rev., 41, 6582-6605).

30. Line 379: Inconsistent citation here compared to line 300 for similar mechanistic argument.We have cited Ehn et al., 2014 and Jokinen et al., 2014 and Mentel et al., 2015 for both

transformations.

31. Figure S14: Why do the authors propose in the case of cyclodecane formation of the hydroperoxide from RO2 + HO2 pathways and subsequent chemistry thereof, but not in the case of decalin in any of Figures 3, 4, and S8? Further, Figure S14 outlines a mechanism from further reaction of the hydroperoxide to get to an epoxide that then enters the particle phase to produce OS-251 and OS-249. This seems like a plausible analogous mechanism to propose for the case of dodecane rather than reactive uptake of a carbonyl hydroperoxide. Why do the authors propose different mechanisms between dodecane and cyclodecane to generate the similar analogs (OS-279, OS-249)?

As mentioned in the point #18 above, cyclodecane-derived OSs might be formed from the heterogeneous chemistry of hydroperoxides. We have added these different pathways in the different Figures as well as in the manuscript. Please note that we have now combined Figures S14 and S15 to present the OS formation from cyclodecane in one Figure.

These changes make the mechanisms proposed for all systems more consistent, though please make it also clear in the discussion on dodecane why analogous pathways (epoxide formation and subsequent uptake, isomerization, etc.) as proposed in the C10 system are also likely/unlikely for the conditions of the experiment. It would be good to discuss the DBE determination in a non-epoxide route to OS-279. It is not mentioned in text of section 3.1 or in Figure 1

We have added a short discussion as suggested by the reviewer.

Lines 272-274: "OS-279 generated from the reactive uptake of the corresponding epoxide  $(C_{12}H_{24}O)$  has been considered but the composition of OS-279 (1 DBE) is inconsistent with reactive uptake of an epoxide."

Lines 428-435: "The formation of compounds such as cyclodecanone  $(C_{10}H_{18}O)$ , cyclodecane hydroperoxide  $(C_{10}H_{20}O_2)$  or cyclodecane hydroxyhydroperoxide  $(C_{10}H_{20}O_3)$  are proposed as intermediate products leading to epoxy-compounds after additional oxidation/isomerization processes, as presented in Figure S14. In addition  $C_{10}H_{20}O_3$ , cyclodecane hydroperoxide ketone  $(C_{10}H_{18}O_3)$  and cyclodecane hydroxyoxohydroperoxide  $(C_{10}H_{18}O_4)$ , proposed as intermediate products, could condense onto wet acidic aerosols and lead to the corresponding OSs through acid-catalyzed perhydrolysis reactions (Figure S14)."

As described above we proposed different fates for the  $RO_2$  radicals:  $RO_2 + RO_2$ ,  $RO_2 + HO_2$  and  $RO_2$  autoxidation reactions in the different schemes proposed for the photooxidation of decalin and cyclodecane.  $RO_2 + HO_2$  reactions are also proposed in the case of the photooxidation of decalin, which likely explain the formations of hydroperoxides as discussed above.

32. Section 3.4: This section is weak and little effort is made to really describe the chemical differences between the systems to interpret the findings. There should be comparisons of

vapor pressures of the precursors and carbon numbers and discussion of previously published yields from these compounds to support the discussion. How does quantification using the available OS surrogate standards potentially affect the OS quantification across these systems/factoring in different sensitivities?

The objective of this study is to establish that OSs may be products of the photooxidation of anthropogenic precursors, such as the alkanes examined here, and thus to demonstrate the relevance of this chemistry observations of aliphatic OSs in urban areas (Mao et al., 2014; Tao et al., 2014). Since the reaction pathways leading to the products observed in this study and in ambient samples are tentative, we feel that discussion at the level of thermodynamics is not justified and have deleted Section 3.4.

33. Line 400: The authors claim that "the presence of acidic aerosols significantly increase OS formation in most cases". However, is this just an effect of using an atomized solution with more sulfate (0.06M ammonium sulfate + 0.06M sulfuric acid) in the acidic case versus only 0.06M ammonium sulfate in the non-acidified case? It may be a concerted effect of more available sulfate in the "acidic" case as well as acidity.

It has been demonstrated in previous studies (cited references) that acidity rather than the concentration of sulfate is a key parameter in the formation of OS. Chan et al. (2011) demonstrate that the formation of OSs from the oxidation of  $\beta$ -caryophyllene is directly correlated with the aerosol acidity ([H+]).

Understood, though please address this point directly in the text in anticipation of the casual reader in the field who may not have as much knowledge of the literature specifically addressing the role and dynamics of sulfate concentration vs aerosol acidity (which in stated reference is still an indirect measurement of the acidity at which the OS actually formed, but an accepted proxy.) As authors state in lines 455-463 and cited literature, there are many ways to form OS, so the concentration of sulfate as a precursor to sulfate anion radical should be a factor in the chemistry.

We have added few sentences (Lines 453-459) as mentioned previously in this review (last paragraph of the introduction part).

34. Line 484: The authors return to claim that the experiments are conducted under dominant "RO2/HO2" chemistry—this is contradictory to the formation of OS-326 containing a nitrate group. This issue has been discussed previously in response to reviewer comments # 10, 13, 19, 26, and 31 above.

35. Lines 484-486: Enhancement of OS due to acidified ammonium sulfate seed needs to be addressed with regard to the effect of just having introduced more sulfate into the experiments compared to the non-acidified case. See earlier Major Comment, 33. As discussed in response to Reviewer comment # 33, enhancement of OSs has been

As discussed in response to Reviewer comment # 55, enhancement of OSs has been demonstrated to result from an increase of the aerosol acidity (cited references) and our work is consistent with these studies.

36. Lines 491-496: The "novel pathway" involving reactive uptake of hydroperoxides is not well- substantiated in the current work and is mostly speculation. The vapor pressure alone of the carbonyl hydroperoxide makes it a potential candidate to partition to the particle phase, not via reactive uptake. There are no direct measurements of hydroperoxides in the gas phase, and insufficient discussion on if hydroperoxides are detected in the particle phase using the UPLC technique. Further, if reactive uptake is at play, why have the authors not seen the corresponding decalin analog of carbonyl hydroperoxide?

We agree with reviewer #3 that "reactive uptake" of hydroperoxides is currently not well substantiated, and have clearly indicated that this pathway is tentative. With regard to reactive uptake and perhydrolysis of carbonyl hydroperoxides generally as an alternative pathway the revised manuscript cites this route as a possibility in the formation of 8 OSs, as indicated in response to earlier comments. Also as discussed above in response to Reviewer comment #14, LC-EI/MS in the negative ion mode, used to identify the OS products, will not detect analytes (such as hydroperoxides) not containing substituents readily yielding negative ions.

**Minor Comments:**

 Line 30: "Both studies strongly support formation of OSs" is awkward. Reword, for example, "Both studies strongly support that OSs can form from the gas-phase oxidation of anthropogenic precursors..."

The wording has been revised on lines 30-33 as follows:

"Both studies strongly support the formation of OSs from the gas-phase oxidation of anthropogenic precursors, as hypothesized on the basis of recent field studies in which aromatic and aliphatic OSs were detected in fine aerosol collected from several major urban locations."

Line 48: Change "aerosol" to "particles", as aerosol is technically defined as both gas + particle.

The wording has been revised as suggested.

- Line 76: Insert comma after "2015)".
   A comma has been inserted.
- Line 83: Change comma to semi-colon after "2007,".
   A semi-colon has been inserted.
- Line 90: Delete "of".
   Use of "of" is appropriate and we have not made this change.
- Line 103: Change "reduce" to "reduces".
   The revision has been made as suggested.
- Line 136: Check misprint on the high humidity range listed as "(4-60%)". The correction has been made (i.e. 40-60%).
- Line 220: Insert after ")", ", hereafter referred to as OS-279, ". The change has been made as suggested.
- 9. Line 268: "ion at m/z 265.0749" should be "ion at m/z 265.0752" according to Figure
  2.
  "at" has been inserted.
- 10. Line 315: Add in "Figure 3, pathway a" to be clear."Figure 3, pathway a" has been inserted.

- 11. Line 345: Change "identical" to "analogous" as the sequence of reactions are certainly not identical as shown in Figure 4."Analogous" has been substituted.
- Line 350: Change chemical formula to include S for OS-281.
   The formula has been corrected.
- 13. Line 352: Rewrite the sentence. The radical reacts with O2, followed by 1,6 H shift, etc.

Sentence has been modified on lines 319-325 to:

"The pathway proposed in Figure S8 pathway **b** is based on gas-phase oxidation of a 4-(cyclohexan-2-one)but-1-yl radical followed by reaction with  $O_2$  and a 1,5-H shift (Crounse et al., 2011; Orlando and Tyndall, 2012) and lead to a  $C_{10}$ -carbonyl-hydroxyhydroperoxide ( $C_{10}H_{18}O_4$ ).  $C_{10}H_{18}O_4$  could then further react with OH radical and by elimination of OH lead to an epoxide (Figure S8, pathway **b**). In addition, OS-281 could arise from acid-catalyzed perhydrolysis of  $C_{10}$ -carbonyl dihydroperoxides ( $C_{10}H_{18}O_5$ ) as proposed in Figure S8, pathway **c**."

14. Lines 362-363: Rewrite the awkward phrasing, "which be reactively taken up to give a sulfate ester".

Sentence has been changed.

Lines 331-334: "However, in contrast to pathway **b**,  $RO_2$  formed by the addition of  $O_2$ undergoes a 1,6-H shift (Crounse et al., 2011; Orlando and Tyndall, 2012) followed by addition of a second  $O_2$  molecule, a 1,5-H shift and elimination of OH to yield an epoxide, which leads to a sulfate ester by reactive uptake onto acidified aerosols."

15. Lines 371-374: Poor grammar. Rewrite sentence.

The sentence has been changed on lines 419-422 to:

"None of the fragment ions observed in the MS2 spectrum suggests the presence of a terminal carbonyl or a carboxyl functional group in the cyclodecane-OSs, which is consistent with conservation of the cyclodecane ring."

16. Line 379: "hydroperoxydes" is spelled wrong. The spelling has been corrected.

31

- 17. Line 459: Add "of" after "oxidation"."of" has been added.
- Line 482: Add "," after "cyclododecane".
   The comma has been added.
- 19. Check Table 1 entry for Decalin Acidified Seed RH Range of 51-49% Initial HC 180 ppb Is RH range correct?Yes the RH range is correct.

**Additional References:**

Atkinson, R.; Arey, J.; Aschmann, S. M. Atmospheric Chemistry of Alkanes: Review and Recent Developments. Atmos. Environ. 2008, 42 (23), 5859–5871.

Crounse, J. D.; Paulot, F.; Kjaergaard, H. G.; Wennberg, P. O. Peroxy Radical Isomerization in the Oxidation of Isoprene. Phys. Chem. Chem. Phys. 2011, 13 (30), 13607–13613.

Dibble, T. S. Cyclization of 1,4-Hydroxycarbonyls Is Not a Homogenous Gas Phase Process. Chem. Phys. Lett. 2007, 447 (1-3), 5–9.

Lambe, A. T.; Onasch, T. B.; Croasdale, D. R.; Wright, J. P.; Martin, A. T.; Franklin, J. P.; Massoli, P.; Kroll, J. H.; Canagaratna, M. R.; Brune, W. H.; et al. Transitions from Functionalization to Fragmentation Reactions of Laboratory Secondary Organic Aerosol (SOA) Generated from the OH Oxidation of Alkane Precursors. Environ. Sci. Technol. 2012, 46 (10), 5430–5437.

Lim, Y. Bin; Ziemann, P. J. Chemistry of Secondary Organic Aerosol Formation from OH Radical-Initiated Reactions of Linear, Branched, and Cyclic Alkanes in the Presence of NOx. Aerosol Sci. Technol. 2009a, 43, 604–619. Lim, Y. Bin; Ziemann, P. J. Kinetics of the Heterogeneous Conversion of 1,4-Hydroxycarbonyls to Cyclic Hemiacetals and Dihydrofurans on Organic Aerosol Particles. Phys. Chem. Chem. Phys. 2009b, 11 (36), 8029–8039.

Orlando, J. J.; Tyndall, G. S. Laboratory Studies of Organic Peroxy Radical Chemistry: An Overview with Emphasis on Recent Issues of Atmospheric Significance. Chem. Soc. Rev. 2012, 41 (19), 6294.

Peeters, J.; Nguyen, T. L.; Vereecken, L. HO X Radical Regeneration in the Oxidation of Isoprene W. Phys. Chem. Chem. Phys. 2009, 11, 5935–5939.

[revised manuscript text omitted]